# Dynamic Configuration for Cutting Plane Separators via Reinforcement Learning on Incremental Graph

**Mingxuan Ye**[1], **Jie Wang**[1][*], **Fangzhou Zhu**[2], **Zhihai Wang**[1], **Yufei Kuang**[1], **Xijun Li**[3],

**Weilin Luo**[2], **Jianye Hao**[2,4], **Feng Wu**[1]

[1] MoE Key Laboratory of Brain-inspired Intelligent Perception and Cognition,
University of Science and Technology of China
[2]Noah's Ark Lab, Huawei      [3]Shanghai Jiao Tong University      [4]Tianjin University
[1]{mingxuanye,yfkuang,zhwangx}@mail.ustc.edu.cn, {jiewangx,fengwu}@ustc.edu.cn
[2]{zhufangzhou,luoweilin3,haojianye}@huawei.com
[3]{lixijun}@sjtu.edu.cn

## Abstract

Cutting planes (cuts) are essential for solving mixed-integer linear programming (MILP) problems, as they tighten the feasible solution space and accelerate the solving process. Modern MILP solvers offer diverse cutting plane separators to generate cuts, enabling users to leverage their potential complementary strengths to tackle problems with different structures. Recent machine learning approaches learn to configure separators based on problem-specific features, selecting effective separators and deactivating ineffective ones to save unnecessary computing time. However, they ignore the dynamics of separator efficacy at different stages of cut generation and struggle to adapt the configurations for the evolving problems after multiple rounds of cut generation. To address this challenge, we propose a novel **dyn**amic **sep**arator configuration (**DynSep**) method that models separator configuration in different rounds as a reinforcement learning task, making decisions based on an incremental triplet graph updated by iteratively added cuts. Specifically, we tokenize the incremental subgraphs and utilize a decoder-only Transformer as our policy to autoregressively predict when to halt separation and which separators to activate at each round. Evaluated on synthetic and large-scale real-world MILP problems, DynSep speeds up average solving time by 64% on easy and medium datasets, and reduces primal-dual gap integral within the given time limit by 16% on hard datasets. Moreover, experiments demonstrate that DynSep well generalizes to MILP instances of significantly larger sizes than those seen during training. The code is released at `https://github.com/MIRALab-USTC/L2O-DynSep`.

## 1 Introduction

Mixed-Integer Linear Programming (MILP) problems are linear programs that involve both discrete and continuous decision variables, which have been widely used in real-world optimization tasks [1–3]. A standard MILP problem has the form:

$$z^* \overset{\triangle}{=} \min_{\mathbf{x}}\{\mathbf{c}^\top\mathbf{x} \mid \mathbf{A}\mathbf{x} \le \mathbf{b}, \mathbf{x} \in \mathbb{R}^n, x_j \in \mathbb{Z} \text{ for all } j \in \mathrm{I}\}, \tag{1}$$

where $\mathbf{c} \in \mathbb{R}^n, \mathbf{A} \in \mathbb{R}^{m \times n}, \mathbf{b} \in \mathbb{R}^m, \mathrm{I} \subseteq \{1, \dots, n\}$ indexes those variables constrained to be integral, and $z^*$ denotes the optimal objective value of the problem in (1).

---

[*]Corresponding Author

39th Conference on Neural Information Processing Systems (NeurIPS 2025).

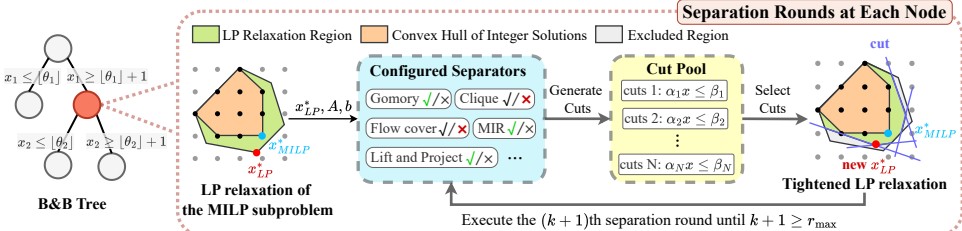

Figure 1: **Separation rounds at each node of B&B tree**. At each tree node, the solver first solves the current LP relaxation. Based on the LP solution $x^*_{LP}$ and the current constraints, a suite of configured separators–of which only the activated ones are invoked–generates a pool of candidate cuts. The solver then selects the most promising cuts, adds them to the model, and re-optimizes the tightened LP relaxation. This separation cycle repeats until a preset maximum round $r_{max}$ is reached.

Modern MILP solvers typically follow the Branch-and-Cut (B&C) paradigm [4, 5], which unifies Branch-and-Bound (B&B) tree search with cutting-plane techniques into a cohesive framework. At each B&B tree node, the solver forms a Linear Programming (LP) relaxation by relaxing integer constraints and solves it to yield a dual bound for the current node. To tighten the dual bound, the solver runs multiple *separation rounds*: find inequalities satisfied by all integer-feasible solutions but violated by the current LP solution, add them as *cutting planes*, re-solve the LP, and repeat–thereby pushing the relaxation closer to the convex hull of the integer points. As shown in Fig. 1, each separation round invokes several cut-generation algorithms (i.e., *separators*), each focuses on a particular family of inequalities (e.g., Gomory cuts, Clique cuts). The solver then selects a small, diverse set of the most effective cuts and adds them to the LP model. These separators are critical to performance because they determine the quality of candidate cuts and thus the solver's convergence behavior. Modern MILP solvers offer multiple separators with tunable configurations, enabling users to flexibly call different separators and combine their complementary strengths to tackle problems with different structures.

To fully exploit separators' potential for speeding up the B&C process, recent machine learning (ML) approaches [6, 7] learn problem-aware configurations and disable ineffective separators to save unnecessary computing time. However, they ignore how separator efficacy changes across cut-generation stages, struggling to adapt configurations as the problem evolves across nodes and separation rounds. We identify two main drivers of this dynamics: (i) decaying marginal gains from successive separation rounds and (ii) interaction effects among separators. First, classic separators (e.g., Gomory, split) exhibit diminishing marginal improvements in the dual bound and introduce potential numerical issues as rounds accumulate [8–10]. Our pilot study (Fig. 2(a)) reveals that solver performance is highly sensitive to the round cutoff–more rounds do not necessarily yield better performance. Second, interactions among separators affect their joint benefit: some separator families yield strong cuts in early rounds, whereas others act as late-stage boosters; some separators provide mutual reinforcement to tighten bounds [11], while others create redundancy or dilute strength when used together [12, 13]. Our experiment that randomizes the activation status of separators at each separation round (Fig. 2(b)) confirms that proper round-aware configurations can deliver substantial performance gains, underscoring the need for dynamic configuration of separators.

Inspired by the analysis above, we propose **DynSep**, a novel **dyn**amic **sep**arator configuration method that models separator configuration in different rounds as a reinforcement learning (RL) task and jointly decides when to halt separation and which separators to activate in each round at settled B&C nodes. To avoid the overhead of re-encoding the entire MILP subproblem in each round, DynSep ingests only the incremental subgraph—the triplet graph formed by newly added cuts–at each decision-making step of the RL agent. Specifically, we first extract graph and node embeddings for each incremental subgraph using a Graph Convolutional Network (GCN) and tokenize these embeddings. We then feed the tokens produced at the current time step into a decoder-only Transformer, which integrates temporal context from past subgraphs and autoregressively predicts the next round's separator configurations. Furthermore, we introduce a *blocked positional encoding* for the Transformer that captures the temporal ordering of the incremental subgraphs while omitting the incidental order of feature tokens within each subgraph. This design ensures that DynSep performs a one-to-one mapping from each separator's features to its configuration decisions in a permutation-equivariant manner, which preserves token-level alignment. Our experiments show that DynSep outperforms other state-of-the-art (SOTA) configuration methods on benchmarks of both

synthetic and large-scale real-world MILP problems. Specifically, DynSep speeds up average solving time by 64% on five easy and medium datasets and reduces average primal-dual gap integral by 16% on four hard datasets, including benchmarks from MIPLIB 2017 [14] and large-scale real-world production planning problems. Moreover, our tests show that DynSep generalizes well to unseen larger MILP instances, and the visualization study shows it truncates unnecessary rounds and automatically captures some known facts of separator efficacy patterns.

## 2 Related Work

Learning-based MILP solvers are now commonplace across industry and academia, spanning industrial solver pipelines [15], solution prediction for warm-starting [16, 17], and accaleration of core solver modules–e.g., cut selection and branching–via machine learning [18, 19]. Building on this landscape, we target ML-based cut generation. Existing data-driven methods for guiding cut generation can be categorized into three classes: generating a parameterized family of cuts [20–23], enhancing existing separators [24, 25], and configuring separators to generate compound cuts [6, 7]. Our work falls into the third category. Related work on ML-driven separator configuration includes L2Sep [6] and LLM4Sep [7]. L2Sep uses a greedy filter to restrict the vast separator parameter space and trains an instance-wise bandit model to guide separator activation. L2Sep can adjust configurations in a few intermediate separation rounds, but requires training a separate model for each round, thus limiting its scalability to arbitrary rounds and node-wise adaptation. LLM4Sep harnesses the large language model (LLM) to generate cold-start configurations for each dataset. While both methods yield notable gains, they remain confined to small or truncated parameter spaces, and ignore the dynamic efficacy and order dependencies of separators, which restricts their capacity for on-the-fly adaptation as the solver progresses.

## 3 Background

**Branch-and-cut.** In B&C, the solver builds a search tree by recursively branching on fractional variables, thus partitioning the MILP into smaller subproblems at each node. Each subproblem is solved via its LP relaxation, yielding an optimal fractional solution that provides a lower bound (dual bound) on the objective of the subsequent subtree. To tighten this relaxation, cutting planes—linear inequalities of the form $\alpha \mathbf{x} \leq \beta$—are iteratively added to the LP problem. As shown in Fig. 1, these cuts strategically remove fractional solutions from the LP relaxation region without eliminating any feasible integer solutions, thus raising the dual bound at the node. Any subtree whose dual bound exceeds the best-known integer solution can be pruned; thus, cutting planes are crucial in reducing the search space and accelerating the solving process.

**Reinforcement Learning.** A standard MDP is formally defined as a tuple $\mathcal{M} := \langle \mathcal{S}, \mathcal{A}, \mathcal{R}, \mathcal{P} \rangle$, where $\mathcal{S}$ is the state space, $\mathcal{A}$ is the action space, $\mathcal{R} : \mathcal{S} \times \mathcal{A} \rightarrow \mathbb{R}$ is the reward function, and $\mathcal{P} : \mathcal{S} \times \mathcal{A} \rightarrow \mathcal{S}$ is the transition function, At each time step $t$, the agent observes the current state $s_t \sim \mathcal{S}$ and generates an action $a_t \in \mathcal{A}$ from its policy $\pi(s_t)$. This action pushes the environment towards the next state $s_{t+1}$, accompanied by a scalar reward $r_t = R(s_t, a_t)$. This interactive process yields an episode of sequence $\tau := (s_0, a_0, r_0, s_1, a_1, \cdots, s_T, a_T, r_T)$, in which T means the episode ends or reaches the maximum time step. The goal of RL is to find an optimal policy $\pi^*$ that maximizes the expected cumulative rewards $\mathbb{E}_{\tau \sim (\pi, \mathcal{M})} \left[ \sum_{t=0}^{T} \gamma^t r_t \right]$.

## 4 Motivation

**Set maximum separation round to decide when to halt.** Empirical results in 2(a) show that altering the maximum separation round $r_{\max}$ for each node causes significant performance variations. For the setcover dataset, the solving process accelerates when increasing $r_{\max}$ to 3, after which further rounds cause degradation. For the anonymous dataset, both solving metrics worsen monotonically as $r_{\max}$ increases. Overall, these trends indicate that adding separation rounds does not universally improve performance, and the optimal value of $r_{\max}$ differs across datasets. These findings motivate us to incorporate the maximum separation round as a decision variable for the agent.

**Set activation status to decide which separator to activate.** In Fig. 2(b), we test 50 random configurations for each instance in several datasets and choose the best configuration specific to each

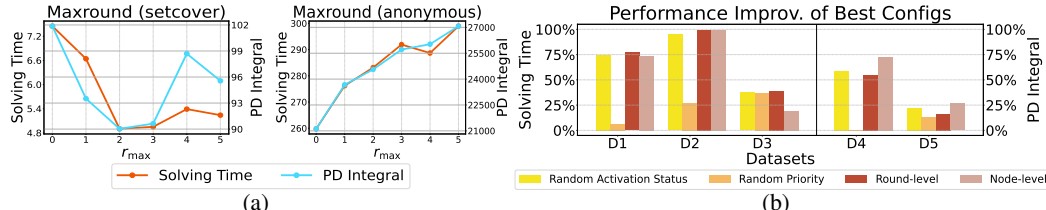

Figure 2: **Motivation results**. **(a)** Effect of varying maximum round $r_{\max}$ on solver performance for two MILP datasets. Each plot shows the average solving time (red line, left y-axis) and PD integral (blue line, right y-axis) across all instances in the dataset. **(b)** Performance improvement of the best configurations found by different random strategies. Datasets D1–D3 use solving time (left) as the metric, while D4–D5 use PD integral (right). The y-axis represents the relative improvement compared to the default setting. Each bar represents a specific strategy to get random configurations.

instance, which illustrates the potential performance gains from different configuration strategies. The yellow bars correspond to randomly varying the activation status of all separators. Specifically, each separator's activation status $\eta_i \in \{-1, 0, +1\}$ is randomly chosen, where 0 indicates deactivation, +1 indicates eager activation, and $-1$ indicates deferred activation. Thus, for each selected configuration, we randomly decide which separators are active and in which phase they are invoked. We provide more implementation details of separator configurations in Appendix B.1. We also randomize the separators' priorities $\{p_i\}_{i=1}^{K}$, which controls the execution orders of the separators in a separation round. As shown by the orange bar in Fig. 2(b), perturbation in priority configs exhibits a minor effect on performance improvement compared to activation status (yellow bars) changes (see more results and analyses in Appendix E.1). These findings motivate us to alter the separators' activation status to decide the order of separator invocation across multiple separation rounds.

**Node-wise&Round-wise Dynamic configurations.** The last two bars of Fig. 2(b) correspond to randomizing the activation status of all separators either at each separation round (red: round-level) or at nodes with regular depths (pink: node-level). Both round- and node-level random configurations show great potential in finding high-quality configurations. This indicates structural shifts exist in the evolving problems when moving to a new separation round or a new node, which leads to the shift of optimal separator configurations. These findings motivate us to configure separators in a more fine-grained manner, adjusting them dynamically in each separation round at multiple nodes.

## 5 Method

Cutting-plane separator configuration is challenging due to the vast combinatorial configuration space and the dynamic cutting preference at different stages of cut generation. Therefore, we model the dynamic separator configuration as a sequential decision-making problem. We employ RL to explore the large action space and leverage performance feedback from the MILP solving process to guide the search for optimal configurations. In Section 5.1, we present the detailed RL formulation for separator configuration. Sections 5.2& 5.3 offer an in-depth description of our proposed DynSep, detailing how to model time-evolving MILP instances via dynamic triplet graphs and how to process the tokenized features using a decoder-only Transformer. Section 5.4 outlines the training process for DynSep. The overall workflow of DynSep is illustrated in Figure 3.

### 5.1 Reinforcement Learning Formulation

In the following, we define the MDP components for dynamic separator configuration.

**The state space $\mathcal{S}$.** Given that the entire MILP problem determines the geometric structure of the feasible solution domain—which in turn provides the core information for cut generation—we include the MILP problem $M_t$ to be tackled in the $t$-th separation round in our state. We also embed per-separator solving feedback (e.g., cost, contribution in finding proper cuts and reducing variable domains, etc.), which guides the agent to learn the overhead and efficacy of each separator. Therefore, following the work [6], we model the above inputs as a triplet graph $G = (V, C, S; \mathcal{E})$ with three

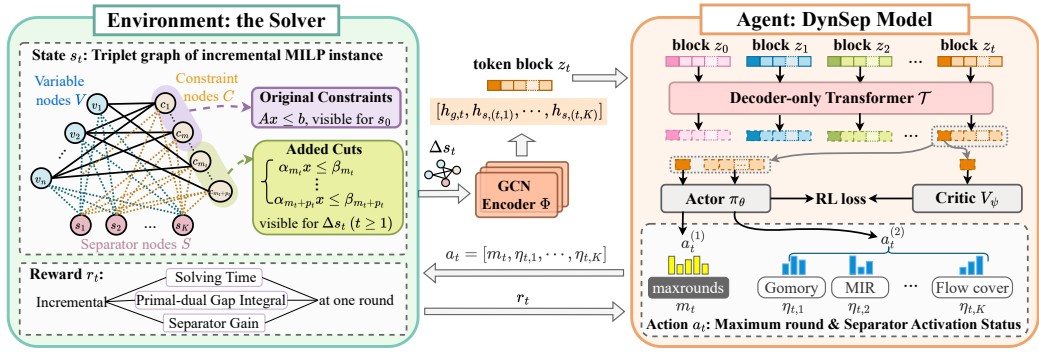

Figure 3: **The RL framework for dynamic separator configuration**. The environment (left) provides an incremental triplet graph $\Delta s_t$ of the MILP instance at each time step. A GCN encoder processes $\Delta s_t$, extracting a token block $z_t$ consisting of a graph-level embedding and $K$ separator node embeddings. A decoder-only Transformer then processes the sequence of such token blocks autoregressively. Finally, the actor $\pi_\theta$ outputs the corresponding categorical distributions of round-wise decisions: the maximum round and all separators' activation statuses. These decisions are optimized via RL loss, with rewards computed from round-level improvements in three aspects.

types of nodes: variable nodes $V$, constraint nodes $C$, and separator nodes $S$. $\mathcal{E}$ denotes the edge set. Detailed node and edge feature definitions appear in Appendix B.2.

Formally, we define the state as the entire triplet graph, i.e., $s_t = G_t$. However, encoding the full graph in each decision step leads to expensive gradient updates during training and a growing inference time as the graph scales. To mitigate this, we instead use the incremental state $\Delta s_t$ as the input at the $t$-th time step. Specifically, $\Delta s_t = \Delta G_t$ captures only the updates after each cut generation round, including the newly added cuts, tightened variable bounds, and updated separator statistics. In practice, $\Delta s_t$ is a triplet subgraph reflecting those changes (see the left panel of Figure 3), and we set $\Delta s_0 = G_0$ using original constraints as constraint nodes.

**The action space $\mathcal{A}$.** Inspired by the motivational results in Section 4, we define the action $a_t$ as the concatenation of two decision components. First, we decide which round to stop generating cuts by $a_t^{(1)} = m_t \in \{1, 2, \cdots, T\}$, where $m_t$ denotes the maximum number of separation rounds permitted at the current node and $T$ denotes the predefined upper limit for separation rounds in the overall B&C process (i.e., the maximum episode length in our RL formulation). Second, we decide which separators to activate in early and late phases by $a_t^{(2)} = \eta_t = (\eta_{t,1}, \ldots, \eta_{tK}) \in \{-1, 0, 1\}^K$, where $\eta_t$ denotes the activation-status vector for the $K$ built-in separators. Each entry of $\eta_{t,i}$ specifies the activation status of the $i$-th separator that would be setup at the round $t$, taking values of $+1$ (activated early), $0$ (deactivated), or $-1$ (activated late). Formally, we represent the overall action at time $t$ as:

$$a_t = \texttt{concat}(a_t^{(1)}, a_t^{(2)}) = [m_t, \eta_{t,1}, \ldots, \eta_{t,K}] \in \mathbb{N}^{K+1} \tag{2}$$

These configuration decisions are made at every separation round for nodes at predefined depths.

**The reward function $\mathcal{R}$.** To leverage the rich feedback dropped in the solving process, we define the reward $r_t$ as a summation of three incremental metrics: runtime penalty, relative dual bound improvement, and separator effectiveness. After the $t$-th round, the agent observes

$$r_t = -\left[\Delta T_t - \frac{1}{N} \cdot T_{\text{base}}\right] + \frac{\Delta B_t}{B_{t-1}} + \Delta \kappa_t, \tag{3}$$

where $\Delta T_t = T_t - T_{t-1}, \Delta B_t = B_t - B_{t-1}, \Delta \kappa_t = \kappa_t - \kappa_{t-1}$. In the first term, $T_t$ denotes the cumulative solving time up to the $t$-th round, $T_{\text{base}}$ is the problem-specific time budget estimated under the default configuration, and $N$ is the total number of separation rounds in an entire B&C process. In the second term, $B_t$ denotes the current dual bound obtained from the optimal solution of the current LP relaxation. Note that we normalize the incremental solving time by its estimated average per round and express the dual-bound improvement as the ratio of the increase to the previous bound, since both metrics can vary greatly across MILP instances and may become excessively large, which would risk training instability. In the third term, $\kappa_t$ signifies the aggregate gain caused by all active separators at

round $t$, which is computed as the sum of: (i) the application rate of the generated cuts, (ii) numbers of domain reduction operations, (iii) numbers of node pruning operations. These three statistics provide meaningful insights into the separators' contribution to domain tightening and search-space pruning (see Appendix B.3 for more details).

**The environment transition $\mathcal{P}$.** The transition function maps the current state and action to the next state. That is, by performing the configuration decision $a_t$, a round of cuts is generated from the current triplet graph $s_t$. After incorporating all selected cuts, the solver forms an incremental triplet graph $s_{t+1}$. We denote the incremental updates as $\Delta s_{t+1}$, which represents a subgraph consisting of newly added cut nodes, updated variable and separator nodes, and newly connected edges.

**The terminal state.** At the first separation round of each node, the agent provides a decision of maximum round $m_0$. Then the solver environment terminates cut generation after $m_0$ rounds or other built-in stopping conditions are met (see Appendix B.4 for more details).

## 5.2 Tokenize Incremental subgraphs

We extract latent embeddings for both the entire graph and its separator nodes using the encoder network $\Phi$ introduced by [6]. The encoder $\Phi$ involves a Graph Convolutional Network (GCN) [26], an attention block on the hidden embeddings of the separator nodes [27], and a global pooling layer that produces a single representation for the input graph. Full details of the message-passing operations are provided in Appendix C; here, we focus on the tokenization procedure that is unique to our approach.

At each time step $t$, feeding the incremetal subgraph $\Delta s_t = \Delta G_t$ into the encoder $\Phi$ yields a graph feature $h_{g,t} \in \mathbb{R}^d$ and $K$ separator node features $h_{s,(t,1)}, \ldots, h_{s,(t,K)} \in \mathbb{R}^d$. We treat each feature vector as a token, and then we map the incremental subgraph $\Delta G_t$ into a stacked *token block*:

$$z_t = \begin{bmatrix} h_{g,t} & h_{s,(t,1)} & \cdots & h_{s,(t,K)} \end{bmatrix}^\top \in \mathbb{R}^{(K+1)\times d}. \tag{4}$$

## 5.3 Decision-Maker: Decoder-only Transformer

The token block $z_t$ is an embedding of the incremental state $\Delta s_t$, containing incremental contexts and separator statistics updated in the last separation round. To recover the entire information of state $s_t$ for informed decision-making, we aggregate the information of all history incremental states $\Delta s_i (0 \leq i \leq t)$ via a decoder-only Transformer $\mathcal{T}$.

**Decision-making in an autoregressive fashion.** At each round $t$, we assemble the history of token blocks as an ordered sequence $w_t = \langle z_0, z_1, \ldots, z_t \rangle$. We input $w_t$ into the Transformer $\mathcal{T}$ and train $\mathcal{T}$ to predict the corresponding configuration actions $\langle a_0, a_1, \ldots, a_t \rangle$ in an autoregressive fashion, where $a_i = [m_i, \eta_{i,1}, \ldots, \eta_{i,K}]$ for $i = 0 : t$. To optimize decision quality in an online RL manner, we replace the common prediction error loss in Transformer models with the RL objective that maximizes the cumulative return over the generated actions. During inference, we feed the historical sequence $w_t$ to $\mathcal{T}$ and generate the next action $a_t$, like the next-token generation in language modeling.

**Blocked positional encoding.** We introduce a blocked positional encoding (PE), enabling the Transformer model to recognize and capture the temporal ordering among token blocks, while omitting any ordering among tokens within each block. Formally, we define the blocked PE of one episode as

$$P = \begin{bmatrix} P_0^\top & P_1^\top & \cdots & P_T^\top \end{bmatrix}^\top \in \mathbb{R}^{(T+1)(K+1)\times d}, \tag{5}$$

where $P_i = [\mathrm{pos}(i) \ \mathrm{pos}(i) \ \cdots \ \mathrm{pos}(i)]^\top \in \mathbb{R}^{(K+1)\times d}$, $\mathrm{pos}(i) \in \mathbb{R}^d$. That is, the blocked PE assigns the same positional encoding $\mathrm{pos}(i)$ to all $K + 1$ tokens in $z_i$. Then the input to $\mathcal{T}$ becomes $\widetilde{w}_t = \langle z_0 + P_0, \ldots, z_t + P_t \rangle$. This design mirrors relative-position schemes that bias attention at the block level while leaving intra-block tokens interchangeable. Similarly, we use a blocked causal mask that forbids attention from block $i$ to any future block $j > i$. At inference time, we take the last $K + 1$ hidden embeddings as an embedding block to predict the next action.

**Permutation equivariance inside each block.** Note that the token block $z_t$ in Eq. (4) and the action in Eq. (2) are consistent in both structure and contextual meaning. Thus, we utilize the embeddings of $K + 1$ tokens in $z_t$ after passing the Transformer $\mathcal{T}$ to obtain the $K + 1$ probability distributions of $K + 1$ configuration components in action $a_t$, respectively. We formalize the *one-to-one decision*

*mapping* as follows:

$$\text{block-level: } \langle a_0, \ldots, a_t \rangle = \mathcal{T}(\langle z_0 + P_0, \ldots, z_t + P_t \rangle) \tag{6}$$

$$\text{token-level: } a_i = [m_i, \eta_{i,1}, \ldots, \eta_{i,K}] = \mathcal{T}([h_{g,i}, h_{s,(i,1)}, \cdots, h_{s,(i,K)}]) = \mathcal{T}(z_i + P_i). \tag{7}$$

In Eq. (7), the first token $h_{g,i}$ of each block $z_i$ informs the global mode of subgraph $\Delta s_i$ to decide which round $m_i$ to stop. The subsequent $K$ tokens $h_{s,(i,1)}, \ldots, h_{s,(i,K)}$ guide separator configs $\eta_{t,i}, \ldots, \eta_{i,K}$. We visualize the data flow of block-level and token-level decision mapping in the right panel of Fig. 3. Such a one-to-one mapping encourages the agent to make context-aware decisions and is guaranteed by the permutation equivariance of the self-attention mechanism when the input tokens share the same position embedding. We formulate such permutation equivariance as follows and provide the corresponding proof in Appendix A.

**Proposition 1.** *The decoder-only Transformer $\mathcal{T}$, equipped with the blocked positional encoding $P$, is permutation equivariant inside each token block. Formally, for any input $X$ and any block-wise permutation matrix $\Pi$, it holds that $\mathcal{T}(\Pi X + P) = \Pi \mathcal{T}(X + P)$.*

### 5.4 Training

We utilize the Proximal Policy Optimization (PPO) algorithm [28] to train our decision network, as PPO strikes a balance between ease of implementation and computational efficiency. PPO inherits from the famous actor-critic framework in which a critic $V_\psi(s)$ estimates the state value function, while an actor $\pi_\theta$ determines the policy. We instantiate two separate decoder-only Transformers for the actor and critic network, respectively, each sharing the architecture defined in Section 5.3. As shown in the right panel of Figure 3, $V_\psi$ only takes the first token (the graph feature) of each block $z_i$ from the Transformer's final hidden embedding and then feeds it into a linear head to predict the state value. The actor $\pi_\theta$ concatenates the Transformer's final token embeddings and feeds them into a joint linear head to produce the probability distributions of all configuration decisions. We provide the algorithm pseudocode in Appendix D.1 and the architecture hyperparameters in Appendix D.2.

## 6 Experiments

### 6.1 Experimental Setup

**Setup.** We use SCIP 8.0.0 [29] as the backend solver, which is a state-of-the-art open source solver widely adopted in the research of ML for combinatorial optimization (ML4CO) [30, 31, 18, 32]. To maintain fair comparison and reproducibility, we retain all the other SCIP parameters as default in all baselines and our method. All of the SCIP solver's advanced features, such as presolve and heuristics, are open, which ensures that our setup is consistent with the practice setting. Evaluation on each instance is limited to a 300-second solving time. To balance performance and computational cost, our approach configures separators at every tenth depth level in the B&B tree by setting the separator frequency in SCIP as 10. We set the maximum separation round at each node as $T = 5$. Configuration is applied to 22 separators built into SCIP; more descriptions of separators are provided in Appendix B.5. More details about the ML setup and hardware specifications are provided in Appendix D.

**Benchmarks.** We evaluate our approach on nine publicly available NP-hard MILP problem benchmarks from the prior work [33], covering three classical synthetic MILP problems and six challenging MILP problems from diverse application areas. The nine problem benchmarks are divided into three categories (easy, medium, and hard) according to the difficulty of solving them using the SCIP 8.0.0 solver. (1) **Easy datasets**: three widely used synthetic MILP problem benchmarks: Set Covering [34], Maximum Independent Set (MIS) [35], and Multiple Knapsack [36]. (2) **Medium datasets**: CORLAT [37] and MIK [38], which are widely used benchmarks for evaluating MILP solvers [39, 40]. (3) **Hard datasets** include two datasets from the ML4CO NeurIPS 2021 Competition [41]: the Load Balancing problem inspired by real-life applications of large-scale systems, and the Anonymous problem inspired by a large-scale industrial application. Moreover, hard datasets contain MIPLIB mixed neos and MIPLIB mixed supportcase, two subsets of the benchmark MIPLIB 2017 [14].

**Baselines.** Our baselines include four human-designed separator configuration rules and three learning-based methods. Human-designed rules include (1) **Nocuts**: pure B&B without adding any

Table 1: **Comparative evaluation on easy, medium, and hard datasets.** Best performance is in bold, with the greatest improvement (Improv.) both bolded and underlined. Sizes of nine benchmarks are in parentheses, with $n$ and $m$ representing the average numbers of variables and constraints, respectively. The values report the mean (standard deviation) of time and PD integral metrics.

| | Easy: Set Covering ($n = 1000$, $m = 500$) | | | Easy: Max Independent Set ($n = 500$, $m = 1953$) | | | Easy: Multiple Knapsack ($n = 720$, $m = 72$) | | |
|---|---|---|---|---|---|---|---|---|---|
| Method | Time(s) ↓ | Improv. ↑ (time, %) | PD integral ↓ | Time(s) ↓ | Improv. ↑ (time, %) | PD integral ↓ | Time(s) ↓ | Improv. ↑ (time, %) | PD integral ↓ |
| NoCuts | 7.45 (5.87) | NA | 101.86 (55.59) | 15.32 (5.82) | NA | 146.4 (56.99) | 13.84 (28.79) | NA | 25.21 (26.6) |
| Default | 5.24 (1.79) | 29.66 | 95.56 (36.86) | 30.4 (8.02) | -98.43 | 289.51 (103.81) | 2.01 (1.82) | 85.48 | 18.6 (10.49) |
| Search(50) | 1.7 (0.44) | 77.18 | 36.77 (8.48) | 4.42 (3.89) | 71.15 | 23.28 (19.02) | 4.12 (5.52) | 70.23 | 13.38 (7.36) |
| Prune | 7.45 (5.14) | 0.00 | 66.83 (45.97) | 5.03 (3.18) | 67.17 | 30.93 (22.39) | 0.64 (0.48) | 95.38 | 9.89 (3.82) |
| L2Sep(R1) | 6.56 (4.35) | 11.95 | 62.12 (39.08) | 5.42 (3.86) | 64.62 | 34.21 (25.97) | 7.45 (12.07) | 46.17 | 12.99 (8.66) |
| L2Sep(R2) | 7.35 (4.88) | 1.34 | 70.00 (43.57) | 5.36 (3.76) | 65.01 | 33.94 (25.33) | 9.14 (12.92) | 33.96 | 14.40 (8.72) |
| LLM4Sepasel | 11.73 (12.09) | -57.45 | 110.73 (91.14) | 5.13 (4.19) | 66.51 | 27.18 (20.17) | 4.8 (5.51) | 65.32 | 17.79 (8.9) |
| DynSep (Ours) | **1.51 (0.27)** | 79.73 | **33.88 (9.34)** | **0.53 (0.20)** | 96.54 | **9.66 (2.40)** | **0.52 (0.24)** | 96.24 | **9.71 (5.39)** |

| | Medium: Corlat ($n = 466$, $m = 486$) | | | Medium: MIK ($n = 413$, $m = 346$) | | | Hard: Anonymous ($n = 37881$, $m = 49603$) | | |
|---|---|---|---|---|---|---|---|---|---|
| Method | Time(s) ↓ | Improv. ↑ (time, %) | PD integral ↓ | Time(s) ↓ | Improv. ↑ (time, %) | PD integral ↓ | Time(s) ↓ | PD integral ↓ | Improv. ↑ (PD Int., %) |
| NoCuts | 74.66 (122.23) | NA | 2687.68 (6209.48) | 190.28 (113.97) | NA | 887.85 (859.76) | 259.77 (75.71) | 21117.12 (9234.01) | NA |
| Default | 111.55 (132.19) | -49.41 | 10573.14 (13070.46) | 16.65 (18.06) | 91.25 | 82.80 (56.24) | 298.92 (4.09) | 27069.58 (4892.8) | -28.19 |
| Search(50) | 55.74 (97.19) | 25.34 | 2910.77 (6585.5) | 24.99 (20.56) | 86.87 | 89.27 (55.85) | 270.68 (65.52) | 24028.68 (9007.57) | -13.79 |
| Prune | 89.09 (125.52) | -19.33 | 2615.71 (5814.74) | 300.01 (0.0) | -57.67 | 2237.28 (1023.8) | 241.75 (100.61) | 17304.91 (9563.3) | 18.05 |
| L2Sep(R1) | 91.14 (124.12) | -22.07 | 3124.07 (6914.50) | 15.50 (17.60) | 91.85 | 61.09 (44.50) | 239.52 (94.82) | 16970.35 (10108.40) | 19.64 |
| L2Sep(R2) | 89.84 (124.30) | -20.33 | 3113.29 (6927.16) | 11.13 (9.09) | 94.15 | **44.69 (25.06)** | 240.54 (93.80) | 16850.57 (10052.83) | 20.20 |
| LLM4Sepasel | 64.03 (110.63) | 14.24 | 2921.73 (6860.21) | 17.94 (17.76) | 90.57 | 85.66 (65.34) | 284.57 (34.79) | 25384.48 (8100.56) | -20.21 |
| DynSep (Ours) | **22.96 (38.93)** | 69.25 | **2233.42 (3868.43)** | **10.99 (9.44)** | 94.22 | 134.15 (44.21) | 241.89 (100.75) | **15656.7 (8996.14)** | 25.86 |

| | Hard: Load Balancing ($n = 61000$, $m = 64304$) | | | Hard: MIPLIB mixed neos ($n = 6958$, $m = 5660$) | | | Hard: MIPLIB mixed supportcase ($n = 19766$, $m = 19910$) | | |
|---|---|---|---|---|---|---|---|---|---|
| Method | Time(s) ↓ | PD integral ↓ | Improv. ↑ (PD Int., %) | Time(s) ↓ | PD integral ↓ | Improv. ↑ (PD Int., %) | Time(s) ↓ | PD integral ↓ | Improv. ↑ (PD Int., %) |
| NoCuts | 300.11 (0.02) | 15093.26 (940.68) | NA | 275.04 (43.23) | 14618.53 (12214.63) | NA | 181.26 (120.25) | 12959.99 (10506.47) | NA |
| Default | 300.14 (0.02) | 15187.19 (936.38) | -0.62 | 282.98 (29.49) | 18500.5 (9386.15) | -26.56 | 244.75 (105.8) | 21561.09 (10434.42) | -66.37 |
| Search(50) | 300.04 (0.05) | 3783.52 (448.59) | 74.93 | 274.23 (44.64) | 15619.98 (11969.47) | -6.85 | 133.36 (131.32) | 10241.17 (10794.69) | 20.98 |
| Prune | 300.07 (0.12) | 10597.31 (671.55) | 29.79 | 249.37 (87.7) | 14464.45 (12569.32) | 1.05 | 158.63 (141.48) | 9827.52 (11433.13) | 24.17 |
| L2Sep(R1) | 300.02 (0.03) | 10548.89 (4474.08) | 30.11 | 242.83 (99.02) | 10383.49 (11808.13) | 28.97 | 162.18 (138.25) | 11318.55 (11796.53) | 12.67 |
| L2Sep(R2) | 300.03 (0.10) | 10860.13 (4348.96) | 28.05 | 242.90 (98.90) | 13989.09 (12116.88) | 4.31 | 166.23 (134.71) | 11489.15 (11849.13) | 11.35 |
| LLM4Sepasel | 300.04 (0.06) | 4769.47 (709.05) | 68.40 | 276.34 (47.32) | 14109.36 (13706.18) | 3.48 | 256.48 (100.88) | 22618.21 (10234.21) | -74.52 |
| DynSep (Ours) | 300.04 (0.08) | **3720.26 (499.37)** | 75.35 | **235.19 (112.26)** | **8511.58 (12413.9)** | 41.78 | **132.50 (130.32)** | **9212.24 (9840.56)** | 28.92 |

cuts; (2) **Default**: default separator configs used in SCIP 8.0.0; (3) **Search($\rho$)**: randomly samples $\rho$ configurations then applies one with the best performance on the validation set; (4) **Prune**: deactivates separators with no contribution during the evaluation on the validation set. For learning-based methods, we compare: (1)**L2Sep(R1)**: L2Sep [6] that learns the configuration only at the first round; (2) **L2Sep(R2)**: L2Sep that learns the configuration only at the first&second rounds; (3) **LLM4Sep** [7]: configure separators via LLM. Please see Appendix D.4 for details of these baselines.

**Evaluation Metric.** We use two widely used evaluation metrics: the average solving time (Time, lower is better), and the average primal-dual gap integral (PD integral, lower is better). We assess different configuration methods by measuring the relative improvement compared to the NoCuts baseline: $\delta(\cdot) = \mathbb{E}_{x \in \mathcal{X}} \left[ \frac{Metric(\text{NoCuts}) - Metric(\cdot)}{Metric(\text{NoCuts})} \right]$, where $\mathcal{X}$ is a given dataset, and $Metric(\cdot)$ reflects the performance of a method under the given metric. For easy and medium datasets, we use solving time as the evaluation metric, as the solver augmented by DynSep can solve all these instances to optimality (i.e., the average of primal-dual gap is zero, see detailed results in Appendix E.2) within the given time limit. For hard datasets where optimality is not always achieved within the time limit, we adopt the PD integral [33, 42, 41, 43] to quantify the cumulative distance between the primal and dual bounds over the solving time, where the primal-dual distance (gap) reflects the solution quality, and the integral over time exhibits the solver's efficiency of converging to the optimal solution.

## 6.2 Experimental Evaluation

**Experiment 1: Comparative Experiments.** Table 1 shows that DynSep significantly outperforms the baselines in both solving time and PD integral. On easy and medium datasets, DynSep accelerates average solving times by 64% compared to the SOTA learning-based baseline. DynSep achieves lower PD integrals than baselines on four out of five easy and medium benchmarks, demonstrating faster convergence enabled by our dynamic configuration method. For MIK, the relatively large PD integral indicates slower convergence in the early stage, but accelerated convergence in the later stages. For CORLAT, We emphasize that DynSep is the only configuration method that solves all instances to optimality with zero primal-dual gaps (see results in Appendix E.2), highlighting the effectiveness of DynSep for fine-grained separator configurations. On hard datasets, DynSep reduces the average PD

Table 2: Ablation study comparing five variants of DynSep on three datasets.

| | Easy: Multiple Knapsack ($n = 720$, $m = 72$) | | | Medium: Corlat ($n = 466$, $m = 486$) | | | Hard: MIPLIB mixed neos ($n = 6958$, $m = 5660$) | | |
|---|---|---|---|---|---|---|---|---|---|
| Method | Time(s) ↓ | Improv. ↑ (time, %) | PD integral ↓ | Time(s) ↓ | Improv. ↑ (time, %) | PD integral ↓ | Time(s) ↓ | PD integral ↓ | Improv. ↑ (PD Int., %) |
| NoCuts | 13.84 (28.79) | NA | 25.21 (26.6) | 74.66 (122.23) | NA | 2687.68 (6209.48) | 275.04 (43.23) | 14618.53 (12214.63) | NA |
| Default | 2.01 (1.82) | 85.48 | 18.6 (10.49) | 111.55 (132.19) | -49.41 | 10573.14 | 282.98 (29.49) | 18500.5 (9386.15) | -26.56 |
| w/o MaxR | 0.67 (0.46) | 95.16 | 10.07 (5.35) | 28.28 (53.99) | 62.12 | 2688.01 (5424.21) | 263.63 (63.0) | 9029.4 (12114.3) | 38.23 |
| w/o TF | 0.64 (0.53) | 95.38 | 9.82 (5.24) | 25.82 (61.58) | 65.42 | 2552.91 (6146.98) | 245.04 (95.2) | 9448.22 (11932.01) | 35.37 |
| w/o DynG | 0.70 (1.23) | 94.94 | 10.03 (5.86) | 28.05 (50.97) | 62.43 | 2635.47 (5086.84) | 270.66 (50.83) | 9233.22 (12003.39) | 36.84 |
| w/o DynG&TF | 0.58 (0.41) | 95.81 | **9.65 (5.28)** | 110.31 (131.6) | -47.75 | 10396.1 (12887.24) | 300.0 (0.0) | 17330.56 (9660.1) | -18.55 |
| w/o BlockPE | 0.93 (1.81) | 93.28 | 11.12 (9.16) | 34.4 (65.33) | 53.92 | 2870.01 (5533.02) | 242.39 (99.79) | **8291.49 (12533.67)** | 43.28 |
| DynSep (Ours) | **0.52 (0.24)** | 96.24 | 9.71 (5.39) | **22.96 (38.93)** | 69.25 | **2233.42 (3868.43)** | 235.19 (112.26) | 8511.58 (12413.9) | 41.78 |

integral by 16% within the 300-second time limit. DynSep significantly accelerates solving time on challenging MIPLIB datasets and shows comparable time on the other two hard datasets. In contrast, other ML-based baselines improve efficiency on some datasets but struggle to maintain performance across all datasets from various scenarios. Search(50) performs well on some instances, but requires evaluating a large number of configurations in advance and suffers from instability across different problem types. We also evaluate the inference latency and memory overhead of our configuration policy in Appendix E.5 and observe that while inference time increases with instance size, it does not compromise the overall efficiency gains achieved by our method. For completeness, we also report two complementary studies in the appendix: evaluation on additional MIPLIB 2017 datasets (Appendix E.3) and an extension of DynSep to broader solver hyperparameters (Appendix E.4), both showing trends consistent with our main results.

**Experiment 2: Ablation Study.** We present ablation studies on Multiple Knapsack, CORLAT, and MIPLIB mixed neos, representing easy, medium, and hard datasets. We provide completed results on nine benchmarks in Appendix E.6.1. To understand the contribution of each component in DynSep, we evaluate the following five variants: (1) **w/o MaxR**: DynSep without the decision-making on the maximum separation round $m_t$ at each node. (2) **w/o TF**: DynSep using a Long Short-Term Memory(LSTM) architecture as the decison model of DynSep, instead of the encoder-only Transformer designed in Section 5.3. (3) **w/o DynG**: DynSep without the design of dynamic graph, instead inputting the entire triplet graph at each separation round. (4) **w/o DynG&TF**: DynSep without the design of dynamic graph, while replacing the Transformer with an LSTM model. (5) **w/o BlockPE**: DynSep without the design of blocked positional encoding, instead using a standard token-level positional encoding that assigns a unique position to each token irrespective of block boundaries. Table 2 shows DynSep overall outperforms the five variants, with comparable results in the few remaining settings, indicating that all components are essential to cope with evolving structures in challenging MILPs. We further report ablations on the encoder architecture and on hyperparameter sensitivity—e.g., separator frequency and maximum separation round—in Appendices E.6.2 and E.6.3.

**Experiment 3: Generaization Tests.** We evaluate the ability of DynSep to generalize across different sizes of MILPs. Following prior works [30, 33], we test the generalization ability of DynSep on Maximum Independent Set (MIS), as we can artificially generate instances with arbitrary sizes for these synthetic MILP problems. we test 4× and 9× larger instances of MIS. Table 3 (right) shows that DynSep significantly outperforms the baselines in terms of the time and the PD integral, demonstrating the superiority of DynSep in generalization capability. We further evaluate cross-domain and general-to-specific generalization; see Appendix E.7.

Table 3: Evaluate the generalization ability of DynSep on MIS.

| | Set Covering ($n = 1000$, $m = 1000$, 2×) | | | Set Covering ($n = 1000$, $m = 2000$, 4×) | | |
|---|---|---|---|---|---|---|
| Method | Time(s) ↓ | Improv. ↑ (time, %) | PD integral ↓ | Time(s) ↓ | Improv. ↑ (time, %) | PD integral ↓ |
| NoCuts | 78.91 (76.53) | NA | 579.09 (507.14) | 282.27 (52.86) | NA | 3109.87 (1013.74) |
| Default | 11.54 (3.55) | 85.38 | 255.69 (92.95) | 19.74 (8.06) | 93.01 | **606.90 (310.79)** |
| Search(30) | 116.95 (92.25) | -48.821 | 912.87 (647.22) | 292.54 (35.54) | -3.64 | 3827.701 (1057.08) |
| Prune | 107.86 (90.55) | -36.69 | 826.38 (660.27) | 295.01 (29.73) | -4.51 | 4805.02 (1302.02) |
| L2Sep(R1) | 105.48 (92.68) | -33.67 | 803.32 (656.89) | 294.85 (31.41) | -4.46 | 4728.1 (1325.35) |
| L2Sep(R2) | 116.43 (95.6) | -47.55 | 905.98 (694.71) | 293.77 (33.96) | -4.07 | 4481.42 (1377.69) |
| LLM4Sepasel | 149.28 (105.45) | -89.18 | 1115.22 (767.04) | 295.4 (29.34) | -4.65 | 3881.22 (1001.51) |
| DynSep (Ours) | **4.03 (0.64)** | 94.89 | **104.62 (20.19)** | **19.05 (26.92)** | 93.25 | 629.28 (1106.36) |

| | Max Independent Set ($n = 1000$, $m = 3946$, 4×) | | | Max Independent Set ($n = 1500$, $m = 5940$, 9×) | | |
|---|---|---|---|---|---|---|
| Method | Time(s) ↓ | Improv. ↑ (time, %) | PD integral ↓ | Time(s) ↓ | Improv. ↑ (time, %) | PD integral ↓ |
| NoCuts | 195.53 (95.78) | NA | 1056.83 (544.51) | 300.01 (0.01) | NA | 2226.72 (370.66) |
| Default | 88.17 (66.05) | 54.91 | 813.36 (512.14) | 177.19 (91.28) | 40.94 | 1782.96 (887.23) |
| Search(30) | 151.51 (98.42) | 22.51 | 462.18 (339.23) | 299.08 (9.17) | 0.31 | 1251.49 (332.08) |
| Prune | 105.83 (86.46) | 45.88 | 396.8 (318.97) | 292.94 (26.42) | 2.36 | 1312.04 (343.31) |
| L2Sep(R1) | 144.67 (94.46) | 26.01 | 546.58 (370.45) | 299.34 (6.19) | 0.22 | 1504.9 (354.98) |
| L2Sep(R2) | 138.82 (92.89) | 29.00 | 530.09 (367.16) | 299.49 (5.2) | 0.17 | 1494.7 (346.61) |
| LLM4Sepasel | 60.68 (42.03) | 68.97 | 222.88 (143.29) | 264.44 (58.96) | 11.86 | 942.49 (337.13) |
| DynSep (Ours) | **5.47 (19.09)** | 97.20 | **36.75 (71.49)** | **26.66 (77.49)** | 91.11 | **151.39 (364.6)** |

bility. We further evaluate cross-domain and general-to-specific generalization; see Appendix E.7.

**Experiment 4: Interpretability Analysis.** We visualize our learned separator configurations in Fig. 4, plotting the averaged value of 22 separators' activation status at $T = 5$ separation rounds. Our findings

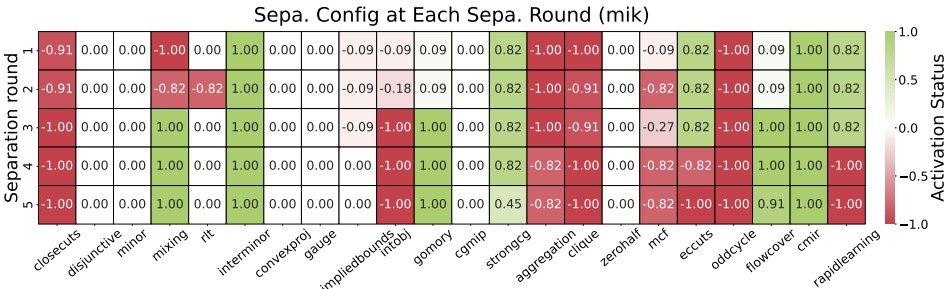

Figure 4: **Separator configs at each separation round for MIK benchmark**. The x-axis lists the 22 separators, while the y-axis shows their average activation status over every node and instance in MIK. Green hue indicates a tendency to apply the separator before the constraint handler, red hue means after the handler, and pale color means the separator is generally disabled.

include: (1) Fig. 4 shows that DynSep consistently activates the CMIR separator (2nd-rightmost col.) in MIK problems, which aligns with the known facts that mixed-integer rounding (MIR) cuts are particularly effective for knapsack-type structures [38]. (2) For easier benchmarks such as MIS and Set Covering, the policy uniformly reduces the maximum number of separation rounds to $r_{max}$ at every node (See Figs. 7 and 8 in Appendix E.8), demonstrating that our learned decision on maximum round effectively prunes unnecessary cutting rounds on simple problems. (3) The heatmap reveals that the separator configuration is not static but varies dynamically across separation rounds (shown along the y-axis), suggesting that the model is timing the application of various separators to coincide with the stage of cut generation. Appendix E.8 provides more results and analyses on other datasets.

## 7 Conclusion

This work proposes a novel **dyn**amic **sep**arator configuration (**DynSep**) method that formulates round-wise separator activation and stopping as an RL task. At each decision step, DynSep processes a tokenized incremental subgraph and uses a decoder-only Transformer to autoregressively predict separator activations and termination. DynSep significantly improves MILP solving efficiency on both synthetic and large-scale real-world benchmarks and generalizes to larger unseen instances. The current implementation of DynSep lacks a lightweight decision model to retain historical information across the global B&B tree. Future research should focus on developing such models to enable incremental decision-making based on the updated graphs.

## 8 Acknowledgements

The authors would like to thank all the anonymous reviewers for their insightful comments. This work was supported in part by the National Key R&D Program of China under contract 2022ZD0119801, National Nature Science Foundations of China grants U23A20388, 62021001 and 624B1011. This work was supported in part by Huawei as well.

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

## Table of Contents for Appendix

# A  Proof of Permutation Equivalence

## A.1  Formulation for Transformer

The standard Transformer architecture comprises two main components: the multi-head self-attention layers (MHA) and the position-wise feed-forward network (FFN). In the following part, we will briefly introduce these blocks. We represent an input sequence as $\mathbf{X} = \langle \mathbf{x}_1, \ldots, \mathbf{x}_N \rangle \in \mathbb{R}^{N \times d}$, where $\mathbf{x}_i$ is the hidden embedding for the token $i$, and $d$ is the dimension of the embeddings. The MHA module projects $\mathbf{X}$ to a triplet $(\mathbf{Q}, \mathbf{K}, \mathbf{V})$, as follows.

$$\mathbf{Q_X} = \mathbf{XW}_Q, \ \mathbf{K_X} = \mathbf{XW}_K, \ \mathbf{V_X} = \mathbf{XW}_V,$$

$$\texttt{Attention}(X) = \texttt{softmax}\left(\frac{\mathbf{Q}_X \mathbf{K}_X^\top}{\sqrt{d_K}}\right) \mathbf{V}_X,$$

where $\mathbf{W}_Q \in \mathbb{R}^{d \times d_K}$, $\mathbf{W}_K \in \mathbb{R}^{d \times d_K}$, $\mathbf{W}_V \in \mathbb{R}^{d \times d_V}$ are learnable weights, with $d_K = d_V = \frac{d}{H}$. Overall, $H$ such projections are performed, resulting in $(\mathbf{Q}_X^{(h)}, \mathbf{K}_X^{(h)}, \mathbf{V}_X^{(h)})$ for $1 \le h \le H$. The self-attention operation is then applied to each triplet:

$$\text{head}_h = \texttt{softmax}\left(\frac{\mathbf{Q}_X^{(h)} \mathbf{K}_X^{(h)\top}}{\sqrt{d_K}}\right) \mathbf{V}_X^{(h)}, \tag{8}$$

$$MHA(\mathbf{H}) = \texttt{concat}(\text{head}_1, \ldots, \text{head}_H)\mathbf{W}_O, \tag{9}$$

where $\mathbf{W}_O \in \mathbb{R}^{d \times d}$ is a learnable weight matrix. The output of the MHA module is then passed through a feed-forward network layer, followed by a residual connection and layer normalization (LN). Finally, the output of the $l$-th layer $\mathbf{H}^l$ is computed as follows:

$$\widehat{\mathbf{H}}^l = \text{LN}(\mathbf{H}^{l-1} + \text{MHA}(\mathbf{H}^{l-1})), \tag{10}$$

$$\mathbf{H}^l = \text{LN}(\widehat{\mathbf{H}}^l + \text{FFN}(\widehat{\mathbf{H}}^l)). \tag{11}$$

To remain consistent with the notation of the common Transformer study, a small subset of definitions in this appendix overlaps with those in the main text. We claim that these symbol definitions are valid only within this chapter.

## A.2  Proof of Proposition 1

We now present the full proof of Proposition 1. Recalling that the blocked positional encoding is defined by

$$P = \begin{bmatrix} P_0^\top & P_1^\top & \cdots & P_T^\top \end{bmatrix}^\top \in \mathbb{R}^{(T+1)(K+1) \times d}, \tag{12}$$

where $P_i = \begin{bmatrix} \text{pos}(i) & \text{pos}(i) & \cdots & \text{pos}(i) \end{bmatrix}^\top \in \mathbb{R}^{(K+1) \times d}$, $\text{pos}(i) \in \mathbb{R}^d$, so that each of the $K + 1$ tokens in block $i$ shares the same vector $\text{pos}(i) \in \mathbb{R}^d$.

**Proposition.** *The decoder-only Transformer $\mathcal{T}$, equipped with the blocked positional encoding $P$, is permutation equivariant inside each token block. Formally, for any input $X$ and any block-wise permutation matrix $\Pi$, it holds that $\mathcal{T}(\Pi X + P) = \Pi \mathcal{T}(X + P)$.*

*Proof.* Because feed-forward, layer-normalization, and residual addition each of them either applies elementwise or row-wise operations, or multiplies on the right by a learnable weight matrix, they automatically commute with any row-permutation of their input. Hence, to prove that the full model is equivariant under block-wise token permutations, it suffices to show that each self-attention module satisfies $\texttt{Attention}(\Pi X + P) = \Pi \cdot \texttt{Attention}(X + P)$.

We define the block-wise permutation matrix $\Pi$ as a permutation matrix that operates independent permutations within each block, which is formulated as follows:

$$\Pi = \begin{bmatrix} \Pi_0 & & & \\ & \Pi_1 & & \\ & & \ddots & \\ & & & \Pi_{t-1} \end{bmatrix},$$

where $\Pi_i \in \{0, 1\}^{(K+1) \times (K+1)}$ is an arbitary permutation matrix, with the condition that each row and each column contains exactly one entry of 1 and the rest are 0. Denote the input by:

$$X = [z_0^\top \quad z_1^\top \quad \cdots \quad z_{t-1}^\top]^\top \in \mathbb{R}^{(K+1)t \times d}, \text{ where } z_i = [h_{i,0} \quad h_{i,1} \quad \cdots \quad h_{i,K}]^\top \in \mathbb{R}^{(K+1) \times d}.$$

Apply $\Pi$ yields $\widetilde{X} = \Pi X = [\Pi_0 z_0^\top \quad \Pi_1 z_1^\top \quad \cdots \quad \Pi_{t-1} z_{t-1}^\top]^\top$. Because each block $P_i$ in our positional encoding consists of identical rows, permitting these rows does no change, i.e., $\Pi_i P_i = P_i$ for any $i$. Hence, we have that

$$\Pi X + P = \Pi X + \Pi P = \Pi(X + P). \tag{13}$$

Next, after performing the blocked positional encoding, we have the attention of $\widetilde{X}$ as follows:

$$\texttt{Attention}(\widetilde{X} + P) = \texttt{softmax}\left(\frac{(\Pi X + P)\mathbf{W}_Q \mathbf{W}_K^\top (\Pi X + P)^\top}{\sqrt{d_K}}\right)(\Pi X + P)\mathbf{W}_V \tag{14}$$

Since `softmax` is applied row-wise, it can be viewed as left-multiplying the input by a matrix. Thus, by the associativity of matrix multiplication, we can freely regroup the products of the matrices in the above formula (14). First, we deduce that

$$
\begin{aligned}
(\Pi X + P)^\top (\Pi X + P) &= \sum_{i=0}^{t} (\Pi_i z_i + P_i)^\top (\Pi_i z_i + P_i) \\
&= \sum_{i=0}^{t} z_i^\top \Pi_i^\top \Pi_i z_i + X_i^\top \Pi_i^\top P_i + P_i^\top \Pi_i X_i + P_i^\top P_i \\
&= \sum_{i=0}^{t} z_i^\top z_i + X_i^\top P_i + P_i^\top X_i + P_i^\top P_i \tag{15} \\
&= \sum_{i=0}^{t} (z_i + P_i)^\top (z_i + P_i) \\
&= (X + P)^\top (X + P), \tag{16}
\end{aligned}
$$

where the deduction of equation (15) utilizes the property that $\Pi^\top \Pi = \mathbf{I}$ and $\Pi_i^\top P_i = P_i \Pi_i^\top = P_i$.

Therefore, substituting equations (16) and (13) into (14), we have

$$\texttt{Attention}(\widetilde{X} + P) = \Pi \cdot \texttt{softmax}\left(\frac{(X + P)\mathbf{W}_Q \mathbf{W}_K^\top (X + P)^\top}{\sqrt{d_K}}\right)(X + P)\mathbf{W}_V \tag{17}$$

$$= \Pi \cdot \texttt{softmax}\left(\frac{\mathbf{Q}_X \mathbf{K}_X^\top}{\sqrt{d_K}}\right)\mathbf{V}_X \tag{18}$$

$$= \Pi \cdot \texttt{Attention}(X + P). \tag{19}$$

Finally, each subsequent layer (feed-forward, layer norm, residual connections) also commutes with block-wise permutation, so the entire model satisfies the block-wise permutation equivariance:

$$\mathcal{T}(\Pi X + P) = \Pi \mathcal{T}(X + P),$$

as required. □

## B   Implementation Details of Separator Configuration in SCIP

### B.1   Frequency and Priority Setup in SCIP

In the SCIP solver, we adjust the activation status configuration by two hyperparameters: the frequency $f_i$ and the priority $q_i$. We provide the detailed description of these two configuration parameters as follows.

`SEPA_FREQ` $f_i$: The frequency parameter determines at which nodes in the branch-and-bound tree a separator is invoked. Specifically, setting the $f_i = -1$ disables the separator entirely while $f_i = 0$ activates the separator in any separation round of any tree node. Any positive $f_i > 0$ activates the

separator at every node whose depth is a multiple of $f_i$. We set $f_i = 10$ for all active separators in our paper.

SEPA_PRIORITY $p_i$: The priority parameter dictates the order in which separators are executed during a separation round at a node. In every separation round, all separators with $p_i \geq 0$ are executed first in descending order of $p_i$, then constraint handlers are applied, and finally separators with $p_i < 0$ run in descending order of $p_i$. By convention, separators implementing fast, high-impact cuts have large non-negative priorities so that their cuts are added early, thus strengthening the LP relaxation sooner. In contrast, more expensive or specialized separators are given negative priorities. Hence, they run later or only if no earlier cuts were found, thereby avoiding unnecessary overhead at the start of each node's separation phase. This division into early ($p_i \geq 0$) versus late ($p_i < 0$) activation phases directly influences the quality of intermediate bounds: running aggressive separators early can dramatically tighten the relaxation and reduce the number of branch-and-bound nodes, while deferring or disabling them can save CPU time when their benefit is marginal at that stage.

**Activation Status** To encapsulate the activation configuration, we define an activation status variable $\eta_i$ for each separator. In detail, $\eta_i = 0$ means that we set the frequency of separator $f_i = 0$ and thus let it never run in all B&B tree nodes. $\eta_i = +1$ means that we set $f_i > 0$, but the priority of the separator $p_i >= 0$, executing it before the constraint handler [44]. $\eta_i = -1$ means that we set $f_i > 0$ and $p_i < 0$, which means the separator is activated but executed after the constraint handler. As shown by the orange bar in Fig. 2(b) of the main text, perturbation in priority configs exhibits a minor effect on performance improvement compared to activation status (yellow bars) changes This phenomenon arises because activation status dictates the order of separators across rounds, whereas priority affects only their relative order within a round; since the LP is not re-solved until the end of a round, reordering separators within the same round has little effect on overall performance.

## B.2 Input Features of the Triplet Graph

**Node Features** Each node type—variables, constraints, and separators—is characterized by a set of features that encapsulate their properties and roles within the optimization process. The input features for variable nodes $V$, constraint nodes $C$, and separator nodes are list in Table 4. Furthermore, we incorporate two graph-level input features–the dual degeneracy rate and the variable-to-constraint ratio. Concretely, the dual degeneracy rate is the fraction of nonbasic variables having zero reduced cost, and the variable-to-constraint ratio is the number of unfixed variables relative to the LP basis size. In practice, we first apply an additive (sum) pooling over the GCN encoder's node representations to obtain a single aggregated node feature vector. We then append (concatenate) the two scalar metrics to this pooled embedding. The combined vector is passed through a multilayer perceptron, yielding a unified graph embedding that fuses the local structural information.

**Edge Features** Like the bipartite graph modeling for common MILP problem, we construct edges between variable and constraint nodes such that a variable node $V_i$ is connected to a constraint node $C_j$ if the variable appears in the constraint with the weight corresponds to the coefficient $A_{ij} \neq 0$, and we set the value of the edge as $A_{ij}$.

## B.3 Separator Statistics

In SCIP's solver-statistics display, each separator reports several key metrics that provide insights into its performance and impact during the solving process. We use these metrics as input features of separator nodes and immediate reward signals at each time step. These metrics include:

**Cut Application Rate.** This rate measures the effectiveness of a separator's generated cuts by calculating the ratio of cuts applied to the LP relaxation to the total number of cuts found. The application rate is computed as:

$$\text{Cut Application Rate} = \frac{\text{Number of Cuts Applied}}{\text{Number of Cuts Found}} \tag{20}$$

A higher application rate suggests that the separator frequently generates cuts deemed valuable and effective by SCIP's internal filtering mechanisms, leading to their inclusion in the LP relaxation. Conversely, a lower rate may indicate that many of the separator's cuts are redundant or less impactful.

**Domain Reductions (DomReds)** Separators can also perform domain reductions by tightening variable bounds through their logic (for example, by deducing $x_i \leq u$ or $x_j \geq l$ from cut coefficients).

Table 4: Description of input features for variable, constraint, separator nodes, and the entire graph.

| Type | Feature | Setting |
|------|---------|---------|
| Vars | type | Variable's type: binary, integer, implicit integer, continuous (one-hot). |
| | has_lb | If the variable has an infinite lower bound. |
| | lb | The variable's lower bound, set 0 if it is infinite. |
| | has_ub | If the variable has an infinite upper bound. |
| | ub | The variable's upper bound, set 0 if it is infinite. |
| | basestat | Simplex basis status: lower, basic, upper, zero (one-hot) |
| | coef_norm | Objective coefficient, normalized by objective norm |
| | redcost_norm | Variable's reduced cost divided by the objective norm, indicating how much the objective would worsen per unit increase at zero slack |
| | age | The number of consecutive LP iterations in which the variable stayed at zero in the basis. |
| | solval | The primal LP solution value of the variable. |
| | solfrac | The fractional part of 'solval'. |
| | sol_is_at_lb | If 'solval' equals the lower bound within numerical tolerance |
| | sol_is_at_ub | If solval equals the upper bound within numerical tolerance |
| | round_num | Index of the current separation round |
| Cons | origin_type | Which mechanism generated this row: unspecified, constraint handler, constraint, separator, reoptimization (one-hot). |
| | origin_sepa | The separator name that produced this row. |
| | basestat | The row's basis status in the LP solution: basic, lower, upper (one-hot). |
| | bias | Unshifted side normalized by row norm. |
| | dualsol | Dual LP solution of a row, normalized by row and objective norm. |
| | is_at_lhs | If the row value equals the left-hand side. |
| | is_at_rhs | If the row value equals the right-hand side. |
| | norm_nnzr | Number of nonzero coefficients in the row, normalized by the total number of LP variables. |
| | age | The count of successive LP iterations for which the row has stayed nonbasic at zero. |
| | int_support | Integral support score of a row. |
| | is_integral | Activity of the row is always integral in a feasible solution. |
| | is_removable | Row is removable from the LP. |
| | is_in_lp | Row is member of current LP. |
| | round_num | Index of the current separation round |
| Sepas | type | Type of the separator (one-hot). |
| | time | Execution time consumed by the separator. |
| | calls | Number of times that the separator has been invoked. |
| | cuts | Number of cuts generated by this separator. |
| | cutoffs | Number of infeasibility detections (cutoffs) found by the separator. |
| | domreds | Number of domain reductions found by the separator. |
| | applied | Number of cuts from the separator that were applied to the LP relaxation. |
| Graph | dual_deg_rate | The proportion of nonbasic variables with reduced cost zero. |
| | var_con_ratio | The ratio of unfixed variables to the size of the LP basis. |

Each time a separator callback successfully reduces a variable's domain, the DomReds counter is incremented. This statistic captures the total number of such bound-tightening operations executed by that separator during the solve. A higher DomReds count signifies that the separator contributes significantly to shrinking feasible regions, which can indirectly improve future LP relaxations and cut generation efficiency.

**Cutoffs.** Within each separation round, when a separator generates one or more cuts that render the LP infeasible or whose bound surpasses the current best primal solution, SCIP immediately prunes that node (i.e., cuts it off) without further processing. The Cutoffs statistic for a separator is simply the total count of these pruned nodes attributable to its cuts. In practice, a high Cutoffs value indicates

that the separator is highly effective at early fathoming of unpromising subproblems, potentially reducing the size of the branch-and-bound tree.

## B.4  Stopping Conditions for a Separation Loop

In SCIP, the separation process at a node is conducted in iterative rounds, where each round involves generating cutting planes to refine the LP relaxation. The separation loop at a node terminates when any of the following conditions are met:

**Maximum Number of Rounds Reached:** A user-defined limit on the number of separation rounds per node is enforced to prevent excessive computation. For experimental stability, we use only the maxround $m_t$ decision from the first separation round at each node to set the maximum number of subsequent separation rounds at the current node.

**Stalling Criterion Triggered:** If consecutive separation rounds fail to yield improvements in the objective bound or integrality, the process is considered to have stalled, prompting termination.

**Relative Bound Distance Exceeded:** Separation is halted if the relative distance between the current node's dual bound and the global primal bound surpasses a predefined threshold, indicating diminishing returns from further separation.

**No Further Separation Requested:** If all separators and constraint handlers indicate that no additional separation is necessary (i.e., none return a status requesting another round), the loop concludes.

These stopping criteria ensure a balance between the thoroughness of the separation process and computational efficiency, preventing unnecessary iterations that offer minimal benefit to the overall solution process.

## B.5  Separators Built In SCIP

We consider 22 separators in our configuration task. We provide the detailed description of these separators as follow.

**closecuts.** Close cuts are a type of cutting plane that focuses on generating cuts that are "close" to the current fractional solution. These cuts are designed to tighten the feasible region by targeting solutions that are near the boundary of the current relaxation. The idea is to improve the quality of the LP relaxation by adding cuts that are particularly relevant to the current solution.

**disjunctive.** Disjunctive cuts are a class of cutting planes used in mixed-integer programming, particularly based on the concept of disjunctions. These cuts are derived from a disjunctive argument that partitions the solution space into different disjunctive sets. By analyzing the infeasible or fractional solutions that arise from linear relaxations, disjunctive cuts can tighten the formulation by excluding these solutions and enforcing integrality conditions more strongly.

**minor.** Derived from graph minor theory, these cuts identify isomorphic substructures in the constraint matrix corresponding to known hard combinatorial subproblems. By recognizing these patterns, the separator generates cuts that exploit the inherent complexity of the substructures.

**mixing.** Generates cuts by combining multiple weak constraints through coefficient mixing, creating stronger aggregated inequalities. The method systematically blends constraints sharing common variable structures while preserving problem feasibility.

**rlt.** Reformulation-Linearization Technique (RLT) converts polynomial constraints into linear inequalities through variable substitution and constraint multiplication. RLT preserves problem structure while creating convex envelopes for nonlinear terms, enabling strong linear relaxations.

**interminor.** This separator extends minor cuts by focusing on medium-scale substructures that appear as intermediate components in larger mixed-integer programming (MIP) models. Balances local pattern matching with global problem structure analysis.

**convexproj.** Convex projection cuts are generated by projecting an infeasible point onto a convex relaxation of the problem and then creating gradient-based cuts at the projection point. These cuts are designed to improve the separation of fractional solutions in convex nonlinear programs. The method

aims to enhance the solver's ability to make progress by refining the feasible region through gradient information at the projected point.

**gauge.** Geometric cuts based on analyzing the gauge function of the feasible region's convex hull. This separator generates deep cuts orthogonal to the objective gradient by exploiting polyhedral geometry.

**impliedbounds.** Implied bound cuts are cutting planes that leverage implications between binary and continuous variables to restrict the feasible region of an MILP problem. They enforce tighter constraints by exploiting logical relationships, such as when a binary variable limits a continuous variable's upper or lower bound.

**intobj.** Integer objective cuts are cutting planes used in MIP when the objective function is integer-valued. These cuts aim to eliminate fractional solutions from the LP relaxation that are not feasible in the integer solution space. They help tighten the LP relaxation by leveraging the fact that certain fractional values of the objective function cannot lead to valid integer solutions.

**gomory.** Gomory cutting planes are derived from the fractional solutions of the LP relaxation of an MIP problem. Once the LP relaxation is solved, any variable with fractional values can be targeted, and a Gomory cut is generated to eliminate these fractional solutions, moving the solution closer to integrality. These cuts can be generated iteratively during the branch-and-bound process to progressively tighten the LP relaxation.

**cgmip.** Chvatal-Gomory cuts are generated by forming non-negative integer combinations of the original linear constraints and then rounding the resulting coefficients to produce a valid inequality. These cuts are designed to tighten the LP relaxation by eliminating fractional solutions. The process involves solving a sub-MIP to identify the best combination of constraints, which ensures that the generated cuts are as effective as possible.

**strongcg.** Strong Chvátal-Gomory (CG) cutting planes are an extension of the classical CG cuts, which are derived from valid inequalities of the linear relaxation of an integer programming problem. These cuts are used to iteratively tighten the LP relaxation by adding inequalities that exclude fractional solutions. The strong variant refers to CG cuts that are particularly effective in reducing the feasible region, leading to a faster convergence to the integer solution.

**aggregation.** This separator generates cuts by aggregating multiple constraints into single strengthened inequalities. It specializes in flow cover inequalities for network problems, combining arc selection variables with flow conservation constraints.

**clique.** Clique cutting planes are a type of valid inequality derived from the set-packing formulation in integer programming, particularly useful in problems involving binary variables. The inequalities are based on identifying cliques in a conflict graph representation of the problem. A clique is a subset of mutually adjacent vertices in a graph, representing a set of constraints that cannot be simultaneously satisfied. The corresponding clique cutting planes exclude infeasible solutions by enforcing that only one element from each clique can be selected.

**zerohalf.** Zero-half cuts are a specific type of Chvátal-Gomory (CG) cuts in integer programming. These cuts are derived using coefficients in $\{0, \frac{1}{2}\}$ instead of integer coefficients. They are used to tighten the relaxation of integer programming problems, bringing it closer to the convex hull of feasible integer solutions.

**mcf.** Flow path cuts are valid inequalities used to strengthen the linear relaxation of MIP problems, specifically for problems involving fixed charge networks. They help in modeling flow through a sequence of nodes where fixed charges are incurred if any flow occurs along a path. Flow path inequalities operate on constraints related to fixed charge paths, where binary and continuous variables govern the flow through a network. These cuts are particularly useful in fixed charge network design and lot-sizing problems. However, the computational consideration is that the structure exploited by these cuts is very specific, meaning they are only applicable to certain problem types.

**eccuts.** This separator specializes in cuts for edge-concave functions in Mixed-Integer Nonlinear Programming(MINLP), generated by constructing supporting hyperplanes at concave function edges. It exploits piecewise-linear approximations of nonlinear constraints.

**oddcycle.** Odd cycle cuts are designed to eliminate infeasible fractional assignments in problems where binary variables represent nodes or edges in a graph. They are particularly effective when

dealing with cycles in a graph that contain an odd number of nodes. For example, in a graph-based problem, assigning fractional values to all variables in an odd cycle is infeasible when the solution must be binary (0 or 1). Odd cycle cuts ensure that such fractional solutions are excluded from the feasible region.

**flowcover.** Flow cover cuts are a type of cutting plane derived from valid inequalities used to tighten the linear relaxations of MIP problems, particularly for binary single-node flow sets. They are useful in network design and fixed charge problems, where variables can represent flows subject to upper bounds and binary decisions.

**cmir.** MIR cuts are a class of cutting planes derived from mixed-integer sets, particularly when dealing with constraints that include both continuous and integer variables. The main idea behind MIR cuts is to generate valid inequalities by rounding coefficients of mixed-integer constraints to tighten the LP relaxation. They are generated using a disjunctive argument, which creates inequalities that separate fractional solutions from the feasible region of the MIP.

**rapidlearning.** Rapid learning is a heuristic technique that temporarily relaxes certain constraints or simplifies the problem to solve a more manageable version. By solving this easier problem, the solver gains insights into the structure of the original problem. The rapid learning separator then uses this information to generate useful cuts or constraints that can immediately improve the quality of the LP relaxation in the original problem.

## C  Network Details

### C.1  The Encoder Network

We provide a detailed description of the neural architecture employed in our encoder network. Our design builds upon the framework introduced in L2Sep [6], incorporating several modifications to enhance performance. The encoder first embeds maps the input features of constraint ($C$), variable ($V$), and separator ($S$) nodes into hidden representations. Subsequently, it performs message passing following the directions of $V \rightarrow C \rightarrow V$, $S \rightarrow V \rightarrow S$, and $S \rightarrow C \rightarrow S$, effectively capturing the interactions among different node types. Then, the $S$ nodes pass through an attention module to emphasize the task of the separator configuration. In contrary to the approach in [45], which outputs a score for each cut node, our encoder applies a global additive pooling on each of the $C$, $V$, and $S$ hidden embeddings, yielding three aggregated embedding vectors. These vectors are concatenated with two graph-level features, as detailed in Section B.2, forming a comprehensive representation. Finally, this combined vector is passed through a multilayer perceptron (MLP) to produce a unified graph embedding that encapsulates both local and global information pertinent to the problem structure.

## D  Implementation Details of our algorithm

### D.1  Algorithm Pseudocode of DynSep

We employ the Proximal Policy Optimization (PPO) algorithm to train our model. PPO alternates between collecting data through interactions with the environment and optimizing a surrogate objective function to update the policy. To ensure stable training, PPO utilizes a clipped surrogate objective that constrains the policy updates, preventing drastic changes that could destabilize learning. Specifically, the policy network is updated to maximize the expected advantage while maintaining the probability ratio between the new and old policies within a predefined threshold. Concurrently, the value network is trained to minimize the mean squared error between the predicted value estimates and the actual returns. The training procedure of our method DynSep, including the formulation of PPO objective functions, is detailed in the pseudocode of Algorithm 1.

### D.2  Hyperparameters

We train DynSep using the ADAM optimizer [46] within the PyTorch framework [47]. Consistent with prior studies [33, 42], we split each dataset into the train and test sets with 80% and 20% instances. We train our model on the train set for 100 epochs, and select the best model on the train set to evaluate on the test set. A complete list of hyperparameters for the SCIP solver, the PPO algorithm, and the decoder-only Transformer appears in Table 5.

**Algorithm 1** Dynamic Separator Configuration via PPO (DynSep)

---

1: Denote parameters of the actor's transformer $\mathcal{T}_\pi$ and policy $\pi$ as $\theta$.
   Denote parameters of the critic's transformer $\mathcal{T}_V$ and value function $V$ as $\psi$.
   Denote parameters of the encoder $\Phi$ as $\beta$.
2: **Initialize** MILP instances set $\mathcal{X}$, replay buffer $\mathcal{D}$, sampling size $N_s$, training epochs $N_e$, clipping factors $\varepsilon$, learning rates $\alpha_\pi, \alpha_V$ and model parameters $(\theta, \psi, \beta)$.
3: **for** $N_e$ epochs **do**
4:     Clear the replay buffer $\mathcal{D}$.
5:     *// Data collection*
6:     **for** $N_s$ sampling steps **do**
7:         Randomly sample an instance $x$ form $\mathcal{X}$.
8:         Run the MILP solver to optimize instance $x$ with configuration policy $\pi$, collecting $N_x$ episodes of data $\left\{ \left\{ \left( s_t^{(i)}, a_t^{(i)}, r_t^{(i)} \right) \right\}_{t=0}^{T(N_x)} \right\}_{i=1}^{N_x}$ from $T(N_x)$ separation rounds at $N_x$ nodes.
9:         Append collected episodes to $\mathcal{D}$.
10:    **end for**
11:    *// Model Optimization via PPO*
12:    Compute returns $\hat{R}_1, \ldots, \hat{R}_T$ and advantage estimates $\hat{A}_1, \ldots, \hat{A}_T$ for each episode in $\mathcal{D}$.
13:    **for** each minibatch $\mathcal{D}_b \subset \mathcal{D}$ **do**
14:        Compute ratio $r_t(\theta) \leftarrow \mathbb{E}_{\mathcal{D}_b} \left\{ \frac{\pi(a_t|\Phi(s_t))}{\pi_{\text{old}}(a_t|\Phi(s_t))} \right\}$.
15:        Compute actor loss $L_{\text{actor}}(\theta) \leftarrow \mathbb{E}_{\mathcal{D}_b} \left\{ \min(r_t(\theta) \cdot \hat{A}_t, \text{clip}\left( r_t(\theta), 1 - \varepsilon, 1 + \varepsilon \right) \cdot \hat{A}_t \right\}$.
16:        Update $\theta \leftarrow \theta + \alpha_\pi \nabla_\theta L_{\text{actor}}(\theta)$.
17:        Compute critic loss $L_{\text{critic}}(\psi, \beta) \leftarrow \mathbb{E}_{\mathcal{D}_b} \left\{ \left( V(\Phi(s_t)) - \hat{R}_t \right)^2 \right\}$.
18:        Update $(\psi, \beta) \leftarrow (\psi, \beta) + \alpha_\beta \nabla_{(\psi, \beta)} L_{\text{critic}}(\psi, \beta)$.
19:    **end for**
20: **end for**

---

Table 5: Hyperparameters used in DynSep.

| Type | Parameter | Value |
|------|-----------|-------|
| SCIP Solver | Number of separators $K$ | 22 |
| | Frequency of each activated separators $f_i$ | 10 |
| | Upper limit of separation rounds $T$ | 5 |
| | time limit per instance | 300 |
| PPO in Alg. 1 | Training epoch $N_e$ | 100 |
| | Sampling size $N_s$ per epoch | 16 |
| | Minibatch size $|\mathcal{D}_b|$ | |
| |     MIPLIB mixed neos | 8 |
| |     MIPLIB mixed supportcase | 6 |
| |     Other benchmarks | 16 |
| | Clipping factor $\varepsilon$ | 0.2 |
| | Optimizer | Adam |
| | learning rates $\alpha_\pi, \alpha_V$ | 0.0001 |
| | learning rate decay for every $k_{\text{lr}}$ step | 5 |
| | learning rate decay rate $\alpha_{\text{lr}}$ | 0.96 |
| Transformer | Embedding dimension of attention $d$ | 64 |
| | Number of attention heads | 4 |
| | Number of attention layers | 4 |
| | Attention dropout | 0.0 |
| | Activation of FFN | GeLU |
| | Embedding dimension of FFN $d_{\text{FFN}}$ | 256 |
| | Number of FFN layers | 2 |
| | Layer number of the linear actor head | 1 |
| | Layer number of the linear critic head | 1 |

### D.3 Hardware Specification

Training and evaluation on the easy and medium datasets were performed on a single machine equipped with eight GPUs (NVIDIA GeForce RTX 2080 Ti) and two Intel E5-2667 v4 CPUs (32 logical cores), while experiments on the hard datasets used a single machine equipped with eight GPUs (NVIDIA GeForce GTX 3090 Ti) and two Intel Gold 6246R CPUs (64 logical cores) for hard datasets.

### D.4 Baselines

We provide additional implementation details for our baseline methods:

**Search($\rho$):** This method randomly samples $\rho$ configurations then applies one with the best performance on the validation set. The validation set is a subset of the training data, with a size equal to that of the corresponding test set for each MILP benchmark.

**Prune:** This method deactivates separators with no contribution during the evaluation on the validation set. Specifically, if a separator's statistics—namely, Cut Application Rate, DomReds, and Cutoffs (as defined in Section B.3)—are all zero, the separator is deactivated. The validation set used here is partitioned identically to that in the Search($\rho$) method.

**L2Sep [6]:** We configure the SCIP solver parameters in alignment with our default settings, except for the learned separator activation statuses. Specifically, we impose a 300-second time limit, set the frequency $f_i = 10$ for all activated separators, and enable both presolve and heuristic mechanisms. The validation set for L2Sep is partitioned identically to that used in the Search($\rho$) method. We define the size of the restricted configuration space as 15 for easy datasets, 20 for medium datasets, and 25 for hard datasets.

**LLM4Sep [7]:** The LLM4Sep baseline utilizes the DeepSeek Chat API to generate separator configurations. The context provided to the language model includes detailed descriptions of each separator, as outlined in Section B.5, along with information regarding the problem structures the separators operate on and their computational characteristics. Additionally, the model receives a comprehensive description of the MILP problem, encompassing the general formulation and the summary text of MILP objectives and constraints.

## E   Additional Results

### E.1   Motivation Results on effects of different configurations

Fig. 2(b) in our main text shows motivation results for five benchmarks, with alternative names of D1–Set Covering, D2–Max Independent Set (MIS), D3–MIK, D4–Load Balancing, D5–MIPLIB mixed neos. Here we provide more results for all nine benchmarks.

The left panel of Fig. 5 shows the best-performing configuration per instance identified by four randomization strategies applied to separator configurations. It extends the analysis from Fig. 2(b) in the main text by presenting results across nine benchmark datasets: D1–Set Covering, D2–Max Independent Set (MIS), D3–Multiple Knapsack, D4–CORLAT, D5–MIK, D6–Anonymous, D7–Load Balancing, D8–MIPLIB Mixed Neos, and D9–MIPLIB Mixed Supportcase. For each instance within these datasets, we evaluated 50 random configurations.

The right panel depicts the average performance across instances for each dataset. Our observations reveal substantial performance variance across all four strategies, underscoring the significant impact of the specific separator parameters and dynamic configurations on solver efficiency. Notably, the Priority strategy exhibits comparatively lower variance in performance. This is attributed to the fact that priority adjustments influence only the relative ordering of separators within a separation round; since the LP relaxation is not re-solved until the round concludes, such reordering has minimal effect on overall solver performance.

Furthermore, Fig. 6 illustrates the impact of varying the maximum number of separation rounds per node, denoted as $r_{max}$, on solver performance across nine benchmark datasets. Each plot shows the

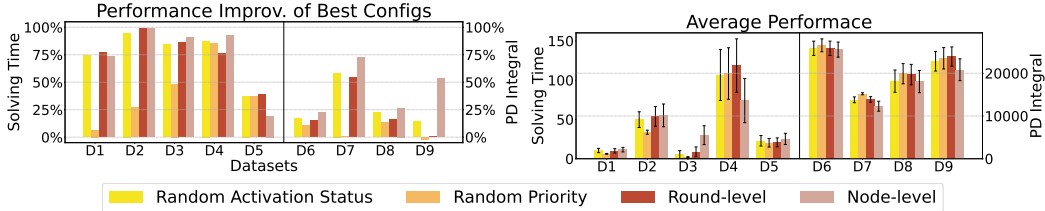

Figure 5: **Left:** Performance improvement of the best configurations found by different random strategies on nine benchmarks. The y-axis represents the relative improvement compared to the default setting. **Right:** Average performance of configurations sampled by different random strategies on nine benchmarks. The y-axis represents the real performance under two metrics. Specifically, Datasets D1–D5 use solving time (left) as the metric, while D6–D9 use PD integral (right). Each bar represents a specific strategy to get random configurations.

average solving time (red line, left y-axis) and PD integral (blue line, right y-axis) across all instances in the dataset. The two metrics exhibit highly consistent trends in each benchmark, indicating a strong correlation between solving time and PD integral. The results also show that changes in $r_{max}$ lead to significant performance variability; however, increasing $r_{max}$ does not universally enhance performance, and the optimal value of $r_{max}$ varies among datasets. Prior work [48] also observes that solver performance is sensitive to the maximum number of cut rounds and learns a data-driven stopping policy; however, it does not model per-round separator configuration, whereas we jointly decide when to halt and which separators to activate each round.

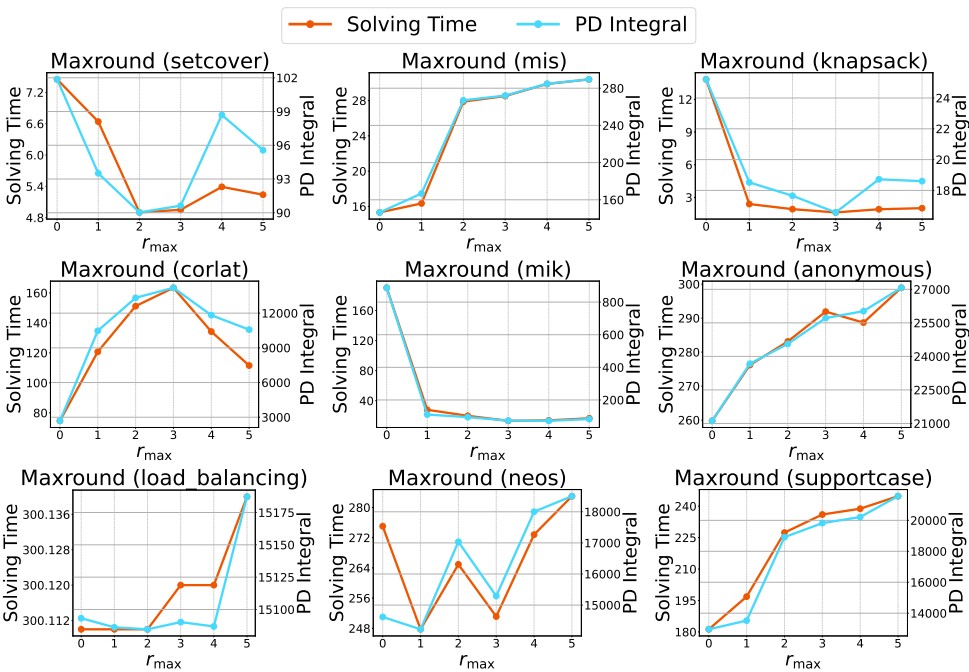

Figure 6: Effect of varying maximum round $r_{max}$ on solver performance for nine benchmarks. Each plot shows the average solving time (red line, left y-axis) and PD integral (blue line, right y-axis) across all instances in the dataset.

## E.2 Evaluation Results on Other Metrics

We provide evaluation results of nine benchmarks for two other metrics of the primal-dual gap (PD gap) and total number of nodes (Nnodes) in Table 6. The PD gap reflects the solution quality achieved by the solver, while Nnodes indicates the size of the B&B search tree—an indirect measure of solving

Table 6: **Evaluation results on nine benchmarks about two other metrics of primal-dual gap (PD gap) and total number of nodes (Nnodes).** Best performance is in bold. The values report the mean (standard deviation) of time and PD integral metrics.

| | Easy: Set Covering | | Easy: Max Independent Set | | Easy: Multiple Knapsack | |
|---|---|---|---|---|---|---|
| Method | PD gap ↓ | Nnodes ↓ | PD gap ↓ | Nnodes ↓ | PD gap ↓ | Nnodes ↓ |
| NoCuts | 0.0 (0.0) | 114.84 (413.86) | 0.0 (0.0) | 529.31 (1703.82) | 0.0 (0.0) | 17847.64 (42453.94) |
| Default | 0.0 (0.0) | 1.11 (1.06) | 0.0 (0.0) | 1.0 (0.0) | 0.0 (0.0) | 36.91 (85.56) |
| Search(50) | 0.0 (0.0) | 1.16 (1.59) | 0.0 (0.0) | 123.41 (347.52) | 0.0 (0.0) | 3164.31 (4942.28) |
| Prune | 0.0 (0.0) | 207.61 (439.61) | 0.0 (0.0) | 403.72 (810.19) | 0.0 (0.0) | 12.19 (34.42) |
| L2Sep(R1) | 0.0 (0.0) | 211.27 (402.0) | 0.0 (0.0) | 386.28 (790.7) | 0.0 (0.0) | 10495.97 (19385.29) |
| L2Sep(R2) | 0.0 (0.0) | 211.81 (401.94) | 0.0 (0.0) | 384.32 (787.77) | 0.0 (0.0) | 12144.57 (19489.3) |
| LLM4Sepasel | 0.0 (0.0) | 179.04 (412.59) | 0.0 (0.0) | 39.12 (64.62) | 0.0 (0.0) | 1883.54 (2884.54) |
| DynSep (Ours) | **0.0 (0.0)** | **1.0 (0.0)** | **0.0 (0.0)** | **1.0 (0.0)** | **0.0 (0.0)** | **5.2 (13.59)** |

| | Medium: CORLAT | | Medium: MIK | | Hard: Anonymous | |
|---|---|---|---|---|---|---|
| Method | PD gap ↓ | Nnodes ↓ | PD gap ↓ | Nnodes ↓ | PD gap ↓ | Nnodes ↓ |
| NoCuts | 2.67e+18 (1.26e+19) | 57516.67 (96475.09) | 0.02 (0.03) | 176742.17 (126999.33) | 1.83e+19 (3.86e+19) | 13034.68 (11248.92) |
| Default | 2.73e+19 (4.46e+19) | **304.77 (456.1)** | 0.0 (0.0) | 5504.77 (6375.84) | 5.5e+19 (4.96e+19) | **1894.13 (4139.96)** |
| Search(50) | 4e+18 (1.96e+19) | 35564.78 (68651.06) | 0.0 (0.0) | 13134.6 (11543.03) | 4e+19 (4.90e+19) | 4602.75 (6696.04) |
| Prune | 2e+18 (1.4e+19) | 99579.13 (141189.24) | 0.09 (0.02) | 532615.47 (169443.58) | 6.67e+18 (2.45e+19) | 22442.38 (14908.91) |
| L2Sep(R1) | 4e+18 (1.98e+19) | 84423.58 (117122.52) | 0.0 (0.0) | 5157.9 (6471.25) | **1.88 (2.19)** | 15856.25 (14404.64) |
| L2Sep(R2) | 2e+18 (1.41e+19) | 81853.7 (116670.12) | 0.0 (0.0) | 3616.2 (3105.22) | 5e+18 (2.24e+19) | 16151.85 (14903.3) |
| LLM4Sepasel | 2e+18 (1.41e+19) | 38808.98 (71956.13) | 0.0 (0.0) | 5653.0 (6682.38) | 6e+19 (5.03e+19) | 3835.15 (7788.15) |
| DynSep (Ours) | **0.0 (0.0)** | 4084.16 (11015.44) | **0.0 (0.0)** | **3076.0 (2666.61)** | 1.99 (2.96) | 15011.2 (8355.63) |

| | Hard: Load Balancing | | Hard: MIPLIB mixed neos | | Hard: MIPLIB mixed supportcase | |
|---|---|---|---|---|---|---|
| Method | PD gap ↓ | Nnodes ↓ | PD gap ↓ | Nnodes ↓ | PD gap ↓ | Nnodes ↓ |
| NoCuts | 0.97 (0.12) | 1.0 (0.0) | 2.5e+19 (4.33e+19) | 148834.67 (93798.58) | 10.94 (25.26) | 2204.17 (3130.9) |
| Default | 0.97 (0.12) | **1.0 (0.0)** | 2.5e+19 (4.33e+19) | **12927.5 (16777.1)** | 2.5e+19 (4.33e+19) | **22.25 (54.64)** |
| Search(50) | 0.09 (0.01) | 10.24 (12.67) | 2.5e+19 (4.33e+19) | 35011.0 (44832.71) | **0.1 (0.26)** | 482.12 (729.38) |
| Prune | 0.49 (0.05) | 150.51 (65.02) | 2.5e+19 (4.33e+19) | 202703.25 (149382.93) | 12.51 (26.94) | 6162.54 (13362.48) |
| L2Sep(R1) | 0.59 (0.37) | 86.36 (55.62) | 2.5e+19 (5e+19) | 66880.0 (103828.32) | 12.45 (23.27) | 1622.38 (3290.69) |
| L2Sep(R2) | 0.59 (0.37) | 85.36 (55.23) | 2.5e+19 (5e+19) | 120425.0 (119640.78) | 9.29 (17.1) | 3414.75 (6492.98) |
| LLM4Sepasel | 0.12 (0.03) | 20.73 (16.85) | 2.5e+19 (5e+19) | 102753.75 (75635.08) | 2.5e+19 (4.63e+19) | 37.38 (102.48) |
| DynSep (Ours) | **0.09 (0.02)** | 9.99 (11.41) | **2.5e+19 (4.33e+19)** | 119616.25 (105217.02) | 7.86 (20.31) | 69.38 (131.34) |

effort, though a smaller tree does not necessarily imply faster solving. The results show that DynSep consistently solves all easy and medium instances to optimality, achieving an average PD gap of zero. Notably, DynSep is the only configuration method that solves all instances to optimality with zero primal-dual gaps, highlighting the effectiveness of DynSep for fine-grained separator configurations.

### E.3 Evaluation on Additional MIPLIB Datasets

We have extended our evaluation beyond MIPLIB mixed neos and mixed supportcase, including two real-world datasets from the Distributional MIPLIB benchmark [49]:

- Maritime Inventory Routing Problem (MIRP). MIRP arises in bulk shipping logistics, integrating vessel routing and port inventory decisions under capacity and inventory constraints. Typical instances of MIRP feature on an average of 15080 binary variables, 19576 continuous variables, and 44430 constraints.
- Seismic-Resilient Pipe Network Planning (SRPN). SRPN involves optimizing municipal water pipe network design to ensure resilience under seismic disturbances, targeting service continuity to critical facilities (e.g. hospitals) while minimizing upgrade or restoration costs within budget. Typical instances of SRPN feature on an average of 3016 binary variables, 3016 continuous variables, and 5917 constraints.

The results are summarized in Table 7. For each dataset, we report solving time (Time), primal–dual gap integral (PD integral), and primal–dual gap (PD gap). All three metrics are lower-is-better, where Time and PD integral reflect solver efficiency and convergence speed, and PD gap quantifies how close the solver comes to the optimal. We set the time limit as 600 seconds for each instance. These results show that DynSep delivers marked performance gains on additional real-world datasets (MIRP and SRPN). Compared to the other configuration methods, DynSep significantly improves

Table 7: Evaluation results on MIPLIB MIRP & SRPN Benchmarks, with 600-second time limit.

| | Hard: MIPLIB MIRP ($n = 34656$, $m = 44430$) | | | Hard: MIPLIB SRPN ($n = 6032$, $m = 5917$) | | |
|---|---|---|---|---|---|---|
| Method | Time(s) ↓ | PD integral ↓ | PD gap ↓ | Time(s) ↓ | PD integral ↓ | PD gap ↓ |
| NoCuts | 486.68 (198.71) | 34735.55 (19014.81) | 6.67e+18 (1.92e+19) | **280.7 (277.17)** | 11020.69 (13133.86) | 0.28 (0.42) |
| Default | 580.98 (61.65) | 52362.46 (13476.17) | 5.38e+19 (4.97e+19) | 332.04 (271.0) | 11687.01 (11993.95) | 0.21 (0.31) |
| Search(20) | 492.85 (190.1) | 33028.81 (16299.41) | 6.30e+18 (2.11e+19) | 384.02 (273.02) | 14571.56 (12206.64) | 0.26 (0.28) |
| Prune | 501.49 (174.33) | 35542.82 (18572.11) | 8.33e+18 (2.19e+19) | 296.08 (277.77) | 9211.76 (10664.7) | 0.19 (0.26) |
| LLM4Sepasel | 511.97 (161.03) | 34573.52 (18681.26) | 5e+18 (2.24e+19) | 300.66 (284.75) | 8421.65 (10209.65) | 0.14 (0.22) |
| DynSep (Ours) | **482.13 (205.78)** | **30838.39 (17377.29)** | **1.38 (1.41)** | 294.77 (274.22) | **7581.16 (8814.39)** | **0.1 (0.17)** |

Table 8: Comparison between default setting and our method (DynSep) on all 235 MIPLIB 2017 instances.

| | Hard: MIPLIB 2017 | |
|---|---|---|
| Method | Time(s) ↓ | PD integral ↓ |
| Default | 258.77 (93.22) | 17153.69 (12674.03) |
| DynSep (Ours) | **238.88 (107.33)** | **15092.16 (12719.67)** |

both convergence speed (as demonstrated by reduced PD integral) and solution quality (evidenced by lower PD gap).

Furthermore, we have tested our method on the full set of 240 MIPLIB 2017 benchmark instances. The results in Table 8 show that DynSep delivers notable improvements in solving efficiency in the challenging MIPLIB 2017 dataset.

Specifically, we set the time limit as 300 seconds and excluded five instances whose presolving time exceeded 300 seconds: *neos-3402454-bohle*, *neos-4722843-widden*, *mzzv42z*, *neos-5052403-cygnet*, *proteindesign121hz512p9*, and *proteindesign122trx11p8*, which is a common removing criterion for MIPLIB2017 benchmark [18, 42]. The remaining 235 instances were split into a 70% training set and a 30% test set. Table 8 reports the overall average performance of our method across all 235 instances. These experiments confirm that our approach delivers notable improvements in solving efficiency, even when evaluated on the more challenging benchmark set.

### E.4 Extended DynSep to Broader Solver Hyperparameters

Our proposed DynSep framework is inherently extensible to a broader set of solver hyperparameters beyond separator activation and timing. This is achieved by adapting the policy network: we model the outputs of additional parameters as a logistic-normal distribution, followed by optional discretization to support both integer and continuous parameters. This enables flexible, differentiable control of arbitrary solver parameters.

To substantiate the above claim, we conducted additional experiments on three critical hyperparameter groups as follows, while retaining the original tuning mechanism for separator activation status $(+1, 0, -1)$ and round termination ($m_t$).

- **para group 1: Cut Depth / Aggressiveness (sepastore/age in SCIP).** Controlled by solver parameters `separating/cutagelimit` and `separating/poolfreq`.
- **para group 2: Cut selection thresholds (e.g., efficacy vs. orthogonality).** Controlled by solver parameters `separating/minefficacy` and `cutselection/hybrid/minortho`.
- **para group 3: Separation frequency per node.** Controlled by solver parameters `separating/cutagelimit` and `separating/poolfreq`.

We provide the experimental results in Table 9. We evaluated our method on three benchmark datasets: Multiple Knapsack, Corlat, and MIPLIB mixed supportcase, measuring solving time (Time), primal–dual gap integral (PD integral), and primal–dual gap (PD gap). All three metrics are lower-is-better, where Time and PD integral reflect solver efficiency and convergence speed, and PD gap quantifies how close the solver comes to the optimal. We compared our original DynSep method with its extended versions incorporating three additional parameter groups. Results show

Table 9: Experiments on three extended hyperparameter groups configured by DynSep.

| Method | Easy: Multiple Knapsack ($n$ = 720, $m$ = 72) | | | Medium: Corlat ($n$ = 466, $m$ = 486) | | | Hard: MIPLIB mixed supportcase ($n$ = 19766, $m$ = 19910) | | |
| | Time(s) ↓ | PD integral ↓ | PD gap ↓ | Time(s) ↓ | PD integral ↓ | PD gap ↓ | Time(s) ↓ | PD integral ↓ | PD gap ↓ |
|---|---|---|---|---|---|---|---|---|---|
| DynSep (Ours) | 0.52 (0.24) | 9.71 (5.39) | 0.0 (0.0) | **22.96 (38.93)** | 2233.42 (3868.43) | **0.0 (0.0)** | **132.50 (130.32)** | 9212.24 (9840.56) | 7.86 (20.31) |
| Para Group 1 | 0.65 (0.59) | 9.01 (4.59) | 0.0 (0.0) | 47.36 (70.18) | 4580.9 (7028.59) | 4e+18 (1.96e+19) | 141.54 (124.79) | 8610.24 (8190.9) | 0.17 (0.28) |
| Para Group 2 | **0.36 (0.23)** | **7.91 (4.51)** | 0.0 (0.0) | 42.17 (79.8) | 1971.18 (3704.0) | 0.01 (0.03) | 167.64 (134.41) | 8661.57 (7499.68) | **0.16 (0.29)** |
| Para Group 3 | 0.65 (0.23) | 10.03 (5.31) | 0.0 (0.0) | 24.71 (54.74) | **1922.77 (3826.2)** | 0.0 (0.02) | 141.09 (125.39) | **8362.53 (6834.04)** | 0.17 (0.28) |

Table 10: Execution Time for each decision step of DynSep to configure separators

| | Set Covering | MIS | Knapsack | CORLAT | MIK | Anonymous | Load Balancing | Neos | Supportcase |
|---|---|---|---|---|---|---|---|---|---|
| Avg. Latency (s) | 0.33 | 0.22 | 0.09 | 0.05 | 0.31 | 0.41 | 3.04 | 0.11 | 0.39 |
| Max. Latency (s) | 1.09 | 1.01 | 0.71 | 0.70 | 1.07 | 1.94 | 4.51 | 2.02 | 6.85 |

that across all datasets, DynSep and its extended variants consistently outperform the default solver configuration reported in the main paper. Furthermore, the differences among the parameter groups are relatively small, yet certain combinations (e.g., Para Group 2 on Knapsack and Para Group 3 on mixed supportcase) yield further performance improvements in both Time and PD integral. These results validate that DynSep can flexibly extend to control a broader set of solver hyperparameters. Furthermore, carefully chosen parameter combinations can yield additional gains in solving speed and convergence.

### E.5 Overhead Evaluation

#### E.5.1 Latency of Policy Inference

We have provided per-decision latency (the latency of policy inference per decision step) and the total inference time through the solving process as follows.

**Per-decision latency.** Table 10 reports the average ("Avg. latency") and worst-case ("Max. latency") time taken for a single policy call across the entire branch-and-cut process. These values reflect the inference latency introduced by DynSep policy for each configuration decision.

Our results reveal that per-decision latency increases as instance size grows. On small to medium-sized datasets, the worst-case latency remains around 1 second per policy call. For the larger and more complex problem instances (with tens of thousands of variables and constraints), the worst-case latency ranges from 1 to 7 seconds per decision. In contrast, the average inference latency stays below 0.5 seconds across all datasets, with the exception of the largest load balancing instance (approximately 61K variables, 64K constraints), where the average latency rises to 3 seconds.

Table 11 provides the total inference time ("Infer. Time") over different datasets and solver time limits, along with the inference overhead rate ("Overhead Rate"), which represents the percentage of policy inference for the total solving time ("Sol. Time"). Table 11 shows that DynSep incurs a modest configuration overhead, contributing a negligible fraction of the total solving time even on large-scale instances. While the configuration time tends to increase with problem size—e.g., from under one second on small benchmarks to several tens of seconds on the largest ones—it remains practically affordable relative to the performance gains achieved. Nonetheless, there is potential to further optimize this step, and future work may explore more lightweight models to reduce configuration latency without compromising effectiveness.

Notably, our configuration policy is encapsulated within a custom SCIP separator. Thus, we report the total inference time via SCIP's built-in `SCIPsepaGetTime()` function. In contrast, the per-decision latency values are logged using wall-clock timing at each policy call, including overhead from time recording and logging. Consequently, the sum of the per-decision latencies naturally exceeds the total inference time, since the latter excludes the additional overhead introduced by frequent timing operations.

#### E.5.2 Memory Overhead

We analyze the memory inference overhead as follows.

**Per-decision memory overhead:** Table 12 below show the peak GPU memory usage per policy call.

Table 11: Inference Time for DynSep to configure separators during solving process. Three blocks: nine datasets in our manuscript (300-second time limit), four hard datasets (3600-second time limit), and MIPLIB MIRP & SRPN (600-second time limit).

| | Set Covering | MIS | Knapsack | CORLAT | MIK | Anonymous | Load Balancing | Neos | Supportcase |
|---|---|---|---|---|---|---|---|---|---|
| Infer. Time (s) | 1.05 (0.34) | 0.41 (0.23) | 0.42 (0.51) | 16.56 (21.07) | 1.57 (0.55) | 14.73 (20.58) | 10.54 (0.73) | 23.9 (27.99) | 14.16 (19.65) |
| Sol. Time (s) | 1.51 (0.27) | 0.53 (0.20) | 0.52 (0.24) | 22.96 (38.93) | 10.99 (9.44) | 241.89 (100.75) | 300.04 (0.08) | 235.19 (112.26) | 132.50 (130.32) |
| Overhead Rate (%) | 69.54 | 77.36 | 80.77 | 72.13 | 14.29 | 6.09 | 3.51 | 10.16 | 10.69 |

| | Anonymous | Load Balancing | MIPLIB mixed neos | MIPLIB mixed supportcase |
|---|---|---|---|---|
| Infer. Time (s) | 21.32 (16.64) | 16.01 (1.57) | 86.96 (114.55) | 12.31 (9.9) |
| Sol. Time (s) | 2397.95 (1551.2) | 3600.04 (0.07) | 2724.85 (1515.82) | 567.82 (1154.86) |
| Overhead Rate (%) | 0.89 | 0.44 | 3.19 | 2.17 |

| | MIPLIB MIRP | MIPLIB SRPN |
|---|---|---|
| Infer. Time (s) | 32.77 (29.27) | 5.06 (2.28) |
| Sol. Time (s) | 482.13 (205.78) | 294.77 (274.22) |
| Overhead Rate (%) | 6.80 | 1.72 |

Table 12: Per-decision memory overhead.

| | Set Covering | MIS | Knapsack | CORLAT | MIK | Anonymous | Load Balancing | Neos | Supportcase |
|---|---|---|---|---|---|---|---|---|---|
| Avg. CPU mem.(MB) | 2902.22 | 2804.20 | 2807.59 | 2786.58 | 2775.91 | 3153.15 | 3811.82 | 2824.13 | 2870.90 |
| Max. CPU mem. (MB) | 3081.70 | 2877.39 | 2889.04 | 2892.15 | 2877.77 | 3930.07 | 4852.27 | 3100.90 | 3688.11 |
| Avg. GPU mem. (MB) | 2.09 | 2.09 | 2.10 | 2.09 | 2.10 | 2.09 | 2.10 | 2.09 | 2.09 |
| Max. GPU mem. (MB) | 2.10 | 2.10 | 2.11 | 2.11 | 2.11 | 2.11 | 2.11 | 2.11 | 2.11 |

**peak memory overhead for overall inference:** The memory footprint required to store the encoder and policy model weights is 2.08 MB for each dataset. We track the peak GPU memories during the reference via the tool `torch.cuda.max_memory_allocated()`, which are list in Table 13.

## E.6 Ablation Study

### E.6.1 Module Ablation Analysis on Other Six Benchmarks

We evaluate DynSep and its ablated variants on other six benchmark datasets using solve time and the primal-dual (PD) gap integral as performance metrics. Table 14 summarizes the results. Specifically, the ablation study shows that while certain DynSep variants (e.g., w/o DynG&TF in Set Covering, w/o TF in MIS, w/o MaxR in Anonymous and Load Balancing) can slightly beat the full model on individual metrics for particular datasets, the full DynSep method consistently delivers robust, near-best performance overall. Overall, each component's removal yields trade-offs, but the full DynSep model demonstrates consistently balanced performance across tasks.

### E.6.2 Encoder Architecture Ablation

Table 15 shows that replacing the encoder's GCN with a custom bipartite graph transformer (GT): a multi-head TransformerConv for edge-aware message passing, followed by residual-connected LayerNorm and a two-layer feed-forward block. *GT-encoder* shows no consistent improvement over *DynSep*, which may be due to increased inference overhead or the need for finer stability/tuning to realize gains from the transformer-style aggregation.

### E.6.3 Hyperparameter Sensitivity Analysis

As shown in Table 16, we conducted robustness ablations over separator *frequency* (1, 5, 10, 20) and *maximum separation rounds* (3, 5, 10, 20) on three benchmarks (MIK, Corlat, and MIPLIB mixed Neos). Overall, across almost all tested settings, DynSep outperforms the default configuration, showing that the approach is reasonably stable to these hyperparameters. Below is our key observations.

*Frequency.* Moderate frequency (e.g., 5&10) gives the better trade-off. That is, small frequency causes separators to fire excessively, incurring high cut-generation overhead, whereas large frequency reduces opportunities for timely dual-bound tightening.

| | Set Covering | MIS | Knapsack | CORLAT | MIK | Anonymous | Load Balancing | Neos | Supportcase |
|---|---|---|---|---|---|---|---|---|---|
| Peak GPU mem. (MB) | 35.56 | 57.85 | 23.11 | 15.68 | 37.41 | 157.17 | 257.60 | 28.85 | 122.28 |

Table 14: Ablation results on other six benchmarks

| | Easy: Set Covering ($n = 1000$, $m = 500$) | | | Easy: Max Independent Set ($n = 500$, $m = 1953$) | | | Medium: MIK ($n = 413$, $m = 346$) | | |
|---|---|---|---|---|---|---|---|---|---|
| Method | Time(s) ↓ | Improv. ↑ (time, %) | PD integral ↓ | Time(s) ↓ | Improv. ↑ (time, %) | PD integral ↓ | Time(s) ↓ | Improv. ↑ (time, %) | PD integral ↓ |
| NoCuts | 7.45 (5.87) | NA | 101.86 (55.59) | 15.32 (5.82) | NA | 146.4 (56.99) | 190.28 (113.97) | NA | 887.85 (859.76) |
| Default | 5.24 (1.79) | 29.66 | 95.56 (36.86) | 30.4 (8.02) | -98.43 | 289.51 (103.81) | 16.65 (18.06) | 91.25 | 82.80 (56.24) |
| w/o MaxR | 1.32 (0.72) | 82.28 | 31.35 (11.32) | 0.57 (0.24) | 96.28 | 10.46 (2.83) | 12.46 (8.81) | 93.45 | 128.91 (67.89) |
| w/o TF | 1.48 (0.35) | 80.13 | 32.86 (6.25) | **0.44 (0.09)** | 97.13 | **9.16 (1.77)** | 14.04 (12.99) | 92.62 | **106.14 (65.99)** |
| w/o DynG | 1.25 (0.68) | 83.22 | 29.63 (10.34) | 0.56 (0.22) | 96.34 | 10.19 (2.68) | 11.88 (9.44) | 93.76 | 116.16 (40.64) |
| w/o DynG&TF | **1.23 (0.69)** | 83.49 | **29.33 (10.42)** | 0.55 (0.18) | 96.41 | 9.97 (2.28) | 12.43 (10.37) | 93.47 | 127.63 (46.03) |
| DynSep (Ours) | 1.51 (0.27) | 79.73 | 33.88 (9.34) | 0.53 (0.20) | 96.54 | 9.66 (2.40) | **10.99 (9.44)** | 94.22 | 134.15 (44.21) |

| | Hard: Anonymous ($n = 37881$, $m = 49603$) | | | Hard: Load Balancing ($n = 61000$, $m = 64304$) | | | Hard: MIPLIB mixed supportcase ($n = 19766$, $m = 19910$) | | |
|---|---|---|---|---|---|---|---|---|---|
| Method | Time(s) ↓ | PD integral ↓ | Improv. ↑ (PD Int., %) | Time(s) ↓ | PD integral ↓ | Improv. ↑ (PD Int., %) | Time(s) ↓ | PD integral ↓ | Improv. ↑ (PD Int., %) |
| NoCuts | 259.77 (75.71) | 21117.12 (9234.01) | NA | 300.11 (0.02) | 15093.26 (940.68) | NA | 181.26 (120.25) | 12959.99 (10506.47) | NA |
| Default | 298.92 (4.09) | 27069.58 (4892.8) | -28.19 | 300.14 (0.02) | 15187.19 (936.38) | -0.62 | 244.75 (105.8) | 21561.09 (10434.42) | -66.37 |
| w/o MaxR | 243.92 (97.55) | **14452.24 (9840.56)** | 31.56 | 300.13 (0.37) | **3252.99 (454.96)** | **78.45** | 167.52 (112.43) | 10158.06 (9568.77) | 21.62 |
| w/o TF | 251.16 (90.02) | 16238.64 (9292.53) | 23.10 | **300.02 (0.03)** | 3740.29 (473.88) | 75.22 | 143.54 (123.86) | 10253.13 (9952.05) | 20.89 |
| w/o DynG | 256.66 (77.74) | 16903.77 (8941.95) | 19.95 | 300.1 (0.02) | 15020.22 (941.63) | 0.48 | 145.76 (121.48) | 11882.87 (11695.35) | 8.31 |
| w/o DynG&TF | 246.48 (93.01) | 18914.82 (9388.91) | 10.43 | 300.04 (0.04) | 3923.57 (539.21) | 74.00 | **131.72 (130.92)** | 11369.53 (12085.07) | 12.27 |
| DynSep (Ours) | **241.89 (100.75)** | 15656.7 (8996.14) | 25.86 | 300.04 (0.08) | 3720.26 (499.37) | 75.35 | 132.50 (130.32) | **9212.24 (9840.56)** | 28.92 |

*Maximum Separation Round.* Setting this value too low produces weak cuts and degrades performance, while setting it excessively high increases the computational cost of cut generation. Although this parameter shows some dataset sensitivity, MaxRound=5 is empirically near-optimal in our tests.

## E.7   Generalization Study

We investigate complementary generalization performance of our method under two more settings: (1) **Cross-Domain Generalization Test**: training on one benchmark and testing on a dataset from a different domain; (2) **General-to-Specific Generalization Test**: training a general model on a mixed-category dataset (e.g. MIPLIB) and then evaluating on a specific class dataset.

### E.7.1   Cross-Domain Generalization Test

To evaluate the across-domain generalization, we train our policy on one problem family and apply the learned policy to unseen problem families. Specifically, we train four separate DynSep policies (on Setcover, Knapsack, MIK, and Supportcase) and evaluate each of them across all nine benchmark families used in our manuscript. The table below lists the results, where we report solving time for easy and medium datasets, while additionally report primal–dual integral (PD integral) for hard datasets.

As shown in Table 17, policies trained on one problem type yield improvements over SCIP's default in most unseen benchmarks, indicating effective transfer of our learned configuration strategy across NP-hard families.

### E.7.2   General-to-Specific Generalization Test

We select 168 diverse instances from the MIPLIB 2017 benchmark [14] as our training set and learn a separator configuration policy on these instances. Notably, because the MIPLIB 2017 instances cover a wide variety of problem types and mixed-scenario structures, this learned policy could serve as a general configuration model. We then evaluate it—without any additional tuning—on four unseen, domain-specific datasets: Corlat, Load Balancing (LB), Maritime Inventory Routing Problem (MIRP), and Seismic-Resilient Pipe Network Planning (SRPN). We have extended our evaluation beyond MIPLIB mixed neos and mixed supportcase, including two real-world datasets from the Distributional MIPLIB benchmark [49]:

Table 15: Comparative results of different encoder architectues for DynSep on four datasets.

| Method | Easy: Multiple Knapsack ($n = 720$, $m = 72$) | | | Medium: Corlat ($n = 466$, $m = 486$) | | |
| --- | --- | --- | --- | --- | --- | --- |
| | Time(s)↓ | Improv.↑ (time, %) | PD integral↓ | Time(s)↓ | Improv.↑ (time, %) | PD integral↓ |
| GT-encoder | 0.71 (0.39) | 94.87 | 11.02 (5.41) | 46.26 (71.41) | 38.04 | 4563.26 (7125.57) |
| DynSep (Ours) | **0.52 (0.24)** | **96.24** | **9.71 (5.39)** | **22.96 (38.93)** | **69.25** | **2233.42 (3868.43)** |

| Method | Hard: MIPLIB mixed neos ($n = 6958$, $m = 5660$) | | | Hard: MIPLIB mixed supportcase ($n = 19766$, $m = 19910$) | | |
| --- | --- | --- | --- | --- | --- | --- |
| | Time(s)↓ | PD integral↓ | Improv.↑ (PD Int., %) | Time(s)↓ | PD integral↓ | Improv.↑ (PD Int., %) |
| GT-encoder | 243.52 (97.82) | 12134.57 (12142.03) | 16.99 | 150.38 (123.13) | **9157.61 (9623.27)** | **29.34** |
| DynSep (Ours) | **235.19 (112.26)** | **8511.58 (12413.9)** | **41.78** | **132.50 (130.32)** | 9212.24 (9840.56) | 28.92 |

Table 16: Sensitivity Analysis of hyperparameters *Frequency* and *Maximum Separation Round* on three datasets.

| Method | Medium: MIK ($n = 413$, $m = 346$) | | | Medium: Corlat ($n = 466$, $m = 486$) | | | Hard: MIPLIB mixed neos ($n = 6958$, $m = 5660$) | | |
| --- | --- | --- | --- | --- | --- | --- | --- | --- | --- |
| | Time(s)↓ | Improv.↑ (time, %) | PD integral↓ | Time(s)↓ | Improv.↑ (time, %) | PD integral↓ | Time(s)↓ | PD integral↓ | Improv.↑ (PD Int., %) |
| NoCuts | 190.28 (113.97) | NA | 887.85 (859.76) | 74.66 (122.23) | NA | 2687.68 (6209.48) | 275.04 (43.23) | 14618.53 (12214.63) | NA |
| Default | 16.65 (18.06) | 91.25 | 82.80 (56.24) | 111.55 (132.19) | -49.41 | 10573.14 | 282.98 (29.49) | 18500.5 (9386.15) | -26.56 |
| Freq=1 | 12.74 (9.89) | 93.30 | 124.22 (43.24) | 53.77 (84.77) | 27.98 | 4582.15 (7398.26) | 252.12 (83.27) | 8756.3 (12305.23) | 40.10 |
| Freq=5 | 13.04 (11.04) | 93.15 | 86.55 (38.54) | 38.44 (49.72) | 48.51 | 3513.13 (4533.24) | 243.91 (97.16) | 9248.52 (12053.99) | 36.73 |
| Freq=10 (Ours) | **10.99 (9.44)** | **94.22** | 134.15 (44.21) | **22.96 (38.93)** | **69.25** | **2233.42 (3868.43)** | **235.19 (112.26)** | 8511.58 (12413.9) | 41.78 |
| Freq=20 | 12.83 (9.6) | 93.26 | 213.15 (124.09) | 49.14 (72.87) | 34.18 | 4658.09 (7252.41) | 261.86 (66.13) | 8789.54 (12249.16) | 39.87 |

| Method | Medium: MIK ($n = 413$, $m = 346$) | | | Medium: Corlat ($n = 466$, $m = 486$) | | | Hard: MIPLIB mixed neos ($n = 6958$, $m = 5660$) | | |
| --- | --- | --- | --- | --- | --- | --- | --- | --- | --- |
| | Time(s)↓ | Improv.↑ (time, %) | PD integral↓ | Time(s)↓ | Improv.↑ (time, %) | PD integral↓ | Time(s)↓ | PD integral↓ | Improv.↑ (PD Int., %) |
| NoCuts | 190.28 (113.97) | NA | 887.85 (859.76) | 74.66 (122.23) | NA | 2687.68 (6209.48) | 275.04 (43.23) | 14618.53 (12214.63) | NA |
| Default | 16.65 (18.06) | 91.25 | 82.80 (56.24) | 111.55 (132.19) | -49.41 | 10573.14 | 282.98 (29.49) | 18500.5 (9386.15) | -26.56 |
| MaxRound=3 | 13.63 (12.04) | 92.84 | 150.19 (46.77) | 107.26 (124.27) | -43.66 | 8965.26 (12401.06) | 239.22 (105.28) | 8581.5 (12378.07) | 41.30 |
| MaxRound=5 (Ours) | **10.99 (9.44)** | **94.22** | 134.15 (44.21) | **22.96 (38.93)** | **69.25** | **2233.42 (3868.43)** | **235.19 (112.26)** | 8511.58 (12413.9) | 41.78 |
| MaxRound=10 | 17.6 (12.68) | 90.75 | 182.71 (121.39) | 40.45 (68.93) | 45.82 | 3717.64 (6764.77) | 249.41 (87.63) | 8846.91 (12224.34) | 39.48 |
| MaxRound=20 | 16.2 (11.25) | 91.49 | 169.8 (71.61) | 36.71 (55.75) | 50.83 | 3609.11 (5562.51) | 252.59 (82.11) | 10945.82 (11486.55) | 25.12 |

As shown in Table 18, the general configuration model consistently outperforms the solver's default settings in both solve time and convergence behavior, demonstrating good generalizability of our method.

## E.8 Visualization of Separator Configurations on Nine Benchmarks

We provide visualization of separator configurations on nine benchmarks in Figs. 7- 15. Figs. 7 and 8 show that our learned policy uniformly uniformly reduces the maximum number of separation rounds to $r_{\max} = 3$ for easy benchmarks, Set Covering and MIS, demonstrating that our learned decision on maximum rounds effectively prunes unnecessary cutting rounds on simple problems. The heatmap reveals that the separator configuration is not static but varies dynamically across separation rounds (shown along the y-axis), suggesting that the model is timing the application of various separators to coincide with the stage of cut generation. Furthermore, the fact that learned activation values are not restricted to $\{-1, 0, 1\}$ but take intermediate real values indicates the policy differentiates between individual instances (and even between nodes) when selecting separators. In other words, it has learned a nuanced, instance-wise (and node-wise) cutting strategy rather than a one-size-fits-all rule.

Table 17: Cross-domain generalization on nine benchmarks. Policies are trained on one problem family and evaluated on unseen families.

| | Easy: Set Covering ($n = 1000$, $m = 500$) | | Easy: Max Independent Set ($n = 500$, $m = 1953$) | | Easy: Multiple Knapsack ($n = 720$, $m = 72$) | |
|---|---|---|---|---|---|---|
| Method | Time(s) ↓ | PD integral ↓ | Time(s) ↓ | PD integral ↓ | Time(s) ↓ | PD integral ↓ |
| Default | 5.24 (1.79) | 95.56 (36.86) | 30.4 (8.02) | 289.51 (103.81) | 2.01 (1.82) | 18.6 (10.49) |
| Train on Setcover | NA | NA | 1.0 (1.38) | 13.57 (9.99) | **0.68 (0.42)** | 10.73 (6.12) |
| Train on Knapsack | **2.02 (0.62)** | **40.96 (9.93)** | 0.76 (0.29) | 12.37 (3.53) | NA | NA |
| Train on MIK | 7.74 (4.33) | 80.78 (40.05) | 5.21 (3.38) | 29.81 (19.2) | 1.2 (1.09) | 11.91 (5.52) |
| Train on supportcase | 2.21 (0.59) | 43.95 (9.35) | **0.67 (0.3)** | **11.72 (3.63)** | 0.78 (1.06) | **10.65 (5.97)** |

| | Medium: Corlat ($n = 466$, $m = 486$) | | Medium: MIK ($n = 413$, $m = 346$) | | Hard: Anonymous ($n = 37881$, $m = 49603$) | |
|---|---|---|---|---|---|---|
| Method | Time(s) ↓ | PD integral ↓ | Time(s) ↓ | PD integral ↓ | Time(s) ↓ | PD integral ↓ |
| Default | 111.55 (132.19) | 10573.14 (13070.46) | 16.65 (18.06) | 82.80 (56.24) | 298.92 (4.09) | 27069.58 (4892.8) |
| Train on Setcover | 43.79 (76.54) | 4042.34 (7652.31) | 24.8 (21.41) | 139.1 (45.45) | **250.46 (87.15)** | 19701.65 (9639.56) |
| Train on Knapsack | 27.56 (58.48) | 2485.67 (5777.55) | **12.48 (10.29)** | **139.08 (44.39)** | 253.74 (80.42) | 19183.3 (8941.1) |
| Train on MIK | 34.54 (69.18) | 2163.38 (4884.15) | NA | NA | 268.43 (56.97) | 20715.25 (8615.27) |
| Train on supportcase | **20.94 (50.99)** | **2016.37 (5095.97)** | 163.6 (98.18) | 785.46 (577.89) | 256.33 (79.04) | **17922.89 (8384.44)** |

| | Hard: Load Balancing ($n = 61000$, $m = 64304$) | | Hard: MIPLIB mixed neos ($n = 6958$, $m = 5660$) | | Hard: MIPLIB mixed supportcase ($n = 19766$, $m = 19910$) | |
|---|---|---|---|---|---|---|
| Method | Time(s) ↓ | PD integral ↓ | Time(s) ↓ | PD integral ↓ | Time(s) ↓ | PD integral ↓ |
| Default | 300.14 (0.02) | 15187.19 (936.38) | 282.98 (29.49) | 18500.5 (9386.15) | 244.75 (105.8) | 21561.09 (10434.42) |
| Train on Setcover | 300.05 (0.05) | **4411.86 (514.13)** | 258.22 (72.37) | 13512.27 (12604.76) | **141.31 (125.51)** | 12171.62 (11385.67) |
| Train on Knapsack | **300.04 (0.05)** | 4583.45 (554.52) | 256.34 (75.63) | **13343.38 (12514.04)** | 166.9 (116.16) | 13216.93 (11281.29) |
| Train on MIK | **300.04 (0.05)** | 5559.96 (1336.87) | 249.83 (86.89) | 13366.13 (12606.74) | 187.09 (127.76) | **11583.1 (9504.89)** |
| Train on supportcase | 300.09 (0.31) | 9421.93 (665.37) | **246.91 (91.95)** | 13639.73 (12439.81) | NA | NA |

Table 18: Generalization performance of our DynSep model trained on MIPLIB 2017, evaluated on four unseen MILP scenarios. (300-second time limit for Corlat & LB; 600-second for MIRP & SRPN)

| | Medium: Corlat ($n = 466$, $m = 486$) | | Hard: Load Balancing ($n = 61000$, $m = 64304$) | |
|---|---|---|---|---|
| Method | Time(s) ↓ | PD integral ↓ | Time(s) ↓ | PD integral ↓ |
| Default | 5.24 (1.79) 2.01 (1.82) | 95.56 (36.86) 18.6 (10.49) | 30.4 (8.02) | 289.51 (103.81) |
| DynSep (ours) trained on MIPLIB 2017 | **46.05** | **4289.33** | **300.08** | **4792.71** |

| | Hard: MIPLIB MIRP ($n = 34656$, $m = 44430$) | | Hard: MIPLIB SRPN ($n = 6032$, $m = 5917$) | |
|---|---|---|---|---|
| Method | Time(s) ↓ | PD integral ↓ | Time(s) ↓ | PD integral ↓ |
| Default | 580.98 (61.65) | 52362.46 (13476.17) | 332.04 (271.0) | 11687.01 (11993.95) |
| DynSep (ours) trained on MIPLIB 2017 | **487.59** | **33193.25** | **288.48** | **7582.60** |

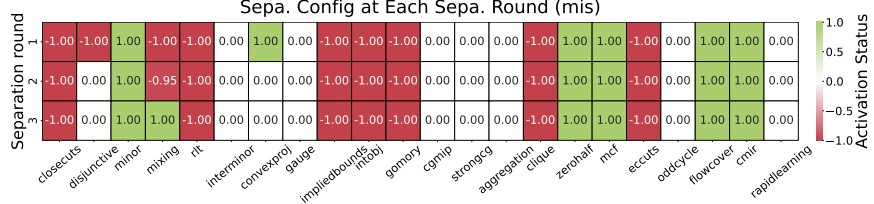

Figure 7: **Separator configs at each separation round of Set Covering benchmark.**

Figure 8: **Separator configs at each separation round for Maximum Independent Set benchmark.**

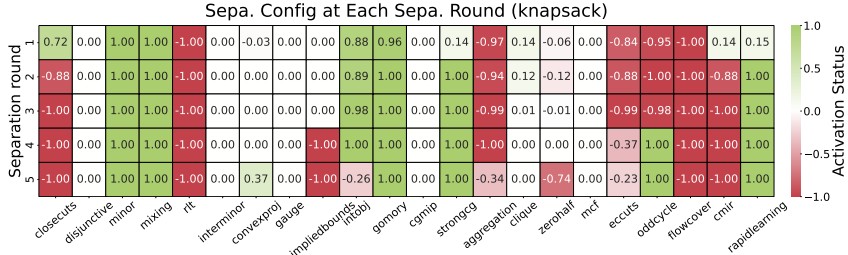

Figure 9: **Separator configs at each separation round for Multiple Knapsack benchmark**.

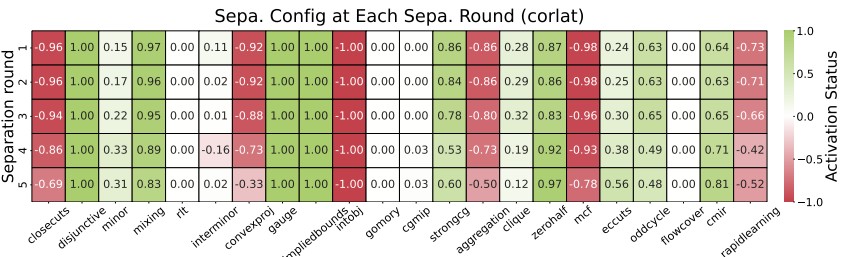

Figure 10: **Separator configs at each separation round for CORLAT benchmark**.

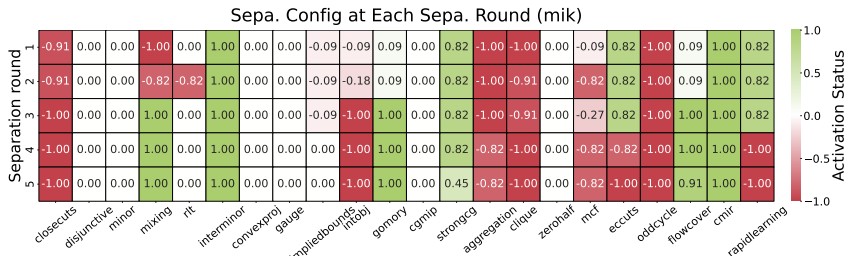

Figure 11: **Separator configs at each separation round for MIK benchmark**.

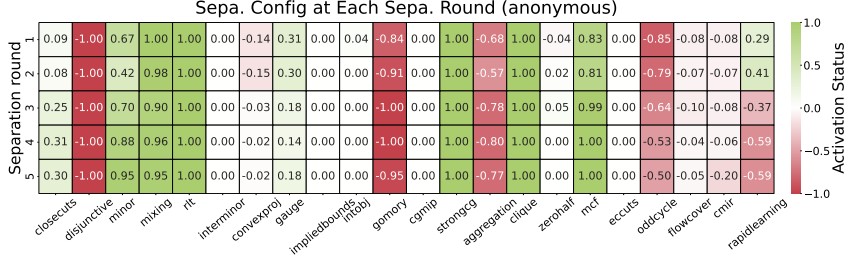

Figure 12: **Separator configs at each separation round for Anonymous benchmark**.

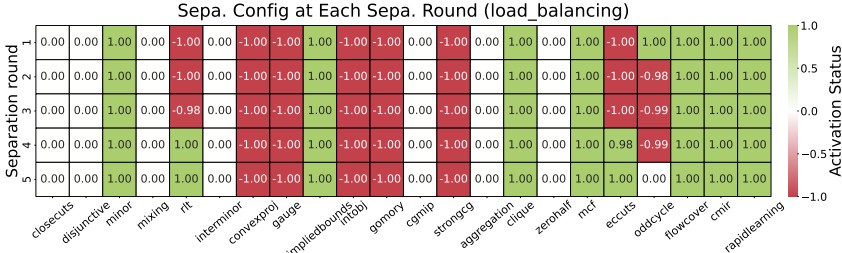

Figure 13: **Separator configs at each separation round for Load Balancing benchmark**.

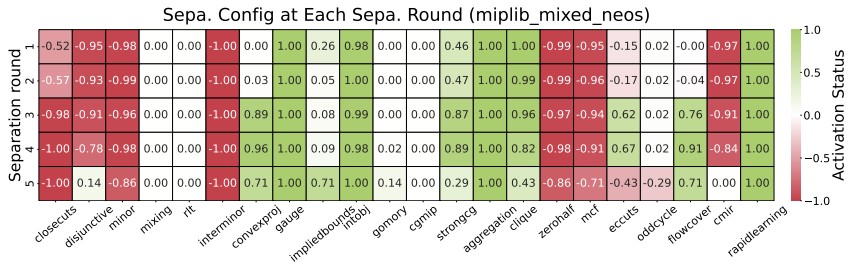

Figure 14: **Separator configs at each separation round for MIPLIB mixed neos benchmark**.

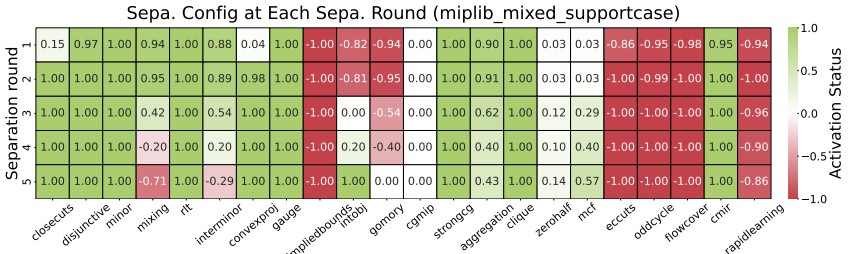

Figure 15: **Separator configs at each separation round for MIPLIB mixed supportcase benchmark**.

