# OpenReview forum: "Dynamic Configuration for Cutting Plane Separators via Reinforcement Learning on Incremental Graph"
_NeurIPS.cc/2025/Conference — NeurIPS 2025 poster_

### Official Review · Reviewer_Fjx6 · 2025-06-24

**Clarity:** 2
**Significance:** 2
**Originality:** 2
**Rating:** 4
**Confidence:** 4

**Summary:**

The paper proposes DynSep, a reinforcement learning-based method for dynamically configuring cutting plane separators in mixed-integer linear programming (MILP) solvers.

**Questions:**

(1) Could you specify which instances you used to test the performance in MIPLIB. What do you mean by mixed neos and mixed suportcase specifically?
(2) For real-world use, what is the latency of policy inference during B&C?

**Ethical Concerns:**

["NO or VERY MINOR ethics concerns only"]

**Final Justification:**

At the last minutes of rebuttal, the author provided more evaluation experiments and addressed my concerns.

**Limitations:**

(1) DynSep ​only optimizes separator activation status​ (+1, 0, -1) and round termination (m_t). It ignores critical hyperparameters like:
Cut depth/aggressiveness (e.g., sepastore/age in SCIP); Cut selection thresholds (e.g., efficacy vs. orthogonality) and Separation frequency per node.

(2) Memory/compute inference overhead is not discussed. Which scenarios and practical cases the data-driven approach is appropriate should be addressed.

**Quality:**

2

**Strengths And Weaknesses:**

The strength of this paper:

(1) Triplet graphs are used to capture evolving problem states during branch and cut with Lightweight subgraph updates
(2) Decoder-only design with blocked positional encoding captures temporal dependencies across rounds while ensuring permutation equivariance

The weakness of this paper:

(1) limited scope of hyperparameter tuning (separator activation/timing)
(2) limited scope of testing datasets (MIPLIB neos and supportcase)

---

> ### Author Rebuttal · Authors · 2025-07-30
>
> We thank the reviewer for the insightful and valuable comments. We respond to your comments as follows and sincerely hope that our rebuttal could properly address your concerns. If so, we would deeply appreciate it if you could raise your score（"3: Borderline Reject"). If not, please let us know your further concerns, and we will continue actively responding to your comments and improving our submission.
>
> **Q1. Lmited scope of hyperparameter tuning, ignoring critical hyperparameters.**
>
> **A1.**
>
> Thank you for the insightful comment. **We claim that our proposed DynSep framework is inherently extensible to a broader set of solver hyperparameters beyond separator activation and timing.** This is achieved by adapting the policy network: we model the outputs of additional parameters as a logistic‑normal distribution, followed by optional discretization to support both integer and continuous parameters. This enables flexible, differentiable control of arbitrary solver parameters.
>
> To validate this, we conducted additional experiments on three critical hyperparameter groups (as suggested), while retaining the original tuning mechanism for separator activation and max round.
>
> + **para group 1: Cut Depth / Aggressiveness (sepastore/age in SCIP).** Controlled by solver parameters `separating/cutagelimit` and `separating/poolfreq`.
> + **para group 2: Cut selection thresholds (e.g., efficacy vs. orthogonality).** Controlled by solver parameters `separating/minefficacy` and `cutselection/hybrid/minortho`.
> + **para group 3: Separation frequency per node.** Controlled by solver parameters `separating/cutagelimit` and `separating/poolfreq`.
>
> We provide the experimental results as follows.
>
> |Instance|Knapsack|Knapsack|Knapsack|Corlat|Corlat|Corlat|Supportcase|Supportcase|Supportcase|
> |-|-|-|-|-|-|-|-|-|-|
> |Metric|Time(s)|PD integral|PD gap|Time(s)|PD integral|PD gap|Time(s)|PD integral|PD gap|
> |DynSep (Ours)|0.52|9.71|0|**22.96**|2233.42|**0**|**132.50**|9212.24|7.86|
> |Para Group 1|0.65|9.01|0|47.36|4580.9| 4e+18  |141.54|8610.24|0.17|
> |Para Group 2|**0.36**|**7.91**|0|42.17|1971.18|0.01|167.64|8661.57|**0.16**|
> |Para Group 3|0.65|10.03|0|24.71|**1922.77**|0|141.09|**8362.53**|0.17|
>
> We evaluated our method on three benchmark datasets: **Knapsack**, **Corlat**, and **MIPLIB mixed supportcase**, measuring solving time (*Time*), primal–dual gap integral (*PD integral*), and primal–dual gap (*PD gap*). All three metrics are *lower-is-better*.
>
> We compared our original DynSep method with its extended versions incorporating three additional parameter groups. Results show that:
>
> 1. Across all datasets, **DynSep and its extended variants consistently outperform the default solver configuration** reported in the main paper.
>
> 2. The differences among the parameter groups are relatively small, yet certain combinations yield **further performance improvements** in both Time and PD integral.
>
> These results validate that DynSep can flexibly **extend to control a broader set of solver hyperparameters**. Furthermore, carefully chosen parameter combinations can yield **additional gains** in solving speed and convergence.
>
>
> **Q2. Limited scope of testing datasets (MIPLIB neos and supportcase)；**
>
> **A2.**
>
> Thank you for raising this point. We **have extended our evaluation beyond MIPLIB mixed neos and mixed supportcase**, including two real-world datasets from the Distributional MIPLIB benchmark [1]:
>
> + **Maritime Inventory Routing Problem (MIRP).** MIRP arises in bulk shipping logistics, integrating vessel routing and port inventory decisions, featuring 15080 binary variables, 19576 continuous variables, and 44430 constraints.
> + **Seismic‑Resilient Pipe Network Planning (SRPN).** SRPN involves optimizing municipal water pipe network design to ensure resilience under seismic disturbances, featuring 3016 binary variables, 3016 continuous variables, and 5917 constraints.
>
> We set the time limit to $600$ seconds. The results are summarized in the table below.
>
> |Instance|mirp|mirp|mirp|srpn|srpn|srpn|
> |-|-|-|-|-|-|-|
> |Metric|Time(s)|PD integral|PD gap|Time(s)|PD integral|PD gap|
> |NoCuts|486.68|34735.55|6.67e+18|**280.7**|11020.69|0.28|
> |Default|580.98|52362.46|5.38e+19|332.04|11687.01|0.21|
> |Search(20)|492.85|33028.81|6.30e+18|384.02|14571.56|0.26|
> |Prune|501.49|35542.82|8.33e+18|296.08|9211.76|0.19|
> |LLM4sepasel|511.97|34573.52|5e+18|300.66|8421.65|0.14|
> |DynSep(Ours)|**482.13**|**30838.39**|**1.38**|294.77|**7581.16**|**0.1**|
>
> These results show that our method (**DynSep**) delivers **marked performance gains** on additional real‑world datasets (MIRP and SRPN). Compared to the other configuration methods, DynSep significantly improves both **convergence speed** (evidenced by reduced PD integral) and **solution quality** (evidenced by lower PD gap).
>
>
> **Q3. Specification of MIPLIB mixed neos and mixed supportcase.**
>
> **A3.**
>
> We apologize for the confusion. *MIPLIB mixed neos* and *mixed supportcase* evaluated in our paper are two well-established subsets derived from MIPLIB 2017 [2], which were constructed by the prior learning-based MILP research [3].
>
> **Each subset comprises a cluster of similar instances selected from MIPLIB 2017, where similarity is quantified using 100 human-designed instance features [1]**. Briefly, a seed instance is first selected: *neos‑1456979* for the mixed neos set, representing knapsack‑constraint problems, and *supportcase40* for the mixed supportcase set, representing set‑packing problems. Then, a data-driven clustering algorithm [3] is applied to gather MIPLIB instances that closely match the seed instance in feature space, constructing the mixed neos and mixed supportcase subsets.
>
> Due to **space limitations** in the rebuttal, we **only present a subset of** MIPLIB mixed neos and mixed supportcase datasets used in our experiments:
>
> + Mixed neos contains 17 instances: neos-1456979, icir97 tension, etc.
>
> + Mixed supportcase contains 37 instances: supportcase40, acc-tight2, acc-tight4, etc.
>
>
> **Q4. The latency of policy inference during B&C.**
>
> **A4.**
>
> Thank you for raising this point. We have provided **per-decision latency** (the latency of policy inference per decision step) and the **total inference time** through the solving process as follows.
>
> + **Per-decision latency**: The table below reports the **average** ("Avg. latency") and **worst-case** ("Max. latency") time taken for a single policy call across the entire branch-and-cut process.
>
> ||Set Cover|MIS|Knapsack|Corlat|MIK|Anonymous|LB|Neos|Supportcase|
> |-|-|-|-|-|-|-|-|-|-|
> |Avg. Latency (s)|0.33|0.22|0.09|0.05|0.31|0.41|3.04|0.11|0.39|
> |Max. Latency (s)|1.09|1.01|0.71|0.70|1.07|1.94|4.51|2.02|6.85|
>
> Our results reveal that **per-decision latency increases as instance size grows**. The **average inference latency** stays below 0.5 seconds across all datasets, with the exception of the largest load balancing (LB) instance.
>
> ---
>
> The two tables below provide the **total inference time** ("Infer. Time") over different datasets and solver time limits, along with the **inference overhead rate** ("Overhead Rate"), which represents the percentage of policy inference in the total solving runtime.
>
> + **Total inference time (nine datasets in our manuscript, $300$-second time limit):**
>
> ||SetCover|MIS|Knapsack|Corlat|MIK|Anonymous|LB|Neos|Supportcase|
> |-|-|-|-|-|-|-|-|-|-|
> |Infer. Time (s)|1.05| 0.41  |0.42|16.56| 1.57  |14.73|10.54| 23.9 |14.16|
> |Overhead Rate (%)|69.54|77.36|80.77|72.13|14.29|6.09|3.51|10.16|10.69|
>
> + **Total inference time (hard datasets in our manuscript, $3600$-second time limit):**
>
> ||Anonymous|LB|Neos|Supportcase|
> |-|-|-|-|-|
> |Infer. Time (s)|21.32|16.01|86.96|12.31|
> |Overhead Rate (%)|0.89|0.44|3.19|2.17|
>
> These tables show that on larger, more complex instances and with longer solver time limits, **a smaller share** of the total solving time is spent on decision making. This implies that **the computational cost of our policy calls becomes more justifiable at scale**, enabling meaningful improvements in overall solver efficiency with only modest overhead investment.
>
>
> **Q5. Memory/compute inference overhead is not discussed.**
>
> **A5.**
>
> We have analyzed the compute inference overhead in Q4. Here, we focus on the memory inference overhead.
>
> + **Per-decision memory overhead:** The table below show the peak GPU memory usage per policy call.
>
> ||SetCover|MIS|Corlat|LB|Neos|
> |-|-|-|-|-|-|
> |Max. CPU mem. (MB)|3081.70|2877.39|2892.15|4852.27|3100.90|
> |Max. GPU mem. (MB)|2.10|2.10|2.11|2.11|2.11|
>
> + **Peak memory overhead for overall inference:**
>
> The memory footprint required to store the encoder and policy model weights is $2.08$ MB for each dataset. We track the peak GPU memories during the reference via the tool `torch.cuda.max_memory_allocated()`, which are list in the table below.
>
> || SetCover |MIS|Corlat|LB|Neos|
> |-|-|-|-|-|-|
> |Peak GPU mem. (MB)|35.56|57.85|15.68|257.60|28.85|
>
>
> **Q6. Which scenarios and practical cases the data-driven approach is appropriatML-guided MILPe should be addressed.**
>
> **A6.**
>
> Our data‑driven configuration approach proves effective across both **synthetic** and **real‑world** MILP benchmarks [1,2,3]. It is particularly well‑suited to scenarios where one has access to a batch of instances drawn from the **similar distribution** as the unseen examples. In these cases, the policy network can be trained on those example instances and then deployed during branch‑and‑cut to flexibly tune solver parameters via the solver’s existing parameter interfaces, dynamically adapting to individual problem structure.
>
> [1] Huang, et al. "Distributional MIPLIB: a multi-domain library for advancing ml-guided milp methods." arXiv 2024.
>
> [2] Gleixner, et al. "MIPLIB 2017: data-driven compilation of the 6th mixed-integer programming library." Mathematical Programming Computation(2021).
>
> [3] Wangi, et al. "Learning cut selection for mixed-integer linear programming via hierarchical sequence model."ICLR (2023).

---

> > ### Comment · Reviewer_Fjx6 · 2025-08-05
> >
> > Since cutting-plane methods play a critical role in solving MILPs, my concerns remain unresolved.
> >
> > (1) Hyperparameters have a significant impact on MILP performance, and even a single cut method—such as the Flow Cover Inequalities Separator in Gurobi—can involve a large number of parameters. The separator mechanism alone includes dozens of them. Therefore, fine-tuning only a limited subset of parameters, as done in the current submission and the rebuttal, is insufficient to fully assess the method's effectiveness.
> >
> > (2) The performance concern also remains. Achieving strong results on a few instances is not particularly challenging. I am interested in seeing how the proposed method performs on the full set of 240 instances in MIPLIB2017, rather than just a limited subset.

---

> > > ### Author Response · Authors · 2025-08-08
> > >
> > > > Q1. Fine-tuning a limited subset of parameters is insufficient to fully assess the method's effectiveness.
> > >
> > > Thank you for the valuable feedback. We have expanded our separator parameter set, **from a total number of $46$ to $144$**, more than tripling its size. The expanded parameter space includes shared or **separator-specific parameters** and covers **various numeric types** (int, float, bool). The results in Table 1 show that our method with an enlarged parameter space delivers **substantial performance gains** over the default solver setting. Below is the detailed description.
> > >
> > > In our original manuscript, our configuration policy controlled $46$ parameters in SCIP: namely,
> > >
> > > + the global parameters for all separators: `separating/maxrounds` and `separating/maxroundsroot` (2 parameters).
> > > + for each of the 22 built-in separators, both its `separating/<separator_name>/freq` and `separating/<separator_name>/priority` parameters (22 × 2 = 44 parameters).
> > >
> > > We appreciate the reviewer’s point that a single separator can involve multiple tuning parameters beyond frequency and priority. Thus, we expand our configuration scope to include approximately 5–7 internal hyperparameters per separator, yielding **a total of 144 parameters, more than three times** our previous controlled parameters.
> > >
> > > As shown in **Table 1**, our learned configuration policy—now encompassing the enlarged parameter space—delivers substantial performance gains over the default solver configuration.
> > >
> > > *Table 1. Evaluation for the default setting (Default) and our configuration method with an enlarged parameter space (DynSep-expended).*
> > >
> > > |Instance|Knapsack|Knapsack|Corlat|Corlat|neos|neos|
> > > |-|-|-|-|-|-|-|
> > > |Metric|Time(s)|PD integral|Time(s)|PD integral|Time(s)|PD integral|
> > > |Default|2.01|18.6|111.55|10573.14|282.98|18500.5|
> > > |DynSep-expended (ours)|**0.57**|**10.46**|**56.03**|**5161.01**|**249.43**|**8602.66**|
> > >
> > >
> > > Compared to traditional automated configuration tools such as ParamILS [1]—typically targeting **81 CPLEX parameters**—and SMAC [2]—focusing on **76 parameters**—as well as recent ML approaches like Deep Metric Learning [3], which tune **17 parameters**, our method supports **a larger and extensible parameter space**. DynSep's policy employs a unified and scalable framework capable of handling both shared and separator-specific parameters across multiple data types (`int`, `float`, `bool`).
> > >
> > > The comprehensive list of newly included parameters is detailed in **Table 2**. Note that there are $142$ separator parameters in the table. Adding two global parameters—`separating/maxrounds` and `separating/maxroundsroot`—makes a total of $144$ parameters.
> > >
> > > *Table 2. List of separator parameters controlled by our configuration policy.*
> > >
> > > |Separator|Shared Params|Specific Params|Separator|Shared Params|Specific Params|
> > > |-|-|-|-|-|-|
> > > |closecuts|freq, priority, maxbounddist, delay, expbackoff|sepacombvalue, closethres|cgmip|freq, priority, maxbounddist, delay, expbackoff|minnodelimit, maxnodelimit|
> > > |disjunctive|freq, priority, maxbounddist, delay, expbackoff|maxinvcuts, maxrank|strongcg|freq, priority, maxbounddist, delay, expbackoff|/|
> > > |minor|freq, priority, maxbounddist, delay, expbackoff|maxminorsconst,  maxminorsfac|aggregation|freq, priority, maxbounddist, delay, expbackoff|maxaggrs, maxsepacuts|
> > > |mixing|freq, priority, maxbounddist, delay, expbackoff|uselocalbounds, iscutsonints|clique|freq, priority, maxbounddist, delay, expbackoff|maxsepacuts, scaleval|
> > > |rlt|freq, priority, maxbounddist, delay, expbackoff|maxusedvars, goodscore|zerohalf|freq, priority, maxbounddist, delay, expbackoff|maxsepacuts, minviol|
> > > |interminor|freq, priority, maxbounddist, delay, expbackoff|mincutviol|mcf|freq, priority, maxbounddist, delay, expbackoff|maxsepacuts, nclusters|
> > > |convexproj|freq, priority, maxbounddist, delay, expbackoff|nlpiterlimit|eccuts|freq, priority, maxbounddist, delay, expbackoff|maxsepacuts, minviolation|
> > > |gauge|freq, priority, maxbounddist, delay, expbackoff|nlpiterlimit|oddcycle|freq, priority, maxbounddist, delay, expbackoff|maxsepacuts, scalingfactor|
> > > |impliedbounds|freq, priority, maxbounddist, delay, expbackoff|usetwosizecliques|flowcover|freq, priority, maxbounddist, delay, expbackoff|/|
> > > |intobj|freq, priority, maxbounddist, delay, expbackoff|/|cmir|freq, priority, maxbounddist, delay, expbackoff|/|
> > > |gomory|freq, priority, maxbounddist, delay, expbackoff|maxsepacuts, away|rapidlearning|freq, priority, maxbounddist, delay, expbackoff|maxcalls, maxnodes|
> > >
> > > ---
> > > [1] Hutter, Frank, et al. "ParamILS: an automatic algorithm configuration framework." Journal of artificial intelligence research 36 (2009).
> > >
> > > [2] Hutter, Frank, et al.  "Sequential model-based optimization for general algorithm configuration." International conference on learning and intelligent optimization (2011).
> > >
> > > [3] Hosny, Abdelrahman, and Sherief Reda. "Automatic MILP solver configuration by learning problem similarities." Annals of Operations Research (2024).

---

> > > > ### Author Response · Authors · 2025-08-08
> > > >
> > > > >  Q2. How the proposed method performs on the full set of 240 instances in MIPLIB2017?
> > > >
> > > > Thank you for your valuable feedback. We have tested our method on **the full set of 240 MIPLIB 2017 benchmark instances**. The results in Table 3 show that our approach (DynSep) delivers **notable improvements in solving efficiency** in the challenging MIPLIB2017 dataset.
> > > >
> > > > Specifically, we set the time limit as $300$ seconds and excluded five instances whose presolving time exceeded 300 seconds: *neos‑3402454‑bohle*, *neos‑4722843‑widden*, *mzzv42z*, *neos‑5052403‑cygnet*, *proteindesign121hz512p9*, and *proteindesign122trx11p8*, which is a common removing criterion for MIPLIB2017 benchmark [4,5]
> > > >
> > > > The remaining **235 instances** were split into a **70% training set and a 30% test set**. **Table 3** reports the overall average performance of our method across all 235 instances, and **Table 4** provides detailed results for every individual instance in the MIPLIB 2017 benchmark. These experiments confirm that our approach delivers **notable improvements in solving efficiency**, even when evaluated on the more challenging benchmark set.
> > > >
> > > > [4] Turner, Mark, et al. "Adaptive cut selection in mixed-integer linear programming." *arXiv preprint arXiv:2202.10962* (2022).
> > > >
> > > > [5] Wang, Jie, et al. "Learning to cut via hierarchical sequence/set model for efficient mixed-integer programming." *IEEE Transactions on Pattern Analysis and Machine Intelligence* (2024).
> > > >
> > > > ---
> > > >
> > > > *Table 3. Comparison between default setting and our method (DynSep) on all 235 MIPLIB 2017 instances.*
> > > >
> > > > |Instance|Miplib2017|Miplib2017|
> > > > |-|-|-|
> > > > |Metric|Time(s)|PD integral|
> > > > |Default|258.77|17153.69|
> > > > |DynSep (ours)|**238.88**|**15092.16**|
> > > >
> > > >
> > > > ---
> > > >
> > > > *Table 4. Per-instance comparison of Default and our method (DynSep) on the full set of 235 MIPLIB 2017 benchmark.*
> > > > ||instance_name|solving_time|primaldualintegral|Method|instance_name|solving_time|primaldualintegral|Method|
> > > > |-|-|-|-|-|-|-|-|-|
> > > > |0|30n20b8|102.77|7918.64|DynSep(Ours)|neos-4387871-tavua|300.00|13809.79|DynSep(Ours)|
> > > > |1|30n20b8|294.10|21053.08|Default|neos-4387871-tavua|300.00|26555.13|Default|
> > > > |2|50v-10|300.00|3753.26|DynSep(Ours)|neos-4413714-turia|300.04|30003.96|DynSep(Ours)|
> > > > |3|50v-10|300.00|3553.53|Default|neos-4413714-turia|302.13|30213.00|Default|
> > > > |4|CMS750_4|119.32|8927.87|DynSep(Ours)|neos-4532248-waihi|300.03|30002.91|DynSep(Ours)|
> > > > |5|CMS750_4|300.20|21969.51|Default|neos-4532248-waihi|300.23|10000.84|Default|
> > > > |6|academictimetablesmall|300.00|30000.43|DynSep(Ours)|neos-4647030-tutaki|300.07|15994.51|DynSep(Ours)|
> > > > |7|academictimetablesmall|302.93|30292.50|Default|neos-4647030-tutaki|302.40|20970.49|Default|
> > > > |8|air05|28.47|724.52|DynSep(Ours)|neos-4738912-atrato|300.00|903.73|DynSep(Ours)|
> > > > |9|air05|222.33|2019.36|Default|neos-4738912-atrato|300.01|992.78|Default|
> > > > |10|app1-1|6.09|594.63|DynSep(Ours)|neos-4763324-toguru|304.60|21333.61|DynSep(Ours)|
> > > > |11|app1-1|18.49|1695.55|Default|neos-4763324-toguru|303.26|25246.69|Default|
> > > > |12|app1-2|187.93|15555.94|DynSep(Ours)|neos-4954672-berkel|300.00|10422.23|DynSep(Ours)|
> > > > |13|app1-2|300.33|29112.20|Default|neos-4954672-berkel|300.00|10030.91|Default|
> > > > |14|assign1-5-8|1.26|18.48|DynSep(Ours)|neos-5049753-cuanza|300.27|30027.25|DynSep(Ours)|
> > > > |15|assign1-5-8|31.90|490.61|Default|neos-5049753-cuanza|300.35|30035.39|Default|
> > > > |16|atlanta-ip|300.00|30000.45|DynSep(Ours)|neos-506428|303.78|30377.81|DynSep(Ours)|
> > > > |17|atlanta-ip|302.42|30241.88|Default|neos-506428|301.50|30150.24|Default|
> > > > |18|b1c1s1|300.00|16908.55|DynSep(Ours)|neos-5093327-huahum|300.00|18587.41|DynSep(Ours)|
> > > > |19|b1c1s1|300.04|27161.92|Default|neos-5093327-huahum|300.22|27226.51|Default|
> > > > |20|bab2|300.36|30036.36|DynSep(Ours)|neos-5104907-jarama|300.35|30034.80|DynSep(Ours)|
> > > > |21|bab2|300.43|30042.55|Default|neos-5104907-jarama|300.47|30046.90|Default|
> > > > |22|bab6|323.32|32331.94|DynSep(Ours)|neos-5107597-kakapo|300.00|28661.57|DynSep(Ours)|
> > > > |23|bab6|300.18|30017.88|Default|neos-5107597-kakapo|300.05|30004.63|Default|
> > > > |24|beasleyC3|300.05|6123.19|DynSep(Ours)|neos-5114902-kasavu|301.44|30144.12|DynSep(Ours)|
> > > > |25|beasleyC3|32.49|1394.72|Default|neos-5114902-kasavu|300.74|10041.43|Default|
> > > > |26|binkar10_1|300.00|283.00|DynSep(Ours)|neos-5188808-nattai|300.00|30000.33|DynSep(Ours)|
> > > > |27|binkar10_1|172.39|1093.98|Default|neos-5188808-nattai|300.31|30031.18|Default|
> > > > |28|blp-ar98|300.00|11455.66|DynSep(Ours)|neos-5195221-niemur|300.01|30000.54|DynSep(Ours)|
> > > > |29|blp-ar98|300.00|17007.64|Default|neos-5195221-niemur|300.33|30033.01|Default|
> > > > |30|blp-ic98|300.00|5253.91|DynSep(Ours)|neos-631710|300.09|0.00|DynSep(Ours)|
> > > > |31|blp-ic98|300.00|6227.12|Default|neos-631710|303.56|0.00|Default|
> > > > |32|bnatt400|300.29|30029.25|DynSep(Ours)|neos-662469|192.55|15312.80|DynSep(Ours)|
> > > > |33|bnatt400|300.08|30008.34|Default|neos-662469|299.14|25187.40|Default|
> > > > |34|bnatt500|300.00|30000.08|DynSep(Ours)|neos-787933|4.36|435.77|DynSep(Ours)|
> > > > |35|bnatt500|300.08|30007.55|Default|neos-787933|5.77|576.29|Default|

---

> > > > > ### Author Response · Authors · 2025-08-08
> > > > >
> > > > > *Table 4 (continued). A representative subset of MIPLIB 2017 instances is shown below for brevity—full results are available upon request.*
> > > > > ||instance_name|solving_time|primaldualintegral|Method|instance_name|solving_time|primaldualintegral|Method|
> > > > > |-|-|-|-|-|-|-|-|-|
> > > > > |36|bppc4-08|5.99|256.75|DynSep(Ours)|neos-827175|128.83|2135.73|DynSep(Ours)|
> > > > > |37|bppc4-08|300.00|2177.85|Default|neos-827175|171.48|3056.52|Default|
> > > > > |38|brazil3|300.00|30000.17|DynSep(Ours)|neos-848589|330.87|33087.08|DynSep(Ours)|
> > > > > |39|brazil3|300.03|30002.79|Default|neos-848589|311.46|11124.61|Default|
> > > > > |40|buildingenergy|300.03|30003.02|DynSep(Ours)|neos-860300|152.44|4393.38|DynSep(Ours)|
> > > > > |41|buildingenergy|300.03|30002.63|Default|neos-860300|152.19|4196.21|Default|
> > > > > |42|cbs-cta|93.76|9371.95|DynSep(Ours)|neos-873061|300.03|30003.18|DynSep(Ours)|
> > > > > |43|cbs-cta|300.10|30010.24|Default|neos-873061|300.03|30003.02|Default|
> > > > > |44|chromaticindex1024-7|300.12|30011.77|DynSep(Ours)|neos-911970|300.00|8842.15|DynSep(Ours)|
> > > > > |45|chromaticindex1024-7|300.15|30015.01|Default|neos-911970|300.00|11450.20|Default|
> > > > > |46|chromaticindex512-7|304.15|16647.67|DynSep(Ours)|neos-933966|300.15|28824.98|DynSep(Ours)|
> > > > > |47|chromaticindex512-7|301.59|25040.11|Default|neos-933966|300.09|29583.14|Default|
> > > > > |48|cmflsp50-24-8-8|300.00|30000.40|DynSep(Ours)|neos-937815|300.04|23919.16|DynSep(Ours)|
> > > > > |49|cmflsp50-24-8-8|300.26|30026.38|Default|neos-937815|300.18|23355.06|Default|
> > > > > |50|co-100|303.37|27088.52|DynSep(Ours)|neos-950242|300.00|30000.30|DynSep(Ours)|
> > > > > |51|co-100|305.11|27836.45|Default|neos-950242|300.01|29970.15|Default|
> > > > > |52|cod105|300.11|15312.43|DynSep(Ours)|neos-957323|368.33|17950.78|DynSep(Ours)|
> > > > > |53|cod105|300.04|16051.30|Default|neos-957323|301.42|17405.53|Default|
> > > > > |54|comp07-2idx|99.30|9929.90|DynSep(Ours)|neos-960392|305.71|30571.17|DynSep(Ours)|
> > > > > |55|comp07-2idx|300.12|30011.60|Default|neos-960392|303.98|30397.71|Default|
> > > > > |56|comp21-2idx|43.22|4091.79|DynSep(Ours)|neos17|16.88|681.30|DynSep(Ours)|
> > > > > |57|comp21-2idx|300.08|30004.57|Default|neos17|29.33|1376.34|Default|
> > > > > |58|cost266-UUE|300.00|3938.64|DynSep(Ours)|neos5|300.00|1800.15|DynSep(Ours)|
> > > > > |59|cost266-UUE|300.05|14030.49|Default|neos5|300.01|2036.73|Default|
> > > > > |60|cryptanalysiskb128n5obj14|300.07|30007.22|DynSep(Ours)|neos8|21.93|2015.16|DynSep(Ours)|
> > > > > |61|cryptanalysiskb128n5obj14|300.13|0.00|Default|neos8|24.73|2361.97|Default|
> > > > > |62|cryptanalysiskb128n5obj16|300.04|0.00|DynSep(Ours)|neos859080|1.26|126.35|DynSep(Ours)|
> > > > > |63|cryptanalysiskb128n5obj16|300.07|10003.04|Default|neos859080|1.77|176.57|Default|
> > > > > |64|csched007|27.22|2720.19|DynSep(Ours)|net12|300.00|23510.79|DynSep(Ours)|
> > > > > |65|csched007|300.01|30001.17|Default|net12|300.20|26938.98|Default|
> > > > > |66|csched008|300.07|30006.74|DynSep(Ours)|netdiversion|300.29|30028.80|DynSep(Ours)|
> > > > > |67|csched008|300.02|30001.81|Default|netdiversion|301.77|30176.71|Default|
> > > > > |68|cvs16r128-89|300.00|11471.81|DynSep(Ours)|nexp-150-20-8-5|19.32|1898.55|DynSep(Ours)|
> > > > > |69|cvs16r128-89|300.16|16150.14|Default|nexp-150-20-8-5|65.86|6465.64|Default|
> > > > > |70|dano3_3|300.00|5711.73|DynSep(Ours)|ns1116954|303.16|30316.21|DynSep(Ours)|
> > > > > |71|dano3_3|300.11|7975.16|Default|ns1116954|300.07|30007.18|Default|
> > > > > |72|dano3_5|300.00|5865.82|DynSep(Ours)|ns1208400|301.12|30111.61|DynSep(Ours)|
> > > > > |73|dano3_5|300.64|7590.88|Default|ns1208400|300.07|30006.55|Default|
> > > > > |74|decomp2|35.72|1475.39|DynSep(Ours)|ns1644855|300.03|29906.66|DynSep(Ours)|
> > > > > |75|decomp2|17.53|1572.22|Default|ns1644855|300.01|29891.60|Default|
> > > > > |76|drayage-100-23|49.37|3359.66|DynSep(Ours)|ns1760995|300.38|0.00|DynSep(Ours)|
> > > > > |77|drayage-100-23|67.45|5195.09|Default|ns1760995|300.53|0.00|Default|
> > > > > |78|drayage-25-23|300.00|2613.06|DynSep(Ours)|ns1830653|300.00|8709.04|DynSep(Ours)|
> > > > > |79|drayage-25-23|300.05|8676.32|Default|ns1830653|300.05|24382.60|Default|
> > > > > |80|dws008-01|300.00|26082.29|DynSep(Ours)|ns1952667|304.78|30478.39|DynSep(Ours)|
> > > > > |81|dws008-01|300.01|30000.88|Default|ns1952667|304.65|30464.90|Default|
> > > > > |82|eil33-2|255.11|4152.53|DynSep(Ours)|nu25-pr12|4.77|356.16|DynSep(Ours)|
> > > > > |83|eil33-2|300.02|4917.81|Default|nu25-pr12|5.33|356.76|Default|
> > > > > |84|eilA101-2|303.30|24543.44|DynSep(Ours)|nursesched-medium-hint03|302.30|30230.15|DynSep(Ours)|
> > > > > |85|eilA101-2|304.72|27328.43|Default|nursesched-medium-hint03|302.10|30182.35|Default|
> > > > > |86|enlight_hard|0.01|1.26|DynSep(Ours)|nursesched-sprint02|300.00|30000.18|DynSep(Ours)|
> > > > > |87|enlight_hard|0.01|1.46|Default|nursesched-sprint02|284.22|24857.05|Default|
> > > > > |88|ex10|148.85|0.00|DynSep(Ours)|nw04|143.64|6569.14|DynSep(Ours)|
> > > > > |89|ex10|203.21|0.00|Default|nw04|183.54|8531.62|Default|
> > > > > |90|ex9|49.94|0.00|DynSep(Ours)|opm2-z10-s4|301.22|27698.56|DynSep(Ours)|
> > > > > |91|ex9|54.95|0.00|Default|opm2-z10-s4|300.02|29157.60|Default|

---

> ### Comment · Area_Chair_HgE9 · 2025-08-04
>
> Dear Reviewer Fjx6,
>
> Thank you for clicking the “Mandatory Acknowledgement” button to confirm you’ve read the authors’ rebuttal. Please also leave comments as soon as possible on whether the authors have addressed your concerns and any follow-up questions in your mind, if any.
>
> Thank you for your continued contributions to the review process.
>
> Best,
> AC

---

### Official Review · Reviewer_E4cm · 2025-06-24

**Clarity:** 4
**Significance:** 3
**Originality:** 3
**Rating:** 4
**Confidence:** 4

**Summary:**

This paper proposes DynSep, a dynamic separator configuration method that models separator configuration in different rounds as a reinforcement learning task and tokenizes the incremental subgraphs with a transformer policy model to autoregressively predict when to half separation and which separators to activate at each round. Evaluated on synthetic and large-scale real-world MILP problems, DynSep substantially accelerates the solving time on easy and medium instances, as well as reducing integrality gap on hard instances.

**Questions:**

See weaknesses.

**Ethical Concerns:**

["NO or VERY MINOR ethics concerns only"]

**Final Justification:**

I'm overall in favor of accepting this paper, as it extends prior work from single-turn cutting-plane hyperparameter configuration to adaptive, multi-turn hyperparameter configuration, and it observes significant improvement from additional configurations. From the rebuttal, one weakness of the paper is that their current implementation cannot easily adapt to configure other sets of hyperparameters in the MILP solver, so the study of the proposed method's generalizability to different sets of MILP hyperparameters is bottlenecked by the engineering challenges. Also, the paper focuses on configuring the SCIP solver, and faces engineering challenges to adapt to other MILP solvers, so the generalizability across solvers is not tested. Given this, I would like to keep my initial score 4, borderline accept.

**Limitations:**

The authors explain the limitation of their methods in the conclusion section “the current implementation of dynsep lacks a lightweight decision model to retrain historical information across the global B&B tree”. I wonder if the authors can slightly expand in the conclusion to propose a few potential ways that can potentially address this limitation?

**Quality:**

3

**Strengths And Weaknesses:**

**Strengths**

1. The paper is clear and well-written.
2. The extension from static separator configuration to dynamic separator configuration is non-trivial. The authors handle this extension with incremental subgraph tokenization with blocked positional encoding for the transformer architecture, and these special treatments in the architecture design makes sense.
3. The authors show significant performance improvement over baselines on a variety of benchmarks, with detailed ablation study to justify the method’s design choices.

**Weaknesses**

1. I feel like the methods developed in this paper is not particularly specific to configuring cutting plane separators, and it should be able to apply to configure other parameters (e.g. branching, heuristics). Can the authors comment on the difficulty / flexibility of adapting their framework to configure other parameters, and potentially do small experiments to show the feasibility of this extension?
2. The authors set separator frequency as 10 and set the maximum separator round as T = 5. It is unclear how these two parameters (i.e. more or less frequent separator configurations) affect the performance of the method. The authors should perform some robustness studies in terms of varying these two parameters and comparing the performance.
3. Can the authors conduct experiments on gurobi (or other MILP solvers) and see if their algorithms can improve gurobi as well?

---

> ### Author Rebuttal · Authors · 2025-07-31
>
> We thank the reviewer for the positive and insightful comments. We respond to each comment as follows and sincerely hope that our rebuttal will properly address your concerns. If so, we would deeply appreciate it if you could raise your score. If not, please let us know your further concerns, and we will continue actively responding to your comments and improving our submission.
>
> **Q1. The difficulty / flexibility of adapting their framework to configure other parameters**.
>
> **A1.**
>
> Thank you for the insightful comment. **We claim that our proposed DynSep framework is inherently extensible to a broader set of solver hyperparameters beyond separators (e.g., branching, heuristics).** This is because DynSep is inherently a node-level adaptive configuration framework in the B&C tree.
>
> Compared to configure separator, adapting DynSep to branching or heuristics requires **nontrivial solver-specific engineering** because they expose different plugin/callback interfaces. Specifically, adapting to branching or heuristics would involve:
>
> 1. Implement analogous instrumentation via the BRANCH and HEURISTIC plugin callbacks, such as `BRANCHEXECLP` and `HEUREXEC`.
> 2. Augment the state representation with relevant branching/heuristic signals provided by the solver.
> 3. Adapt the policy network of DynSep: model the outputs of other parameters as a logistic‑normal distribution, followed by optional discretization to support both integer and continuous parameters. This enables flexible, differentiable control of arbitrary solver parameters.
> 4. Train the policy with the same PPO-based scheme.
>
> Due to time constraints, we have not yet migrated DynSep to branching or heuristic configuration, **but we have already extended it to additional cut-related parameters to demonstrate feasibility**. We conducted additional experiments on three critical hyperparameter groups, while retaining the original tuning mechanism for separator activation and max round.
>
> + **para group 1: Cut Depth / Aggressiveness (sepastore/age in SCIP).** Controlled by solver parameters `separating/cutagelimit` and `separating/poolfreq`.
> + **para group 2: Cut selection thresholds (e.g., efficacy vs. orthogonality).** Controlled by solver parameters `separating/minefficacy` and `cutselection/hybrid/minortho`.
> + **para group 3: Separation frequency per node.** Controlled by solver parameters `separating/cutagelimit` and `separating/poolfreq`.
>
> We provide the experimental results as follows.
>
> |Instance|Knapsack|Knapsack|Knapsack|Corlat|Corlat|Corlat|Supportcase|Supportcase|Supportcase|
> |-|-|-|-|-|-|-|-|-|-|
> |Metric|Time(s)|PD integral|PD gap|Time(s)|PD integral|PD gap|Time(s)|PD integral|PD gap|
> |DynSep (Ours)|0.52|9.71|0|**22.96**|2233.42|**0**|**132.50**|9212.24|7.86|
> |Para Group 1|0.65|9.01|0|47.36|4580.9| 4e+18  |141.54|8610.24|0.17|
> |Para Group 2|**0.36**|**7.91**|0|42.17|1971.18|0.01|167.64|8661.57|**0.16**|
> |Para Group 3|0.65|10.03|0|24.71|**1922.77**|0|141.09|**8362.53**|0.17|
>
> We evaluated our method on three benchmark datasets: **Knapsack**, **Corlat**, and **MIPLIB mixed supportcase**, measuring solving time (*Time*), primal–dual gap integral (*PD integral*), and primal–dual gap (*PD gap*). All three metrics are *lower-is-better*.
>
> We compared our original DynSep method with its extended versions incorporating three additional parameter groups. Results show that:
>
> 1. Across all datasets, **DynSep and its extended variants consistently outperform the default solver configuration** reported in the main paper.
>
> 2. The differences among the parameter groups are relatively small, yet certain combinations yield **further performance improvements** in both Time and PD integral.
>
> These results validate that DynSep can flexibly **extend to control a broader set of solver hyperparameters**. Furthermore, carefully chosen parameter combinations can yield **additional gains** in solving speed and convergence.
>
>
>
> **Q2. Robustness studies in terms of separator frequency and maximum separation round.**
>
> **A2.**
>
> We conducted robustness ablations over separator **frequency** (1, 5, 10, 20) and **maximum separation rounds** (3, 5, 10, 20) on three benchmarks (MIK, Corlat, and MIPLIB mixed Neos). Overall, **across almost all tested settings, DynSep outperforms the default configuration**, showing that the approach is reasonably stable to these hyperparameters.
>
> On each benchmark, we report three metrics: solving time (*Time*), primal–dual gap integral (*PD integral*), and primal–dual gap (*PD gap*). All three metrics are *lower-is-better*. Below is our key observations.
>
> + **Frequency:** Moderate frequency (e.g., 5&10) gives the better trade-off. That is, small frequency causes separators to fire excessively, incurring high cut-generation overhead, whereas large frequency reduces opportunities for timely dual-bound tightening.
>
> |Instance|MIK|MIK|MIK|Corlat|Corlat|Corlat|Neos|Neos|Neos|
> |-|-|-|-|-|-|-|-|-|-|
> |Metric|Time(s)|PD integral|PD gap|Time(s)|PD integral|PD gap|Time(s)|PD integral|PD gap|
> |Default|16.65|82.80|0.0|111.55|10573.14|2.73e+19|282.98|18500.5|2.5e+19|
> |Freq=1|12.74|124.22|0.0|53.77|4582.15|4e+18|252.12|8756.3|2.5e+19|
> |Freq=5|13.04|**86.55**|0.0|38.44|3513.13|0.0|243.91|9248.52|2.5e+19|
> |Freq=10 (Ours)|**10.99**|134.15|0.0|**22.96**|**2233.42**|**0.0**|**235.19**|**8511.58**|2.5e+19|
> |Freq=20|12.83|213.15|0.0|49.14|4658.09|0.0|261.86|8789.54|2.5e+19|
>
> + **Maximum separation rounds:** Setting this value too low produces weak cuts and degrades performance, while setting it excessively high increases the computational cost of cut generation. Although this parameter shows some dataset sensitivity, MaxRound=5 is empirically near-optimal in our tests.
>
> |Instance|MIK|MIK|MIK|Corlat|Corlat|Corlat|Neos|Neos|Neos|
> |-|-|-|-|-|-|-|-|-|-|
> |Metric|Time(s)|PD integral|PD gap|Time(s)|PD integral|PD gap|Time(s)|PD integral|PD gap|
> |Default|16.65| 82.80       | 0.0    | 111.55    | 10573.14    | 2.73e+19 | 282.98     | 18500.5     |2.5e+19|
> |MaxRound=3|13.63|150.19|0.0|107.26|8965.26|2.4e+19|239.22|8581.5|2.5e+19|
> |MaxRound=5 (Ours)|**10.99**| **134.15**  | 0.0    | **22.96** | **2233.42** | 0.0      | **235.19** | **8511.58** |2.5e+19|
> |MaxRound=10|17.6|182.71|0.0|40.45|3717.64|2e+18|249.41|8846.91|2.5e+19|
> |MaxRound=20|16.2|169.8|0.0|36.71|3609.11|0.0|252.59|10945.82|2.5e+19|
>
>
>
> **Q3. Can the authors conduct experiments on gurobi (or other MILP solvers)?**
>
> **A3.**
>
> We appreciate the reviewer’s suggestion to apply our method to ohter MILP solvers like Gurobi.  Although our current experiments are conducted in SCIP, **translating our approach for separator-activation tuning to Gurobi is technically feasible.** Below, we outline the feasibility and limitations of such an implementation.
>
> + **Feasibility.** Gurobi provides separators like `CliqueCuts`, `FlowCoverCuts`, `MIRCuts`, `StrongCGCuts`, etc., each accepting a configuration parameter in {0, 1, 2} (or –1 for automatic). 0 disables the cut separator, 1 enables it moderately, and 2 enables it aggressively—precisely matching DynSep’s configuration schema. These parameters can be set statically via `model.setParam("CliqueCuts", value)` and can also be changed dynamically at the node level via `cbSetIntParam(...)` within a `where == GRB.Callback.MIPNODE` callback. By using these interfaces, our separator configuration method can be ported to Gurobi fairly directly.
> + **Limitation.** Gurobi’s callback mechanism only invokes user code once per node for separators, which prevents us from adjusting parameters at a finer, per-separation-round granularity. However, node-level tuning remains effective for most practical scenarios.
>
> Given the preceding discussion, the adaptation **requires substantial solver-specific engineering** to rewire state collection, accommodate Gurobi’s callback semantics, and translate our dynamic decision logic to its API. Due to time constraints, we have not yet completed the Gurobi experiments. Nevertheless, we remain confident that DynSep’s separator-configuration strategy can be properly implemented in Gurobi. We will **prioritize these tests** and include preliminary Gurobi results once feasible.
>
>
>
> **Q4. A lightweight decision model to retrain historical information across the global B&B tree.**
>
> **A4.**
>
> Here we provide more discussion on the potential ways that can address the mentioned limitation. The current DynSep only reasons over the **local, per-node** incremental graph. To capture **global, historical dynamics** across the entire branch-and-bound tree, one could:
>
> - **Maintain a tree-wide memory** of separator performance and variable bound changes (e.g. via a lightweight recurrent summary or graph coarsening).
> - **Aggregate cut efficacy statistics** at each depth level (or over recent sibling nodes) to inform decisions in new nodes—trading off memory overhead against improved long-term planning.
>
> While logging every cut and bound update could incur runtime overhead, designing rational sampling techniques could keep this overhead minimal. Embedding such a lightweight global module would allow DynSep to adapt not only to per-round shifts but also to the evolving solver trajectory across the entire search tree.

---

> > ### Comment · Reviewer_E4cm · 2025-08-05
> >
> > Thanks for the response. I do not have any follow up questions and would like to keep my score as I think the adaptive cutting plane parameter configuration is a valuable contribution to the community given prior works mainly focus on single turn configuration. I understand the engineering difficulty, but I encourage the authors to explore adaptive configuration for other MILP parameters and potentially other MILP solvers to further strengthen the work.

---

> > > ### Author Response · Authors · 2025-08-09
> > >
> > > Thank you very much for your encouraging feedback. We are glad to know that you view the adaptive cutting-plane parameter configuration as a valuable contribution. We fully appreciate your insightful suggestion regarding extending adaptive configuration to other MILP parameters and solvers, and we look forward to exploring these directions in our future work.
> > >
> > > Notably, **we have conducted new experimental trials exploring an enlarged MILP parameter space.** Although these additional parameters focus on the separator module due to engineering complexity, we hope that the results could demonstrate **the extensibility of our method across a broader parameter space**.
> > >
> > > Specially, we expand our controlled separator parameter set **from a total number of $46$ to $144$**, more than tripling its original size.  The expanded parameter space includes shared or separator-specific parameters and covers **various numeric types** (int, float, bool).
> > >
> > > The results in Table 1 show that our method (DynSep) with an enlarged parameter space delivers **substantial performance gains** over the default solver configuration.
> > >
> > > *Table 1. Evaluation for the default setting (Default) and our configuration method with an enlarged parameter space (DynSep-expended).*
> > >
> > > |Instance|Knapsack|Knapsack|Corlat|Corlat|neos|neos|
> > > |-|-|-|-|-|-|-|
> > > |Metric|Time(s)|PD integral|Time(s)|PD integral|Time(s)|PD integral|
> > > |Default|2.01|18.6|111.55|10573.14|282.98|18500.5|
> > > |DynSep-expended (ours)|**0.57**|**10.46**|**56.02**|**5161.01**|**249.43**|**8602.66**|
> > >
> > >
> > > Compared to traditional automated configuration tools such as ParamILS [1]—typically targeting **81 CPLEX parameters**—and SMAC [2]—focusing on **76 parameters**—as well as recent machine learning approaches like Deep Metric Learning [3], which tune **17 parameters**, our method supports **a larger and extensible parameter space**. It employs a unified and scalable framework capable of handling both shared and separator-specific parameters across multiple data types (`int`, `float`, `bool`).
> > >
> > > The comprehensive list of newly included parameters is detailed in **Table 2**. Note that there are $142$ separator parameters in the table. Adding two global parameters—`separating/maxrounds` and `separating/maxroundsroot`—makes a total of $144$ parameters.
> > >
> > > *Table 2. List of separator parameters controlled by our configuration policy.*
> > >
> > > |Separator|Shared Params|Specific Params|Separator|Shared Params|Specific Params|
> > > |-|-|-|-|-|-|
> > > |closecuts|freq, priority, maxbounddist, delay, expbackoff|sepacombvalue, closethres|cgmip|freq, priority, maxbounddist, delay, expbackoff|minnodelimit, maxnodelimit|
> > > |disjunctive|freq, priority, maxbounddist, delay, expbackoff|maxinvcuts, maxrank|strongcg|freq, priority, maxbounddist, delay, expbackoff|/|
> > > |minor|freq, priority, maxbounddist, delay, expbackoff|maxminorsconst,  maxminorsfac|aggregation|freq, priority, maxbounddist, delay, expbackoff|maxaggrs, maxsepacuts|
> > > |mixing|freq, priority, maxbounddist, delay, expbackoff|uselocalbounds, iscutsonints|clique|freq, priority, maxbounddist, delay, expbackoff|maxsepacuts, scaleval|
> > > |rlt|freq, priority, maxbounddist, delay, expbackoff|maxusedvars, goodscore|zerohalf|freq, priority, maxbounddist, delay, expbackoff|maxsepacuts, minviol|
> > > |interminor|freq, priority, maxbounddist, delay, expbackoff|mincutviol|mcf|freq, priority, maxbounddist, delay, expbackoff|maxsepacuts, nclusters|
> > > |convexproj|freq, priority, maxbounddist, delay, expbackoff|nlpiterlimit|eccuts|freq, priority, maxbounddist, delay, expbackoff|maxsepacuts, minviolation|
> > > |gauge|freq, priority, maxbounddist, delay, expbackoff|nlpiterlimit|oddcycle|freq, priority, maxbounddist, delay, expbackoff|maxsepacuts, scalingfactor|
> > > |impliedbounds|freq, priority, maxbounddist, delay, expbackoff|usetwosizecliques|flowcover|freq, priority, maxbounddist, delay, expbackoff|/|
> > > |intobj|freq, priority, maxbounddist, delay, expbackoff|/|cmir|freq, priority, maxbounddist, delay, expbackoff|/|
> > > |gomory|freq, priority, maxbounddist, delay, expbackoff|maxsepacuts, away|rapidlearning|freq, priority, maxbounddist, delay, expbackoff|maxcalls, maxnodes|
> > >
> > >
> > > ---
> > > [1] Hutter, Frank, et al. "ParamILS: an automatic algorithm configuration framework." Journal of artificial intelligence research 36 (2009).
> > >
> > > [2] Hutter, Frank, et al.  "Sequential model-based optimization for general algorithm configuration." International conference on learning and intelligent optimization (2011).
> > >
> > > [3] Hosny, Abdelrahman, and Sherief Reda. "Automatic MILP solver configuration by learning problem similarities." Annals of Operations Research (2024).

---

> ### Comment · Area_Chair_HgE9 · 2025-08-04
>
> Dear Reviewer E4cm,
>
> Thank you for clicking the “Mandatory Acknowledgement” button to confirm you’ve read the authors’ rebuttal. Please also leave comments as soon as possible on whether the authors have addressed your concerns and any follow-up questions in your mind, if any.
>
> Thank you for your continued contributions to the review process.
>
> Best,
> AC

---

### Official Review · Reviewer_rt7i · 2025-06-29

**Clarity:** 4
**Significance:** 4
**Originality:** 3
**Rating:** 4
**Confidence:** 4

**Summary:**

This paper introduces DynSep, a dynamic separator configuration method for cutting plane selection in Mixed-Integer Linear Programming (MILP) solvers. DynSep formulates the round-wise activation and deactivation of cutting plane separators as a reinforcement learning (RL) problem. By modeling the evolving MILP instance as an incremental triplet graph, DynSep uses a GCN encoder and a decoder-only Transformer to autoregressively predict both when to halt separation and which separators to activate at each round. Experiments on synthetic and large-scale real-world MILP benchmarks demonstrate that DynSep significantly accelerates solving time (by 64% on average for easy/medium datasets) and reduces the primal-dual gap integral (by 16% on challenging datasets) compared to both human-designed and learning-based baselines. The method also shows strong generalization to much larger MILP instances than those used in training.

**Questions:**

- Could you add comparisons with these mainstream commercial solvers (such as Gurobi or CPLEX) so I can better understand the current level of DynSep’s capabilities in an industrial context?
- How does DynSep handle the addition of new, user-defined, or more advanced separators beyond SCIP’s built-in set? What is required to support extensibility?

**Ethical Concerns:**

["NO or VERY MINOR ethics concerns only"]

**Final Justification:**

Thank you for your detailed response. It has clarified the questions I raised and resolved my earlier concerns. Given this, I am confident in the contributions of the work and maintain my positive score of 4.

**Limitations:**

Yes

**Quality:**

3

**Strengths And Weaknesses:**

Strengths:
- DynSep delivers substantial improvements over both human-designed and ML baselines across a wide range of MILP benchmarks, including large-scale industrial and public datasets.
- The method generalizes to larger instances and unseen problem sizes, a key requirement for practical MILP solving.
- Ablation studies confirm the necessity of each component, and visualization of learned separator policies aligns with domain insights.

Weaknesses:
- All experiments are conducted within SCIP; comparisons to commercial solvers (e.g., Gurobi, CPLEX) or other solver backends are missing, which limits claims of generality and practical impact.
- The approach configures only the built-in separators in SCIP, and does not explore integration with advanced or user-defined cutting plane families, potentially limiting extensibility.

---

> ### Author Rebuttal · Authors · 2025-07-31
>
> We thank the reviewer for the positive and insightful comments. We respond to each comment as follows and sincerely hope that our rebuttal will properly address your concerns. If so, we would deeply appreciate it if you could raise your score. If not, please let us know your further concerns, and we will continue actively responding to your comments and improving our submission.
>
> **Q1. Extension to commercial solvers. (e.g., Gurobi, CPLEX)**
>
> **A1.**
>
> We appreciate the reviewer’s suggestion to apply our method to commercial solvers like Gurobi or CPLEX. Although our current experiments are conducted in SCIP, **translating our approach for separator-activation tuning to Gurobi is technically feasible.** Below, we outline the feasibility and limitations of such an implementation.
>
> + **Feasibility.** Gurobi provides separators like `CliqueCuts`, `FlowCoverCuts`, `MIRCuts`, `StrongCGCuts`, etc., each accepting a configuration parameter in {0, 1, 2} (or –1 for automatic). 0 disables the cut separator, 1 enables it moderately, and 2 enables it aggressively—precisely matching DynSep’s configuration schema. These parameters can be set statically via `model.setParam("CliqueCuts", value)` and can also be changed dynamically at the node level via `cbSetIntParam(...)` within a `where == GRB.Callback.MIPNODE` callback. By using these interfaces, our separator configuration method can be ported to Gurobi fairly directly.
> + **Limitation.** Gurobi’s callback mechanism only invokes user code once per node for separators, which prevents us from adjusting parameters at a finer, per-separation-round granularity. However, node-level tuning remains effective for most practical scenarios.
>
> Given the preceding discussion, the adaptation **requires substantial solver-specific engineering** to rewire state collection, accommodate Gurobi’s callback semantics, and translate our dynamic decision logic to its API. Due to time constraints, we have not yet completed the Gurobi experiments. Nevertheless, we remain confident that DynSep’s separator-configuration strategy can be properly implemented in Gurobi. We will **prioritize these tests** and include preliminary Gurobi results once feasible.
>
>
>
> **Q2. Integration with advanced or user-defined cutting plane families.**
>
> **A2.**
>
> SCIP exposes a **plugin API** for **user-defined separators**: once a custom separator is registered in SCIP, it appears in the separator nodes of our triplet graph alongside the other built-in separators. DynSep treats every separator node uniformly—built-in or custom—so any new separator simply becomes an additional token whose activation status can be tuned. In fact, our own “configuration separator” is implemented via this same API, and is invoked ahead of all other separators to adjust their parameters dynamically.

---

> > ### Comment · Reviewer_rt7i · 2025-08-05
> >
> > Thank you for your detailed explanation. I understand that due to time constraints, the relevant experiments have not yet been completed. However, adding these experiments in the future would greatly enhance the impact and strength of your paper. Overall, your response has addressed my concerns.

---

> > > ### Author Response · Authors · 2025-08-09
> > >
> > > Thank you for your thoughtful feedback and understanding. We’re pleased that our responses addressed your concerns, and we fully appreciate the importance of the additional experiments. We are committed to conducting them and will include the results in future updates.

---

> ### Comment · Area_Chair_HgE9 · 2025-08-04
>
> Dear Reviewer rt7i,
>
> It appears that you have not yet replied to the authors' rebuttal for Submission 22899, and the discussion period (July 31–August 6) is halfway over. Please get involved as soon as you can to give us time for a fruitful conversation.
>
> In particular, do you feel the authors have addressed your concern regarding the applicability and the impact of the proposed method?
>
> Lastly, don’t forget to click the “Mandatory Acknowledgement” button to confirm you’ve participated.
>
> Thank you for your continued contributions to the review process.
>
> Best,
> AC

---

### Official Review · Reviewer_ZkPt · 2025-07-03

**Clarity:** 3
**Significance:** 4
**Originality:** 4
**Rating:** 5
**Confidence:** 4

**Summary:**

This work proposes DynSep, a reinforcement learning–based method to dynamically control how cutting planes are added during the Branch-and-Cut process in solving MILP optimization. The authors claim that instead of relying on fixed or manually tuned separator settings, DynSep learns to decide both which separators to activate in each separation round and when to stop adding more cuts at each node while adapting these choices as the problem evolves across different rounds and nodes. To keep it efficient, DynSep only looks at the incremental subgraphs added in each round, then using GCN and a decoder-only Transformer to track the context from earlier rounds and predict the next separator configuration. The authors claim this design helps the solver avoid unnecessary separation rounds, capture useful separator interaction patterns automatically, and improve both speed and solution quality. In experiments, DynSep reduced average solve times by up to 64% and improved solution quality on harder benchmarks, while also generalizing well to larger, unseen MILP instances.

**Questions:**

- It is unclear how the GCN encoder is trained, is it included in the transformer training?
- How might DynSep be adapted or extended to handle nonlinear or mixed-integer nonlinear programming (MINLP) problems?
- How does the blocked positional encoding impact performance?
- Could alternative architectures such as graph transformers improve the modeling of separator dynamics?
- Is the training based on the same types of MILPs in the test benchmarks? How well does it generalize to unseen types MILPs?

**Ethical Concerns:**

["NO or VERY MINOR ethics concerns only"]

**Final Justification:**

The authors have addressed my concerns on clarity of presentation and generalizability of the proposed method across problem sets. Therefore I raise my original rating of 4 to 5.

**Limitations:**

- The method is limited to MILP problems and does not cover nonlinear formulations that frequently occur in real-world applications.
- The method maybe limited to the types of MILPs seen during training, and generaliability to unseen types of MILPs is unclear.

**Paper Formatting Concerns:**

Please ensure consistency by using "decoder-only Transformer," as stated in Experiment 2: Ablation Study, instead of "encoder-only Transformer."

**Quality:**

4

**Strengths And Weaknesses:**

## Strengths

- The main strength of this work is that DynSep is an adaptive method that learns not just which separators to use but also when to stop adding cuts, adjusting dynamically as the Branch-and-Cut process evolves.
- Its design is efficient, since it processes only new cuts each round and uses a Transformer decoder to capture temporal context.
- The proposed DynSep has demonstrated consistent and significant improvements in runtime and/or primal-dual gaps on various benchmarks, and datasets. The ablation study also helps with interpretability.
- Built on the open-source SCIP solver, DynSep seems to be also practical and reproducible.

## Weaknesses
- There is a lack of clarity in the formulation of the solutions. For example, it is unclear the difference between cutting planes and separators until the very end of experiments. As a key concept in the solution and graph model, the separator and how it generates cutting planes are not explained, and it is unclear if the set of separator nodes remain the same for each triplet graph. It is also unclear how separator nodes are represented in the triplet graph, do they have node features or simply represented by node identity.
- The transformer generates an action based on the history of token blocks. But the tokenization or encoding of the vectorized action is unclear in the paper. Is the action a sequence of tokens or a single token? If an action is a token, is the token space the same as the action space?
- Within each block, node embeddings have identical positional encoding. How about graph positional encoding such as spectral PE from graph transformer literature? If the authors deliberately choose not to use graph PE, some explanation would be good.
- Many notations are used before their first introduction, such as PD integral, MDP, Node and Round-wise dynamic configurations.
- It is unclear about the generalizability of the DynSep due to the lack of training information. Is it trained on the same types of NP-hard problems in the benchmarks? If so, how well does it generalize to other unseen benchmarks? This could be a major limitation and need more clarification.

---

> ### Author Rebuttal · Authors · 2025-07-30
>
> We thank the reviewer for the positive and insightful comments. We respond to each comment as follows and sincerely hope that our rebuttal will properly address your concerns. If so, we would deeply appreciate it if you could raise your score. If not, please let us know your further concerns, and we will continue actively responding to your comments and improving our submission.
>
> **Q1. Lack of clarity in the formulation of the solution.**
>
> **A1.**
>
> We apologize for the confusion. Below, we provide the detailed specifications as suggested, and we will include them in the revised manuscript.
>
> 1. **Concept of separators.** Separators are core components of a mixed‑integer programming (MIP) solver. Each separator implements a dedicated cutting‑plane algorithm to generate a family of valid inequalities (i.e., cuts) to tighten the LP relaxation. For example, the *Gomory cut separator* derives Gomory cuts from the current simplex tableau to remove fractional solutions.
> 2. **Difference between cutting planes and separators.** Cutting planes are valid inequalities that tighten the LP relaxation, which exclude fractional solutions while preserving all feasible integer solutions. Separators are built-in algorithms that generate these cutting planes.
> 3. **Separator nodes.** In our triplet graph, each separator is represented as a dedicated node. **Separator node features** comprise a one-hot encoding of its type, plus the separator's performance statistics (e.g., execution time, number of applied cuts, and dual-bound improvement) to capture historical efficacy. Full definitions of these features appear in Table 4 of the appendix. **Notably, the set of separator nodes is fixed** (corresponding to 22 SCIP’s built‑in separators) and remains the same for every instance, while their feature values are updated dynamically at each decision step.
>
> **Q2. How are actions tokenized/encoded for the transformer?**
>
> **A2.**
>
> Each action $a_t = [m_t, \eta_{t,1},\dots, \eta_{t,K}]\in\mathbb{N}^{K+1}$ could be viewed as a block of tokens, consisting of one token for the maximum round threshold $m_t$ and $K$ tokens $\eta_{t,1},\dots, \eta_{t,K}$ for $K$ separators' activation status. Below, we describe the tokenized formulation of the action.
>
> + **Maximum‑round token** $m_t$: drawn from an $r_{\max}$‑dimensional categorical distribution (its token space is $ \\{1,\dots,r_{\max} \\} $).
> + **Separator‑activation tokens** $\eta_{t,i}$: for each separator, one token drawn from a 3‑dimensional categorical distribution over $ \\{-1,0,1\\} $ indicating three distinct activation statuses.
>
> **Notably, we clarify that we do not feed the sampled action $a_t$ back into the transformer.** Instead, we execute the action $a_t$ in the solver environment and obtain the next incremental state $\Delta s_{t+1}$ (i.e., the incremental graph). We input $\Delta s_{t+1}$ as the subsequent token block into the transformer. This yields an autoregressive loop for both the training and reference phases--each next-step prediction $a_t$ conditions on the entire history of past incremental states $\\{\Delta s_{1:t}\\}$ (equal to $s_t$). Such procedure matches the modeling of standard online RL policy $a_t=\pi(s_t)$, where the policy $\pi$ depends only on the current state $s_t$, not on prior actions.
>
>
>
> **Q3. Why not use graph-based positional encodings (PEs) (e.g., spectral PE)?**
>
> **A3.**
>
> We acknowledge that, since all separator nodes are fully connected to variable and constraint nodes, they share the same graph topology, thus receiving identical spectral PE. **However, we deliberately opt for blocked PE for two main reasons**:
>
> 1. **Computational Efficiency:**
>    Computing spectral or other graph‑based PEs requires eigendecomposition or additional operations on the graph Laplacian (or related matrices), which **scales poorly on large graphs**. In contrast, our blocked PE leverages the prior knowledge that separator nodes have no inherent sequence order and is aligned with minimal overhead.
> 2. **Dynamic Temporal Modeling:**
>    Graph‑based PEs encode static topological similarity and **cannot capture the evolving, time‑series nature** of the dynamic separator configuration. In contrast, our blocked PE preserves the chronological order of each incremental subgraph while treating separator nodes as an unordered set. This ensures the model focuses on the temporal progression of separator activations rather than a fixed structural pattern.
>
>
> **Q4. Several notations appear before their first introduction.**
>
> **A4.**
>
> We appreciate this feedback. We’ll move those definitions earlier for clarity.
>
>
> **Q5. The generalization of DynSep.**
>
> **A5.**
>
> Thank you for raising this point. In our evaluation,  DynSep policies are trained and tested within the same problem class.
>
> + **To evaluate the across-domain generalization**, we train our policy on one problem family and apply the learned policy to **unseen problem families**. Specifically, we train 3 separate DynSep policies (on Knapsack, MIK, and Supportcase) and evaluate each of them across all nine benchmark families used in our manuscript. The table below lists the results, where we report solving time for easy and medium datasets, while reporting PD integral for hard datasets.
>
>   As shown in the table, policies trained on one problem type **yield improvements over SCIP’s default in most unseen benchmarks**, indicating effective transfer of our learned configuration strategy across NP-hard families.
>
> ||SetCover|MIS|Knapsack|CORLAT|MIK| Anonymous   |LB|Neos|Supportcase|
> |-|-|-|-|-|-|-|-|-|-|
> ||Time (s)|Time (s)|Time (s)|Time (s)|Time (s)|PD integral|PD integral|PD integral|PD integral|
> |Default|5.24|30.4|2.01|111.55|16.65|27069|15187.19|18500.5|21561.09|
> |Train on Knapsack|**2.02**|0.76|/|27.56| **12.48** |19183.3|**4583.45**|**13343.48**|13216.93|
> |Train on MIK|7.74|5.21|1.2|34.54|/|20715.25|5559.96|13366.13|**11583.1**|
> |Train on supportcase|2.21|**0.67**|**0.78**|**20.94**|163.6|**17922.89**|9421|13639.73|/|
>
> + To further demonstrate generalization, **we also evaluated on real-world datasets**. Although they come from the same application domain, geographic, temporal, and related variations induce significant distribution shift, thus cross-dataset performance could serve as **a proxy for transfer to unseen problems**.
>
> As shown in the table below, our method (**DynSep**) delivers **marked performance gains** on two real‑world datasets (MIRP and SRPN) from the Distributional MIPLIB benchmark [1], providing evidence of its ability to generalize to unseen, distribution-shifted real-world instances.
>
> |Instance|mirp|mirp|mirp|srpn|srpn|srpn|
> |-|-|-|-|-|-|-|
> |Metric|Time(s)|PD integral|PD gap|Time(s)|PD integral|PD gap|
> |Default|580.98|52362.46|5.38e+19|332.04|11687.01|0.21|
> |DynSep(Ours)|**482.13**|**30838.39**|**1.38**|**294.77**|**7581.16**|**0.1**|
>
>
> **Q6. How the GCN encoder is trained**?
>
> **A6.**
>
> We use separate decoder-only Transformers for actor and critic. The GCN encoder is updated only during critic training (backpropagated with the critic loss); it is frozen during actor updates.
>
>
> **Q7. Can DynSep handle nonlinear or MINLP problems?**
>
> **A7.**
>
> Thank you for the insightful comment. We believe **DynSep can naturally extend to MINLPs** because SCIP’s unified Constraint Integer Programming (CIP) framework handles both MILPs and MINLPs within the same branch‑cut‑and‑price architecture. When solving MINLPs, SCIP automatically applies nonlinear handlers and MINLP-specific separators. like *RLT cut separator* and *minor cut separator*, to strengthen relaxations via valid cuts.
>
> By incorporating MINLP‑specific separators into the action space, our instance‑aware configuration method could extend seamlessly to nonlinear problems. Meanwhile, we could enrich our state representation to capture nonlinear structure following the existing approach [2]. In other words, the core decision‑making paradigm—choosing which separators to fire at each node and when to stop—remains valid for both MILP and MINLP, making this extension both straightforward and feasible.
>
>
> **Q8. How does the blocked positional encoding impact performance?**
>
> **Q9. Would alternative architectures like graph transformers improve separator modeling?**
>
> **A8. & A9.**
>
> We merged the results for **Q8** and **Q9** into a single table for compactness and easier comparison.
>
> + **w/o BlockPE (for Q8):** This row show the results that replace the blocked positional encoding with a standard token-level positional encoding. Its **degraded performance relative to DynSep** indicates that masking intra-block order information (i.e., using BlockPE) benefits training and improves final performance.
> + **GT (for Q9):**  This row show the results that replace the encoder’s GCN with a custom bipartite **graph transformer (GT)**: a multi-head TransformerConv for edge-aware message passing, followed by residual-connected LayerNorm and a two-layer feed-forward block. **"GT" shows no consistent improvement over DynSep**, which may be due to increased inference overhead or the need for finer stability/tuning to realize gains from the transformer-style aggregation.
>
> |Instance|Knapsack|Knapsack|Corlat|Corlat|neos|neos|supportcase|supportcase|
> |-|-|-|-|-|-|-|-|-|
> |Metric|Time(s)|PD integral|Time(s)|PD integral|Time(s)|PD integral|Time(s)|PD integral|
> |w/o BlockPE|0.93|11.12|34.4|2870.01|242.39|**8291.49**|153.29|12715.64|
> |GT|0.71|11.02|46.26|4563.26|243.52|12134.57|150.38|**9157.61**|
> |DynSep|**0.52**|**9.71**|**22.96**|**2233.42**|**235.19**|8511.58|**132.50**|9212.24|
>
>
> **Q10. Typo in Ablation Study.**
>
> **A10.**
>
> Thank you for pointing this out. We will keep all statements consistent.
>
>
> [1] Huang, et al. "Distributional MIPLIB: a multi-domain library for advancing ml-guided milp methods." arXiv 2024.
>
> [2] Tang, Bo, Elias Boutros Khalil, and Jan Drgona. "Learning to Optimize for Mixed-Integer Nonlinear Programming." (2024).

---

> > ### Comment · Reviewer_ZkPt · 2025-08-05
> >
> > Thanks for the authors for their detailed rebuttal on the issues I raised.
> >
> > The authors' response improved the clarity of the work. However, my major concern on the generalizability of the proposed method across different problem sets and distributional shift is only partially addressed. The authors show that it is capable of generalize across problem types, but users still need to pick different trained models for their problems to be able to obtain the performance gain. A truly generalizable approach should be not dependent on such handpicked ML models. Therefore the contribution is still limited. Unless the authors can justify the real world merit of such manually selecting ML models for particular problem sets, I would maintain my current rating.

---

> ### Comment · Area_Chair_HgE9 · 2025-08-04
>
> Dear Reviewer ZkPt,
>
> It appears that you have not yet replied to the authors' rebuttal for Submission 22899, and the discussion period (July 31–August 6) is halfway over. Please get involved as soon as you can to give us time for a fruitful conversation.
>
> In particular, please share whether their responses adequately address your concerns, especially regarding the technical clarity and the generalization of the proposed method.
>
> Lastly, don’t forget to click the “Mandatory Acknowledgement” button to confirm you’ve participated.
>
> Thank you for your continued contributions to the review process.
>
> Best,
> AC

---

> ### Author Response · Authors · 2025-08-08
>
> Thank you for raising the concerns about our generalization study. Below, we’ve **improved our generalization study** and **detailed the real-world merits of domain-specific models**.
>
> + **Improved generalization study for our method.**
>
> We select $168$ diverse instances from the MIPLIB 2017 benchmark [1]  as our training set and learn a separator configuration policy on these instances. Notably, because the MIPLIB 2017 instances cover **a wide variety of problem types and mixed-scenario structures** [1], this learned policy could serve as **a general configuration model.** We then evaluate it—without any additional tuning—on four unseen, domain-specific datasets: Corlat, Load Balancing (LB), Maritime Inventory Routing Problem (MIRP), and Seismic‑Resilient Pipe Network Planning (SRPN).
>
>   As shown in Table 1, the general configuration model **consistently outperforms** the solver’s default settings in both solve time and convergence behavior, **demonstrating good generalizability of our method.**
>
> *Table 1. Generalization performance of our DynSep model trained on MIPLIB 2017, evaluated on four unseen MILP scenarios. (300-second time limit for Corlat & LB; 600-second for MIRP & SRPN)*
>
>
> ||Corlat|Corlat|LB|LB|MIRP|MIRP|SRPN|SRPN|
> |-|-|-|-|-|-|-|-|-|
> ||Time (s)|PD integral|Time (s)|PD integral|Time (s)|PD integral|Time (s)|PD integral|
> |Default|111.55|10573.14|300.14|15187.19|580.98|52362.46|332.04|11687.01|
> |DynSep (ours) trained on miplib2017|**46.05**|**4289.33**|**300.08**|**4792.71**|**487.59**|**33193.25**|**288.48**|**7582.60**|
>
> ---
>
> + **Real-world merits of domain-specific models.**
>
> Although a domain-general model reduces the workload of manually picking or retraining for each new scenario, **the structural diversity of MILP instances makes domain-specific models considerably more competitive**—especially in computationally intensive industrial settings, where these domain-specific models play a pivotal role in accelerating the solving process and enhancing solution quality within a time limit for particular problem sets. For example,
>
>   + By applying Gurobi’s parameter-optimization model to 97 quadratic knapsack instances from Beasley’s OR-Library, the solver highlights five key parameters and achieves up to 7‑fold runtime improvements. [2]
>   + SMAC, a configuration method applied to CPLEX’s 76 expert parameters,  delivers up to 50‑fold runtime improvements over default settings on domain‑specific tasks such as combinatorial auction and knapsack problems.[3]
>
>   + Configuration approaches such as ParamILS [4] and irace [5] have shown significant, problem-specific speed-ups on real-world wildlife-corridor design problems.
>
>
>    These case studies illustrate that **scenario-aware tuning could incur minimal overhead while delivering critical latency reductions** in real-world scenarios.
>
>
> + **Clarification of our original generation results.**
>
> In our across-domain evaluation for the original rebuttal, **each of the three trained policies outperforms** the default configuration on the vast majority of problem families (see Table 2).  As a result, any of the three trained policies can individually deliver **reliable speed-ups** on all tested datasets, eliminating the need for the model-selection step.
>
> *Table 2. Relative improvement over the default solver settings in solving time (Time) or primal-dual integral (PD integral), for three models trained on different problem families (Knapsack, MIK, Supportcase).*
>
> ||SetCover|MIS|Knapsack|Corlat|MIK|Anonymous|LB|Neos|Supportcase|AVERAGE Improvements|
> |-|-|-|-|-|-|-|-|-|-|-|
> ||Time (%)|Time (%)|Time (%)|Time (%)|Time (%)|PD integral (%)|PD integral (%)|PD integral (%)|PD integral (%)||
> |Train on Knapsack|+61.45|+97.50|/|+75.29|+25.00|+29.13|+69.82|+27.88|+38.70|**+53.10%**|
> |Train on MIK|-32.30|+82.86|+40.30|+68.14|/|+23.47|+63.39|+27.75|+46.28|**+39.99%**|
> |Train on Supportcase|+57.82|+97.80|+61.19|+81.23|-89.82|+33.79|+37.97|+26.27|/|**+38.25%**|
>
>
> ****
>
> [1] Gleixner, et al. "MIPLIB 2017: data-driven compilation of the 6th mixed-integer programming library." Mathematical Programming Computation(2021).
>
> [2] Rando, D., et al. "The importance of fine-tuning Gurobi parameters when solving quadratic knapsack problems: A guide for OR practitioners. Pure and New Mathematics in AI (2024).
>
> [3] Hutter, Frank, Holger H. Hoos, and Kevin Leyton-Brown. "Sequential model-based optimization for general algorithm configuration." International conference on learning and intelligent optimization (2011).
>
> [4] Hutter, Frank, et al. "ParamILS: an automatic algorithm configuration framework." Journal of artificial intelligence research 36 (2009).
>
> [5] Pérez Cáceres, L., et al. "An experimental study of adaptive capping in irace: Supplementary material." (2017).

---

### Note · Authors · 2025-08-12

We sincerely thank the Area Chair and all reviewers for their thoughtful feedback and valuable suggestions.

We are encouraged by the recognition of our work's **valuable contribution** to adaptive configuration of cutting-plane parameters (Reviewer E4cm), the **efficient design** of our incremental triplet graph (Reviewer ZkPt), **significant performance gains** on various benchmarks (Reviewers ZkPt, rt7i, E4cm), and the **generalization to larger unseen instances** (Reviewer rt7i).

In response, we completed the following key additions:

1. **Enlarged separator parameter space** (from Reviewers Fjx6 & E4cm). We expanded our dynamically-tuned parameter set from a total number of 46 to 144, covering 5-7 internal hyperparameters for each of 22 cutting plane methods (e.g., Flow Cover, Gomory, MIR Separators) and spanning various numeric types (int, float, bool). The results show the extensibility and effectiveness of our method on a broader parameter space.
2. **Evaluation on MIPLIB 2017** (from Reviewer Fjx6). We evaluated our method on the full MIPLIB 2017 benchmark (240 instances), which is challenging due to its mixed-scenario problem classes and diverse problem structures. The results show that our method delivers notable improvements over the default solver setting.
3. **Improved generalization study** (from Reviewer ZkPt). We trained a general configuration model on a training dataset covering diverse instances from MIPLIB 2017, and evaluated it on four unseen, domain-specific datasets. The results demonstrate the good generalizability of our method.
4. **Comprehensive analytical study** (from Reviewers ZkPt, E4cm & Fjx6).  We added sensitivity analyses for *frequency* and *MaxRound*, evaluations of encoder alternatives, an ablation isolating our block positional encoding, and measurements of computational overhead, enabling a fuller assessment of our method.

In summary, we propose a general and effective framework for dynamically configuring separator parameters in the mixed-integer programming solver, with substantial performance gains across multiple MILP benchmarks.

We are grateful for the discussions that have substantially strengthened the paper. Once again, thank the Area Chair and all reviewers for their time and efforts throughout the review period.

---

### Decision · Program_Chairs · 2025-09-17

**Decision:**

Accept (poster)

**Comment:**

This paper introduces DynSep, a reinforcement learning-based framework aimed at enhancing a backbone MILP solver through dynamic separator configuration. The approach integrates an incremental subgraph-based state representation into a transformer policy model, which autoregressively predicts both when to terminate separation and which separators to activate during branch-and-cut. Extensive empirical evaluation demonstrates that the method significantly accelerates the SCIP solver across diverse datasets, while also exhibiting strong generalization to unseen MILP instances.

There is a clear consensus among the reviewers that the proposed method is technically sound, effective, and delivers notable performance improvements. The extensive experimental results, generalization capability, and the inclusion of interpretability analysis are all strengths of the paper. Most of the key claims are well substantiated through ablation studies. No critical flaws are found. However, the main limitation lies in the lack of study on the broader applicability of the method, particularly its applicability to commercial solvers such as Gurobi, and a wider range of branch-and-cut parameters. A further concern is the limited discussion on the effectiveness of the incremental triplet graph compared to full-graph encoding. While the paper attributes its use to improved inference speed, ablation results show that it also improves performance. The authors are encouraged to revise the paper to include a more detailed analysis explaining this.

Despite these cons, the reviewers agree that the paper makes a meaningful contribution to the field. The authors' rebuttal addressed the technical and empirical concerns effectively, providing additional evaluations that further validate the approach. Following the discussion, all reviewers recommend acceptance. The required revisions are expected to be manageable in the camera-ready version. I am happy to recommend acceptance, provided that the camera-ready version includes the promised experiments and adequately addresses the above weaknesses.